# Sensitivity of the Eocene Climate to CO₂ and Orbital Variability

John S. Keery[1], Philip B. Holden[1], Neil R. Edwards[1]

[1]School of Environment, Earth & Ecosystem Sciences, The Open University, Milton Keynes, MK7 6AA, UK

*Correspondence to*: John S. Keery (john.keery@open.ac.uk)

**Abstract.** The early Eocene, from about 56 Ma, with high atmospheric $CO_2$ levels, offers an analogue for the response of the Earth's climate system to anthropogenic fossil fuel burning. In this study we present an ensemble of 50 Earth system model runs with an early Eocene palaeogeography and variation in the forcing values of atmospheric $CO_2$ and the Earth's orbital parameters. Relationships between simple summary metrics of model outputs and the forcing parameters are identified by linear modelling, providing estimates of the relative magnitudes of the effects of atmospheric $CO_2$ and each of the orbital parameters on important climatic features, including tropical-polar temperature difference, ocean-land temperature contrast, Asian, African and S. American monsoon rains, and climate sensitivity. Our results indicate that although $CO_2$ exerts a dominant control on most of the climatic features examined in this study, the orbital parameters also strongly influence important components of the ocean-atmosphere system in a greenhouse Earth. In our ensemble, atmospheric $CO_2$ spans the range 280 - 3000 ppm, and this variation accounts for over 90% of the effects on mean air temperature, southern winter high-latitude ocean-land temperature contrast and northern winter tropical-polar temperature difference. However, the variation of precession accounts for over 80% of the influence of the forcing parameters on the Asian and African monsoon rainfall, and obliquity variation accounts for over 65% of the effects on winter ocean-land temperature contrast in high northern latitudes, and northern summer tropical-polar temperature difference. Our results indicate a bimodal climate sensitivity, with values of 4.36°C and 2.54°C, dependent on low or high states of atmospheric $CO_2$ concentration respectively, with a threshold at approximately 1000 ppm in this model, and due to a saturated vegetation-albedo feedback. Our method gives a quantitative ranking of the influence of each of the forcing parameters on key climatic model outputs, with additional spatial information from singular value decomposition providing insights into likely physical mechanisms. The results demonstrate the importance of orbital variation as an agent of change in climates of the past, and we demonstrate that emulators derived from our modelling output can be used as rapid and efficient surrogates of the full complexity model, to provide estimates of climate conditions from any set of forcing parameters.

## 1 Introduction

In the early Eocene several episodes of global warming coincided with carbon isotope excursions (CIEs), pulses of isotopically light carbon injected into the atmosphere and oceans, and recorded in high-resolution marine and terrestrial sediments (Kennett and Stott, 1991). In one large CIE, at the Palaeocene-Eocene transition at ~56 Ma, the Palaeocene-

Eocene Thermal Maximum (PETM), evidence from both tropical (e.g. Zachos et al., 2003) and polar (e.g. Sluijs et al., 2006) regions indicates that temperatures increased by ~5°C in less than 10 kyr. Although the greenhouse gas (GHG) sources, and the duration of the onset phase of the PETM are uncertain, the relatively short time scale and global extent of the PETM strongly suggest that a large and sudden increase in GHGs in the atmosphere was the primary climatic forcing factor (Zachos et al., 2007). Since the PETM is the most recent period in Earth's history for which estimated atmospheric GHG concentrations are similar in magnitude to those of the present-day, and expected to arise from fossil fuel burning, the PETM may provide a valuable analogue for anthropogenic climate change (e.g. McInerney and Wing, 2011; Zeebe et al., 2016; Zeebe and Zachos, 2013).

The CIEs of the early Eocene show similar regularity in their timing to periodic changes in the Earth's orbit around the sun (Lourens et al., 2005), and the search for causal relationships between orbital cycles and Paleogene climate is an active area of research (e.g. Lauretano et al., 2015; Laurin et al., 2016; Lunt et al., 2011).

Although the climatic state in the early Eocene cannot be directly measured, much information on temperature and biogeochemical conditions can be inferred from measurements of proxy data: preserved natural records of climate variability, which can be linked to the property of interest through physical processes (Jones and Mann, 2004). But there are major uncertainties in proxy data from the Eocene due to incomplete preservation and alteration over time, with additional uncertainties as to the seasonality of contributory processes, and for ocean proxies, the depth at which the property of interest, e.g. temperature, influences the proxy (Dunkley Jones et al., 2013). Climate models therefore have an important role to play in exploring the mechanistic functioning of palaeoclimates (Huber, 2012).

Climate simulations with high temporal and spatial resolution can be obtained from General Circulation Models (GCMs), but the requirement of GCMs for powerful computers and long run-times makes them difficult to deploy for large ensembles of model simulations and restricts their ability to investigate the large uncertainties in forcings and model parameterisations. Such ensembles are more practical with more heavily parameterised and hence more computationally efficient Earth system Models of Intermediate Complexity (EMICs), (Weber, 2010), although we note that Araya-Melo et al. (2015) and Lord et al. (2017) have deployed the GCM HadCM3 in ensemble-based studies of orbital forcing effects on climates of the Pleistocene and late Pliocene respectively.

In this study we deploy an EMIC, PLASIM-GENIE (Holden et al., 2016), in an ensemble of model runs to investigate the effects of varying GHG concentration and orbital parameters on the palaeoclimate of the Earth, with an Eocene configuration of the oceans and continents. We reduce the dimensionality of the model output by computing simple scalar metrics to denote key climatic features of each ensemble member, and we apply singular value decomposition (SVD) to identify the principal components (PCs) of temperature and precipitation fields in the full ensemble, for comparison with the variation in the forcing parameters.

By applying the linear modelling and emulation methods of Holden et al. (2015), we regress both the simple scalar metrics and the SVD reduced dimension model outputs onto the forcing parameters, and from the derived relationships, we infer main effects denoting the effect of each explanatory term in the linear model, and total effects denoting the effect of each

forcing parameter, on the variation in the scalar metrics and on the temperature and precipitation output fields. We demonstrate that emulators derived in respect of tropical precipitation metrics can be used to estimate Eocene monsoonal responses to any combination of GHG and orbital forcing parameter values.

## 2 The Early Eocene and the PETM

### 2.1 Climate of the Early Eocene

During the Eocene, the Earth remained in the 'greenhouse' state, which had persisted since the early Cretaceous, with polar air temperatures remaining above 0°C for most of the year (Wing and Greenwood, 1993), no permanent polar ice-caps, reduced equator-pole temperature gradients, and lower ocean-land temperature contrasts, inferred from fossil and isotope indicators of temperature and environmental conditions. Climate modellers have experienced difficulty in simulating Cretaceous and Palaeogene 'equable climates' (Sloan and Barron, 1990; Wing and Greenwood, 1993) with sufficient warming at high latitudes, without overheating the tropics, although Huber and Caballero (2011), hereafter HC11, have demonstrated that with sufficiently high levels of $CO_2$ (as a proxy for all forms of radiative forcing), climate models can generate global air temperature distributions in broad agreement with the proxy temperature measurements.

The onset of the PETM, at approximately 55.9 Ma (Westerhold et al., 2009), is recognised as the boundary between the Palaeocene and Eocene epochs (Aubry et al., 2007), and is characterised by a large CIE, indicating large GHG emissions, accompanied by a sudden rise in global temperature (Kennett and Stott, 1991), extensive extinction and origination of nannoplankton (Gibbs et al., 2006), and widespread ocean anoxia (Dickson et al., 2012). There is some evidence from analysis and modelling of the timing and duration of variations in $\delta^{13}C$ and $\delta^{18}O$ observed in nannoplankton fossils that some of the GHG emissions were initially in the form of $CH_4$ (Dickens, 2011; Lunt et al., 2011; Thomas et al., 2002), which is rapidly oxidised in the atmosphere to $CO_2$. The PETM is also marked by enhanced precipitation and continental weathering (Carmichael et al., 2016; Chen et al., 2016; Penman, 2016), rapid and sustained surface ocean acidification (Penman et al., 2014; Zachos et al., 2005), and shares many features of the global-scale oceanic anoxic events of the Cretaceous and Jurassic periods (Jenkyns, 2010). See McInerney and Wing (2011) for a review of PETM research.

The duration of the onset phase of the PETM is uncertain. Cui et al. (2011) have suggested that the peak rate of addition of $CO_2$ to the atmosphere was much lower than the present-day rate of anthropogenic GHG emissions, but this is disputed by Sluijs et al. (2012). Zeebe et al. (2016) have estimated that the initial release of carbon at the onset of the PETM lasted at least 4 kyr, at a rate which was little more than one tenth of the present rate of anthropogenic emissions, so the Earth may already be in a 'no-analogue' state, with anthropogenic climate change likely to exceed that of the PETM. However rapid the onset, the greenhouse conditions of the early Eocene, and particularly the PETM, provide an opportunity to apply lessons from the past, with a view to improving predictions of the future (Lunt et al., 2013).

**2.2 Palaeogeography of the Early Eocene**

The arrangement of the continents and oceans in the Early Eocene was broadly similar to that of the present, with the Earth's land mass divided into the same major continents, and with most of the land mass in the northern hemisphere. India had not yet collided with the Eurasian continent, and the closure of the Tethys Ocean was not yet complete. Such tectonic movements may have effected some changes to the climate system. In particular, the configuration of ocean gateways strongly influences modes of ocean circulation, and hence affects energy transport throughout the climate system (Lunt et al., 2016; Sijp et al., 2014).

**2.2.1 Continental and Ocean Configurations during the Early Eocene**

Although the Bering Strait was closed throughout the Palaeogene (Marincovich et al., 1990), and the Western Interior Seaway linking the Arctic to the Pacific was closed by the end of the Cretaceous (Slattery et al., 2015), the Arctic Ocean was connected to the major oceans during the early Eocene through the Turgai Strait, also known as the Western Siberian Seaway (Akhmetiev et al., 2012; Radionova and Khokhlova, 2000). The Lomonosov Ridge, from which core samples have been obtained by the Arctic Coring Expedition (ACEX) of the Integrated Ocean Drilling Program Expedition (IODP) 302 (Backman et al., 2008), was on the edge of the Arctic basin rather than across the pole as in the present configuration (O'Regan et al., 2008).

Both the Drake Passage between South America and Antarctica (Barker and Burrell, 1977) and the Tasman Gateway between Australia and Antarctica (Exon et al., 2004) were closed during the early Eocene, preventing the development of an Antarctic Circumpolar Current and allowing greater southern hemisphere meridional heat transport than in the modern world.

**2.2.2 Orbital Configurations**

Throughout Earth's geological history, oscillations in the relative positions of the Earth and Sun have influenced both the Earth's climate, and rates of sedimentation in some climate-sensitive environmental settings (Hinnov and Hilgen, 2012). The main oscillations are the eccentricity of the Earth's orbit around the Sun, with periods of ~100 kyr and 405 kyr, the obliquity or tilt of the Earth's axis of rotation, with a period of ~40 kyr, and precession, the relative timing between perihelion and the seasons, with a period of ~20 kyr (Berger et al., 1993). By correlating oscillations preserved in the geological record with computed time series of changes in insolation received by the Earth, an absolute astronomical time scale may be constructed for recent time-spans with a complete sedimentary record, but where the geological evidence is incomplete, or where uncertainties in the orbital model are too great further back in time, only a relative time scale may be derived (Hilgen et al., 2010). An absolute astronomical solution has been computed back to 50 Ma (Laskar et al., 2011), and an absolute age of 55.53 ±0.05 Ma has been proposed for the onset of the PETM at the start of the Eocene epoch by Westerhold et al. (2012).

Lourens et al. (2005) noted the apparent astronomical pacing of global warming events in the late Palaeocene and early Eocene, with correlations to both the long and short periods of eccentricity. Sexton et al. (2011) suggested that although the smaller hyperthermal events of the early Eocene were driven by cycles of carbon sequestration and release in the ocean, paced by the eccentricity cycles, the PETM was likely to have been driven by carbon injection from a sedimentary source. Laurin et al. (2016) applied a method which allows the phase of the 405 kyr eccentricity cycle to be identified from interference patterns and frequency modulation of the ~100 kyr eccentricity cycle, and concluded that four hyperthermals in the early Eocene were initiated at 405 kyr eccentricity maxima, but in a study of terrestrial sediments with apparent correlation to the ~100 kyr eccentricity cycle, Smith et al. (2014) suggested that hyperthermals occurred during eccentricity minima, rather than maxima.

## 3 Methods

### 3.1 The PLASIM-GENIE Model

PLASIM-GENIE (Holden et al., 2016) is an intermediate complexity AOGCM. We apply the model at a spectral T21 atmospheric resolution, which corresponds to a triangular truncation applied at wave number 21 and a horizontal resolution of 5.625°, with 10 layers, and a matching ocean grid with 32 depth levels. We apply the calibrated parameter set of Holden et al (2016). The component modules are as follows:

PLASIM (Fraedrich, 2012) is built around the 3D primitive equation atmosphere model PUMA (Fraedrich et al., 2005). The radiation scheme considers two wavelength bands in the short wave and uses the broad band emissivity method for long wave. Fractional cloud cover is diagnosed. Other parameterised processes include large-scale precipitation, cumulus and shallow convection, dry convection and boundary layer heat fluxes.

GOLDSTEIN is a 3D frictional-geostrophic ocean model (Edwards and Marsh, 2005; Marsh et al., 2011), dynamically similar to classical GCMs, except that it neglects momentum advection and acceleration. Barotropic flow around the four continental islands (Fig. 1) is derived from linear constraints that arise from integrating the depth-averaged momentum equations.

GOLDSTEINSEAICE (Edwards and Marsh, 2005) solves for the fraction of the ocean surface covered by ice within a grid cell and for the average sea-ice height. A diagnostic equation is solved for the ice surface temperature. Growth or decay of sea ice depends on the net heat flux into the ice (Hibler III, 1979; Semtner Jr, 1976). Sea-ice dynamics are represented by diffusion and advection by surface currents.

ENTS (Williamson et al., 2006) models vegetative and soil carbon densities, assuming a single plant functional type. Photosynthesis depends upon temperature (with a double-peaked response representing boreal and tropical forest), atmospheric $CO_2$ concentration and soil moisture availability. Self-shading is parameterised. Land surface albedo, moisture bucket capacity and surface roughness are parameterised in terms of the simulated carbon pool densities.

The computational efficiency of PLASIM-GENIE is achieved mainly through low spatial resolution (~5°) and, relative to

high-complexity Earth system models, simplifying assumptions in physical processes.  These include, for instance,

simplified parameterisations of radiative transport and convection in the atmosphere, the neglect of momentum transport in

the ocean, and the representation of all vegetation as a single plant functional type.  Climate sensitivity, the response of the

climate to a doubling of atmospheric $CO_2$ concentration, including feedbacks, is an emergent property of the model.

**3.2 Model Configuration**

**3.2.1 Model Grid**

This study was designed before Lunt et al. (2017) presented their 'DeepMIP' guidelines for model simulations of the latest

Paleocene and early Eocene.  However, our palaeogeography is based on the high-resolution digital reconstruction of the

early Eocene published by Herold et al. (2014), and which Lunt et al. (2017) recommended should be used as the standard

for all palaeoclimate simulations within the DeepMIP framework.  We have used the dataset of Herold et al. (2014) as an

initial configuration for the tectonic layout, topography and bathymetric boundary conditions in our study.  We have reduced

the resolution of the Eocene palaegeography provided by Herold et al. (2014) to a configuration of 64 longitude x 32 latitude

cells, with each cell representing 5.625° in each orientation. Cells at high latitudes therefore represent smaller land areas than

cells at low latitudes.  Our vertical resolution is 32 ocean depths and 10 atmospheric layers.  We have incorporated the ocean

gateway configurations discussed in section 2.1.1.  The Turgai Strait is open in our configuration, and is the only connection

between the Arctic Ocean and other oceans.  The Drake Passage and Tasman Gateway are both closed.

The palaeogeography (Fig. 1) comprises four land masses: N America and Eurasia; Antarctica combined with S America and

Australia; Africa; and India.  Red rectangles in Fig. 1 indicate the boundaries of areas used to calculate simple metrics of

centennially averaged seasonal precipitation, as empirical indicators of African, Asian and S. American monsoons.

**3.2.2 Forcing and Other Input Parameters**

In order to investigate the sensitivity of the Eocene climate to variation in atmospheric $CO_2$ and orbital parameters, we have

constructed an ensemble of 50 model configurations, each with a unique set of forcing parameters comprising atmospheric

$CO_2$, eccentricity ($e$), obliquity ($\varepsilon$) and precession ($\omega$), the angle on the Earth's orbit around the Sun between the moving

vernal equinox and the longitude of perihelion (Berger et al., 1993).  When $e$ is zero, the Earth's distance from the Sun is

constant at all points on the orbit, so there is no precessional effect.  The magnitude of precessional effects is controlled by $e$,

while phase is controlled by $\omega$, so precessional effects are commonly described by the precession index given by $e\sin\omega$.  The

precession index is at its maximum value when perihelion occurs at the December solstice, its minimum value when

perihelion is at the June solstice, and has a value of 0.0 when perihelion is at either the March or September equinox.  The

only orbital parameter which alters the total annual solar radiation received by the Earth is $e$, although the range of variation

is very small.  We include $e$ and $\omega$ as separate and independent forcing parameters, rather than combined as the precession

index, or in the form $e\cos\omega$. An additional dummy parameter is included to test for possible overfitting of relationships

between forcing parameters and model output fields.

Although the maximum mass of $CO_2$ injected into the atmosphere during CIEs, and in particular the PETM, remains

uncertain, there is broad agreement that the atmospheric concentration of $CO_2$ did not exceed 3000 ppm (e.g. Gehler et al.,

2016), and that it did not fall below the pre-industrial level of 280 ppm at any time during the early Eocene. We allocate

these values as the limits of a uniform range from which our ensemble of $CO_2$ values is selected.

Since the absolute astronomical time scale for the early Eocene has an uncertainty which is greater than the periods of the

obliquity and precession cycles, and there remains disagreement as to which phases of the eccentricity cycles are related to

CIEs, there are no combinations of the orbital forcing parameters which can be known a priori to be of greater importance in

their effects on the Eocene climate in general, and on their contributions to the initiation, duration and termination of the

CIEs in particular. We therefore select values of orbital parameters independently, and from the full range of each

parameter's variation during the early Eocene.

To ensure the best coverage of the five-dimensional state-space comprised of the four forcing parameters and the additional

dummy parameter in a limited number of model runs, we apply the Latin hypercube method (McKay et al., 1979), a

constrained Monte Carlo sampling scheme in which the range to be sampled for each variable is divided into non-

overlapping intervals, and one value from each interval is randomly selected (Wyss and Jorgensen, 1998). This provides

adequate coverage of the state space more efficiently than can be achieved by a simple Monte-Carlo sampling approach

(Rougier, 2007). The present study has been designed to facilitate direct comparison between the results for specific

ensemble members and their direct counterparts in a future study using the EMIC model GENIE-1 (Edwards and Marsh,

2005), which will include additional forcing parameters not used by this PLASIM-GENIE study. We have applied an

iterative method to generate a pair of corresponding hypercubes with five and eleven dimensions for the PLASIM-GENIE

and GENIE-1 studies respectively, in which the minimum Euclidean distance between any two points is maximised, and

linear correlation between any two parameters is minimised. We note that our selection of values for $\omega$, an angular

parameter, is from 0-360°, treated as a linear range, with the consequence that the maximin criterion within the Latin

hypercube algorithm is incorrectly calculated. However, given the dimensionality of our experimental design, this is

unlikely to result in a significant reduction in the efficiency with which design points are distributed throughout the very

sparsely populated state-space. We draw readers' attention to an approach presented by Bounceur et al. (2015), in which

independent values of $e\sin\omega$, $e\cos\omega$ and $\varepsilon$ are sampled, with rejection of absolute values of $e\sin\omega$ and $e\cos\omega$ which equal or

exceed the maximum value of $e$. This experimental design allows values of $e$ and $\omega$ for any design point to be identified by

trigonometric analysis, while efficiently sampling the state space. Details of the steps taken to generate the hypercubes are

provided in Appendix A. The absolute value of the r correlation coefficient does not exceed 0.1 for any pair of input

(forcing and dummy) parameters. Uniform ranges for each of the forcing parameters and the dummy parameter are shown in

Table 1, and the values applied in all 50 PLASIM-GENIE ensemble members are shown in Table 2.

The intensity of radiation emitted by the Sun has increased steadily over time, and we apply the linear model of Gough (1981), and select a solar constant of 1358.68 W m$^{-2}$. We note that Lunt et al. (2017) have recommended that a modern value of 1361.0 W m$^{-2}$ should be applied to studies within the DeepMIP framework, in order to facilitate comparison between simulations with modern and pre-industrial levels of $CO_2$, and to offset the absence of elevated levels of $CH_4$.

### 3.2.3 Running the Models

Each simulation was run for a spin-up period of 1000 years to reach a quasi-steady state, with key output fields recorded as seasonal averages for each of the three-month periods December, January and February (DJF) and June, July and August (JJA), representing both winter and summer seasons in both the northern and southern hemispheres. Although model output includes time series of some fields and output values every 100 years, in this study only the field values recorded at the end of the 1000 years of modelling are used for analysis of the results.

### 3.3 Analysis of Model Output

Comparison of the forcing parameters applied in the ensemble with the model output fields can be more efficiently achieved by reducing the dimensionality of the model output while retaining information on key components of the climate system.

### 3.3.1 Simple Metrics

In studies of the Earth's modern climate, it is recognised that the tropical-polar temperature difference (TPTD) influences poleward energy flux, and the ocean-land temperature contrast (OLC) affects monsoon intensity (Jain et al., 1999; Karoly and Braganza, 2001; Peixoto and Oort, 1992). Although atmospheric circulation patterns in the early Eocene will have differed from those in the modern world, in selecting latitude regions to represent the TPTD, we adopt the approach of Abbot and Tziperman (2008), who configured their model of the Cretaceous climate with latitude ranges of 0–30°, 30–60°, and 60–90°, the approximate boundaries of the Hadley, Ferrel and Polar cells observed in the modern world (Peixoto and Oort, 1992). On our model grid in which each cell spans 5.625° of latitude, for the purposes of deriving scalar metrics, we define the tropical regions to be between 0.0° and 33.75° North and South, and the polar regions to be between 56.25° to 90° North and South.

From the output values of air temperature in the lowest level of the atmosphere, weighted by grid cell area, we derive scalar values for each model run, of global annual mean air temperature (MAT), northern and southern hemisphere seasonality (mean area-weighted DJF-JJA temperature differences in the above-defined polar regions), TPTD for summer and winter in each hemisphere, and OLC for summer and winter in tropical and polar regions in each hemisphere.

Monsoons are related to seasonal variations in tropical and subtropical winds and precipitation (Trenberth et al., 2006). Wang and Fan (1999) noted that the choice of an index to denote monsoon behaviour in the modern world is difficult and arbitrary, with commonly applied indices based on average summer precipitation, maximum summer precipitation, winter-summer difference in precipitation, or wind circulation patterns within defined geographical areas. In this study, we derive

simple scalar metrics to denote indices for monsoons for Asia, Africa and South America by subtracting winter rainfall from

summer rainfall, for defined geographical regions, denoted on Fig. 1, and selected for their similarity to monsoonal regions

in the modern continental configuration.

**3.3.2 Singular Value Decomposition, Linear Modelling and Model Emulation**

We perform a singular value decomposition to identify the PCs and empirical orthogonal functions (EOFs) of temperature

and precipitation fields in the full ensemble , although we note that climate variability may not be due to physical processes

which vary orthogonally, and identification of PCs can be influenced by aspects of the experimental design. A detailed

presentation of the use of this method in the analysis of climate data is given by Hannachi (2004).

We use the linear modelling method of Holden et al. (2015), to regress both the simple scalar metrics and the SVD reduced

dimension model outputs onto the forcing parameters. Values of the forcing parameters $CO_2$, $e$ and $\varepsilon$ (with its very small

angular range considered to be approximately linear) were normalised to the range [-1, 1] and combined with $\sin\omega$ and $\cos\omega$

to form 50-element column vectors representing the forcing factors. Each 2-D (32 x 64) result field for each ensemble

member was unrolled to form a column vector of 2048 elements, comprising a single column within a 2048 x 50 matrix of

full ensemble values.

SVD was applied to decompose the full ensemble matrix for each 2-D result field, providing a 2048 x 50 matrix of PCs, a 50

x 50 matrix of PC scores, and a 50 x 50 matrix of diagonal values.

Linear modelling was applied to determine relationships between the normalised forcing factors and the first six columns of

the PC scores, including products of pairs of forcing factors, and squares of each forcing factor, with the best fitting

relationships selected according to the Akaike information criterion (Akaike, 1974) then refined using Bayes information

criterion (Schwarz, 1978). Burnham and Anderson (2003) provide a detailed discussion of the application of information

criteria in model selection. The resulting relationship provides a simple emulator which can be used to estimate a PC score

for the 2-D model field, given a single set of forcing parameter values. Applying derived emulators in respect of temperature

and precipitation for both seasons, demonstrated high correlation between emulated PC scores and PC scores derived

directly through SVD (Table 3).

Our emulator approach uses linear regression, rather than a Gaussian process (GP), and is therefore simpler than the methods

applied by Bounceur et al. (2015) in a study of the response of the climate-vegetation system in interglacial conditions to

astronomical forcing, and by Araya-Melo et al. (2015) in their study of the Indian monsoon in the Pleistocene. Unlike linear

models, GP models are intrinsically stochastic and give a more accurate quantification of their own error in emulating the

input data. However, GP models can become computationally demanding in high dimensional space, and their results can be

more difficult to interpret.

In order to analyse the results of each of our linear models, we apply the method described in detail by Holden et al. (2015)

to derive the main effects (Oakley and O'Hagan, 2004), which provide a measure of the variation in the linear model output

due to each of the terms (first order, second order and cross products), derived from their coefficients, and total effects

(Homma and Saltelli, 1996), which separate the effect of each forcing parameter on the variation in the model output. Although the forcing factors are all scaled within the range [-1, 1], the trigonometrical precession terms are not uniformly distributed across this range. We have therefore computed the variances of the first order, second order and cross product terms directly for all parameters, rather than applying the respective approximations of $\frac{1}{3}$, $\frac{1}{9}$ and $\frac{4}{45}$, and we have applied these values as scaling factors in calculating the main effects and total effects.

## 4 Results

### 4.1 Model Output - Temperature and Precipitation

Analysis of the model results has focused on variation in surface air temperature and precipitation in both winter and summer in each hemisphere, although it should be noted that our experiment has not been designed such that mean values in our ensemble output represent direct estimates of the Eocene climate mean. In the left column of Fig. 2, median temperatures at each grid cell for the full ensemble are plotted for DJF (top) and for JJA (bottom), with the standard deviations plotted in the right column.

Ranges of median temperatures over land are greater than over the oceans, but TPTD is smaller in both seasons and both hemispheres than simulated in the modern world (see Fig. 2, Holden et al 2016). It is apparent from the standard deviation field that the tropical-polar temperature difference varies substantially across the ensemble, particularly in northern winter. The temperature distributions are similar to those of the 2240 ppm $CO_2$ simulation of HC11, regarded as their "mid to late Eocene" analogue (they consider elevated $CO_2$ as a proxy for all radiative forcing, including uncertain climate sensitivity). The principal difference is in high northern latitude winter temperatures; the Arctic ocean remains above freezing in HC11. We note that the Arctic winter median air temperature is below freezing over both land and sea in the PLASIM-GENIE ensemble, (see Fig 3) and the Arctic does not remain ice-free throughout the year in any of the 50 simulations in our study. Tropical temperatures in excess of 35°C were simulated in some cases, as in HC11, which they regarded as their "most troubling result", although they note observational data is currently insufficient to rule this out. Finally, we note that multi-model ensembles have found significant inter-model differences including, for instance a 9°C spread in global average temperature under the same $CO_2$ forcing (Lunt et al 2012). Quantification of model-related uncertainty is beyond the scope of the present study.

Full ensemble distributions of mean latitudinal distributions of annual mean sea surface temperature (SST), with mean latitudinal distributions of maritime and continental surface air temperature in both DJF and JJA are plotted in Fig. 3, together with ensemble medians and 5% and 95% percentiles of global annual mean SST, and maritime surface air temperature in both DJF and JJA. The greater range of temperatures below rather than above median values reflects our use of a uniform range of $CO_2$ forcing values, and the logarithmic response of temperature to increasing $CO_2$ concentration. There is substantial variation of mean temperature across the ensemble, around 20 degrees over land, but the temperature

offset varies little with latitude outside of polar regions where snow and ice greatly reduce winter temperatures in the colder

simulations. The variation in TPTD across the ensemble thus appears to be essentially driven by the strength of snow and ice

albedo feedbacks.

Our ensemble distributions of sea and air temperatures are in broad agreement with the values from the Eocene model

studies compared by Lunt et al. (2012), hereafter L12, and with the tables of marine and terrestrial proxy data compiled by

L12Lunt et al. (2012) and HC11, covering the early Eocene, and including some records from the very latest Paleocene, but

not including the PETM. Our palaeogeography specifically represents the early Eocene, but our range of $CO_2$ and orbital

inputs is more representative of the variation in forcing across the whole era. L12 have summarised variations of SST with

latitude from their proxy data set, in their Fig. 1, including large error bars representing uncertainty which they attribute to

assumptions about seawater chemistry, possible non-analogous behaviour between modern and ancient systems, and

uncertainty in calibrations of relationships between proxy data and properties of the palaeoclimate. Our median values of

SST are close to the median estimates of SST in L12 at mid latitudes, and well within the uncertainty indicated by error bars

at high latitudes.

Median values and standard deviations of precipitation at each grid cell are plotted in Fig. 4. Higher precipitation values and

variation are largely confined to the tropics, especially to regions associated with monsoons in the present day: Africa and S.

America in DJF, and S.E. Asia in JJA.

**4.2 Simple Metrics**

In Figs. 5 and 6, $CO_2$, obliquity ($\varepsilon$) and precession index ($e\sin\omega$) are plotted against MAT, northern seasonality, northern

winter TPTD and northern summer TPTD (Fig. 5), and southern winter polar OLC, northern winter polar OLC, Asian

monsoon index, African monsoon index and American monsoon index (Fig. 6). Subplots for obliquity and precession index

in Figures 5 and 6 denote the $CO_2$ level on a continuous colour scale. The dominant effect of $CO_2$ on MAT and northern

seasonality is apparent in Fig. 5, and it can also be seen that $CO_2$ strongly affects the northern TPTD in the winter, but not in

the summer, when the combined influence of obliquity and precession index is discernible, suggesting that temperature

proxies with seasonal bias may have a significant orbital imprint. The plot of atmospheric $CO_2$ against N. Winter TPTD

shows a change in gradient at approximately 1000 ppm $CO_2$ and 32°C. This may be related to the logarithmic dependence of

radiative forcing on $CO_2$ concentration, the disappearance of ice above some threshold level, and a minimum level of land

surface albedo related to maximum vegetation cover. A possible sea ice related threshold mechanism influencing both SST

and maritime air temperature in high northern latitudes may be observed in Fig. 3, and this is strongly associated with the

increase in northern winter TPTD at low $CO_2$ levels. Zeebe et al. (2017) have analysed a high resolution benthic isotope

record covering the late Palaeocene - early Eocene, and have concluded that orbitally paced cycles are unlikely to have been

driven by high latitude mechanisms, but our PLASIM-GENIE modelling suggests that while northern TPTD is not orbitally

paced in the winter, being controlled by $CO_2$, it is orbitally paced in the summer, by a combination of obliquity and

precession.

It can be observed in Fig. 6 that there is strong correlation between $CO_2$ and southern winter polar OLC. The African and

Asian monsoon indices are both correlated with the precession index, a well established feature of Quaternary records (e.g.

Cruz et al., 2005). The American monsoon index is fairly strongly correlated with the precession index at high levels of

$CO_2$, and negatively correlated with $CO_2$ at low levels of $CO_2$. In each of the other examples, there is no apparent correlation

between the simple metric and two of the three forcing factors. We have selected these simple metrics with visible

correlations to the forcing parameters for further analysis with the linear modelling and emulation methods. Total effects on

the simple metrics have been calculated for each of the forcing parameters, with eccentricity and precession considered

separately, rather than combined within the precession index, and are shown in Table 4.

The total effects of $CO_2$ on MAT, northern winter TPTD and southern winter polar OLC, and of precession on both the

Asian and African monsoon indices are all very high (> 0.90), and the total effects of obliquity on northern winter polar OLC

and northern summer TPTD, are both fairly high (> 0.65), providing quantitative confirmation of the correlations visible in

Figs. 5 and 6.

### 4.3 Climate Sensitivity and Mean Air Temperature

Figure 7 shows the relationship between $CO_2$ (plotted on a logarithmic scale), and MAT, with an abrupt change of gradient

clearly visible at a $CO_2$ concentration of 1000 ppm. From the two gradients, we derive climate sensitivity values for a

doubling of $CO_2$ concentration at $CO_2$ levels below 1000 ppm, and at $CO_2$ levels above 1000 ppm, of 4.36°C and 2.54°C

respectively. We note that our modelled values of carbon in vegetation in the ENTS module remain low outside of the

tropics at low $CO_2$ concentration, but as $CO_2$ concentration increases, land areas at higher latitudes reach maximum values of

carbon in vegetation, with all land areas showing no further capacity for increased carbon in vegetation at an atmospheric

concentration of ~1000 ppm. The increase in land vegetation cover, with corresponding reduction in albedo, acts as a

positive feedback to rising temperature caused by increasing $CO_2$, but this feedback mechanism ceases to operate when all

available land is at its maximum vegetation capacity, with a consequent reduction in the climate sensitivity.

For a pre-industrial atmospheric $CO_2$ concentration of 280 ppm, the value of MAT indicated by our results for our early

Eocene palaeogeography is 14.0°C. Holden et al. (2016) applied an identically configured PLASIM-GENIE to a modern

geography, and their results show that with a pre-industrial $CO_2$ concentration, the model climate sensitivity is 3.8°C, and

MAT is 12.9°C.

Our results also indicate values of global MAT for double, and four-times pre-industrial levels of $CO_2$ of 18.5°C and 22.5°C

respectively; both these values are within the ranges of results for land near-surface air temperature in the modelling studies

compared by L12, and shown in their Fig. 2b.

### 4.4 Singular Value Decomposition

Figure 8 shows the first three PCs of surface air temperature in DJF and JJA, with the percentages of temperature variation

explained by each PC. Each of these plots illustrates the PC scaled by the standard deviation of the PC scores, thereby

reflecting the variability across the ensemble. Note the variable scales for each of the subplots. In both DJF and JJA, PC1 explains over 95% of the variance, with TPTD clearly visible in both hemispheres in DJF, but apparent only in the southern hemisphere in JJA. OLC is apparent in the plots of PC1 in both DJF and JJA. OLC is discernible in PC2 for DJF temperature, which explains 2.4% of variance, but less apparent, at least in the southern hemisphere, for JJA temperatures, in which PC2 explains 2.6% of the variance. For temperature in both DJF and JJA, PC3 explains less than 1% of the variance, with some indication of TPTD and OLC in DJF, but only of weak OLC at high latitudes in JJA. It is worth noting that even though lower order PCs explain small percentages of global variances, these PCs are generally associated with specific regions where they are comparably important to the first PC.

In their presentation of the SVD method applied in this study, Holden et al. (2015) investigated the effects of orbital parameters on the Earth's climate in the present day, but without including $CO_2$ as a forcing parameter in their ensemble, and found that obliquity had a dominant effect on the PC score of annual average surface air temperature. In our study of the Eocene climate, $CO_2$ is strongly correlated with N. seasonality (Fig. 5), and obliquity is weakly correlated with TPTD in JJA (Fig. 5) and with OLC in DJF (Fig. 6). The first three PCs of precipitation in DJF and JJA are shown in Fig. 9. PC1 explains approximately 55% of the variance in both seasons, with PC2 and PC3 explaining over 20% and over 5% respectively, in both seasons. In both PC2 and PC3, areas of high seasonal contrast appear to correspond to areas which experience monsoons in the modern world.

Correlations between the PC scores of temperature and precipitation are provided in Table 5. The first PC scores of temperature, reflecting a global warming signal, are highly correlated with the first PC scores for precipitation, suggesting that these PCs reflect a strengthening of the hydrological cycle in response to warming. Similar considerations reveal connections between lower order PC scores, though we note that the 2nd (3rd) component of DJF temperature is associated with the 3rd (2nd) component of DJF precipitation. In order to address the drivers of these modes, we first consider the correlation coefficients, r, between forcing factors and the PC scores, shown in Table 6. These demonstrate that for each output there is a mode of variability driven by $CO_2$ and another mode driven by precession, suggesting they reflect global warming (and associated hydrological strength) and precessional forcing of the monsoon system.

There is strong correlation ($r^2 > 0.5$) between $CO_2$ and the first PC scores of temperature in DJF and JJA. There are also strong correlations between precession index and the third PC scores for DJF temperature, and between precession index and the second PC scores for JJA temperature.

$CO_2$ is strongly correlated with the first PC scores of precipitation in both DJF and JJA, and there is a strong relationship between precession index and the second PC scores of precipitation in both DJF and JJA. An increase in the second PC scores for JJA precipitation in the Asian monsoon region (Fig. 9) corresponds to a decrease in the second PC scores for JJA temperature (Fig. 8), and as already noted, the second PC scores for both temperature and precipitation in JJA are strongly correlated to the precession index. This temperature reduction during the Asian monsoon was also observed by Holden et al. (2014), and attributed to a reduction in incoming solar radiation associated with increased cloud cover and surface evaporation.

## 4.5 Linear Modelling and Emulation

The relationships between the forcing parameters (with precession expressed as both $\sin\omega$ and $\cos\omega$) and the simple metrics, and between the forcing parameters and the PC scores of 2-D fields, derived through linear modelling, include first and second order terms of forcing factors, together with products of forcing factors. In all cases most of the main effects are confined to the first order terms, and in no case does eccentricity have a significant effect independently of either of the precession terms. All significant effects of the precession terms are accompanied by a small effect of eccentricity.

In Fig. 10, we plot the main effects of the forcing parameters on the first three PCs of temperature and precipitation for DJF. Figure 11 shows the main effects of the forcing parameters on the first three PCs of temperature and precipitation plotted for JJA.

In both seasons, PC1 for temperature and precipitation can be almost entirely explained by $CO_2$, reinforcing the earlier conclusion that these describe a connected mode, global warming with associated effects on the hydrological cycle. The main effects also suggest connections between the modes of variability of temperature and precipitation in lower-order components. In both seasons, and apparent in both variables, there is a mode that is driven by precession; we interpret this as a monsoon signal, given precessional forcing and spatial patterns of rainfall that are characteristic of modern monsoons (Figs. 8 and 9). In JJA this is the second component of both variables. The mode is associated with precipitation variability of ~2.5 mm/day and temperature variability of ~3°C, with increased precipitation associated with a surface air cooling (note the negative correlation in Table 3, so that positive change in one field is associated with negative change in the other). In both cases, the local magnitude of variability is comparable to that driven by $CO_2$. In DJF the precessional signal is again apparent in the second mode of precipitation, but the third mode of temperature. This mode is notable, in that it drives changes in simulated precipitation over East Africa (5 mm/day) that exceed $CO_2$-driven variability. The remaining modes are more complex, and may not represent a clear mode of variability that can be straightforwardly attributed. For instance, the third-order mode of JJA temperature is driven by an interaction between $CO_2$ and obliquity, but in precipitation can be explained by a combination of precession and $CO_2$.

All of the terms in the linear models derived from the forcing factors and the three monsoon indices are shown in Table 7. The Asian and African models are dominated by precession terms, roughly equally distributed between first order $\sin(\omega)$ and the cross product of $e$ and $\sin(\omega)$, with $|\sin(\omega)|$ being approximately five times, and eight times larger than $|\cos(\omega)|$ for the Asian and African models respectively. The American model identifies significant influence of $CO_2$, in both the negative first order, and positive second order terms, with a similar magnitude of influence from combined precession terms, and with $|\sin(\omega)|$ being approximately three times larger than $|\cos(\omega)|$. All of the models have small contributions from first or second order, or cross products of $\varepsilon$, and from those terms of $e$, in addition to significant contributions from $e\sin(\omega)$. The terms in the models clearly reflect the relationships between the three monsoon indices and the two forcing factors $CO_2$ and $e\sin(\omega)$ shown in Fig. 6.

We apply these linear models as emulators to estimate values of monsoon indices corresponding to the full range of

precession ($\omega$), with eccentricity fixed at its high limit of 0.06, low and high values of $CO_2$ (300 ppm and 3000 ppm), and

low and high values of obliquity (22.0° and 24.5°). Precession index ($e\sin\omega$) and emulated values of the Asian, African and

American monsoon indices for all four combinations of high and low $CO_2$ and obliquity are plotted in Figures 12, 13 and 14

respectively. The elliptical form of each of the plots is controlled by model terms which include $\cos(\omega)$, and which identify

seasonal processes in the development of the monsoons. Running each of the emulators with all of the terms in $\cos(\omega)$

excluded, generates points on a straight line between each apex of the ellipses generated by the full emulator. In each of the

12 plots in Figs. 12-14, $\omega$ increases anticlockwise from a value of 0° in the centre of the lower arc of the ellipse (with

perihelion at the March equinox), through a value of 180° in the centre of the upper arc (with perihelion at the September

equinox). Relationships between the precession index and the monsoon indices which are visually suggested in Figure 6 are

shown with clear structure in Figures 12, 13 and 14. In each of the monsoon areas, the highest levels of precipitation occur

when perihelion coincides with the summer solstice, in June for the Asian monsoon in the Northern Hemisphere, and in

December for the African and American monsoons in the Southern Hemisphere. For the Asian and African monsoons,

precipitation is increased by high $CO_2$, particularly when perihelion is at the summer solstice, but for the American monsoon,

high $CO_2$ decreases precipitation. The plots of the emulated African and American monsoons (Figs. 13 and 14) show the

lowest and highest degrees of non-stationarity respectively, due to the relative magnitude of the $\cos(\omega)$ terms in the linear

models.

**5 Summary and Conclusions**

Our ensemble of 50 model runs of the EMIC PLASIM-GENIE has used an early Eocene palaeogeography incorporating

recent understanding of the configuration of the continents and ocean gateways, with climate forcing by a randomly selected

combination of atmospheric GHG emissions and orbital parameters for each model run. Relationships between forcing

parameters and scalar summaries of model results have been derived through linear modelling.

Given the input range of $CO_2$, our results show that, at the global scale, variability in patterns of surface air temperature is

strongly dominated by a single mode of variation with a strong imprint of TPTD, focused in northern winter, that is entirely

controlled by $CO_2$ (> 95% variance in both seasons). We note, however, that regions under the influence of monsoon

systems exhibit precession-driven temperature variability that is comparable in magnitude to the variability driven by $CO_2$

(in large part the high proportion of variance explained by the $CO_2$ mode arises because the signal is global). In contrast to

the unimodal dominance of $CO_2$ on the modelled global temperature fields, precipitation shows a somewhat more nuanced

response. The first mode of precipitation, while still controlled entirely by $CO_2$, is much less dominant (maximum 57%

variance in DJF cf 21% for PC2). In the second and third spatial modes of precipitation variability, $CO_2$ is still important, but

no more so than orbital parameters, with PC2 controlled more strongly by precession index.

The importance of orbital forcing to precipitation signals is seen more clearly in the OLC and monsoon indices. In spite of large variation in atmospheric $CO_2$, variation in obliquity accounts for well over half of the variation in high northern latitude ocean-land temperature contrast, and the variation in precession is the dominant influence on seasonal variation in precipitation in tropical Africa and Asia, and combines with $CO_2$ to influence seasonal precipitation in tropical America. Our results strongly suggest the presence of monsoons in the early Eocene, but these climatic features would have developed without the effects of orography and high altitude plateau heating which are important factors in the modern south Asian monsoon (Boos and Kuang, 2010).

We note that the relative amplitude of the $CO_2$-driven modes depends critically on the actual amplitude of $CO_2$ variability in the period of interest. While the ranges for orbital parameters are well defined, this is less true of $CO_2$ variability over the Eocene. If atmospheric $CO_2$ remained within a narrower range throughout the period, for example in the range 700 to 1800 ppm, indicated for the early Eocene by Anagnostou et al. (2016) in a recent study using boron isotopes, then outside of short-lived hyperthermals, the relative influence of $CO_2$ and orbital inputs might have been more evenly balanced. Our modelling results suggest that climate sensitivity is state dependent, with a value of 4.36°C in a low $CO_2$ state, and 2.54°C in a high $CO_2$ state, due to a positive feedback mechanism in which albedo reduces as vegetation increases to its maximum value when $CO_2$ concentration reaches 1000 ppm.

We have demonstrated that emulators derived from linear modelling of the PLASIM-GENIE ensemble results can be used as a rapid and efficient method of estimating climate conditions from any set of forcing parameters, without the need for further deployment of the EMIC.

PLASIM-GENIE is to our knowledge the most sophisticated climate model that has been applied to an ensemble of Eocene simulations, but we note that increasing computing power is now enabling ensembles of simulations with moderately higher resolution models, such as HadCM3 (3.75° × 2.5°) (e.g. Araya-Melo et al., 2015; Lord et al., 2017), to be run, although with some limitation in the model years in each simulation. It will never be possible to apply state of the art climate models to large ensembles because, given the continual striving for the highest possible resolution, single simulations with such models will always be at the limits of what is practicable with available computing power. EMICs therefore have an important role in furthering our understanding of past, present and future climate systems, and in the rapid identification of influencing factors and modes of response which may be targeted for study by slower but more powerful models.

Our study of the early Eocene climate and the PETM using PLASIM-GENIE has shown that variability in orbital parameters can exert significant climatic influence, particularly in regard to tropical temperature and precipitation, and they should not be ignored in modelling studies of climates of the past.

**Data Availability**

Details on access to the model code, and instructions on compiling the model are given in Holden et al. (2016).

# Appendix A Hypercube Generation

This study has been designed together with a future study using the EMIC model GENIE-1 (Edwards and Marsh, 2005). The GENIE-1 model will use all four of the forcing parameters and the dummy parameter, used in the present study, together with an additional six forcing parameters not used by the PLASIM-GENIE study. For PLASIM-GENIE we have run 50 simulations with five parameters, while in GENIE-1 we will run 100 simulations with 11 parameters, so that the number of runs in each ensemble is approximately 10 times the input dimension (Loeppky et al., 2012).

The overall design for both studies is based on a maximin Latin hypercube with 100 rows and 11 columns produced by repeatedly invoking the lhsdesign function in MATLAB (MathWorks), with the command:

```
hyperCube = lhsdesign(100, 11, 'criterion', 'maximin', 'iterations', 100);
```

to select from 100 iteratively generated hypercubes, the one which best fits the maximin criterion, i.e. where the minimum Euclidian distance between points in hyperspace is at a maximum. This MATLAB command is repeated until the absolute value of correlation between columns falls below a selected value, or until a selected number of attempts has been made. The ability of this 'brute force' approach to produce a hypercube which satisfies the maximin criterion, with the required low correlation between columns decreases rapidly with an increasing number of columns, and a decreasing target correlation, but in several minutes it can generate a hypercube with 100 rows, each representing a design point for an ensemble member, and 11 columns, each representing a forcing or dummy parameter, with correlation between any two parameters not exceeding 0.1.

We then modify the overall design by first picking a subset of 50 of the 100 design points to give good coverage of the PLASIM-GENIE subspace. We randomly select an initial point, and iteratively select from the remainder, without replacement, the point which provides the largest increase in the number of populated sectors across all the two-dimensional projections of PLASIM-GENIE parameter space defined by dividing each two-dimensional subspace into 6 x 6 equal sectors.

This defines a template comprising a 50-member subset of 11 parameter values.

Copying the template and discarding the six parameters which are only used in the GENIE-1 ensemble yields the final hypercube design for the PLASIM-GENIE ensemble, comprising 50 sets of five parameters.

A second copy of the template forms the top half of the GENIE-1 hypercube, and the bottom half is partially constructed by duplicating only the five PLASIM-GENIE parameters from the first 50 rows, with the remaining six parameters determined by choosing a previously unselected point, without replacement, from the initial 100 x 11 hypercube that maximises the Euclidean distance between the pair of points in the subspace of the remaining six parameters.

Following this procedure, the two hypercubes for the PLASIM-GENIE and GENIE-1 studies both show very good state-space coverage and low correlation, and each member of the PLASIM-GENIE ensemble has two corresponding members in

the GENIE-1 ensemble, with identical values for the parameters in common, but widely differing sets of values for the parameters only used by GENIE-1.

## Author Contribution

J. Keery and P. Holden designed and prepared the ensemble configurations and analysed the model outputs with advice from N. Edwards. J. Keery prepared the manuscript with contributions from both co-authors.

## Competing Interests

The authors declare that they have no conflict of interest.

## Acknowledgements

The authors gratefully acknowledge support from NERC, with funding for project NE/K006223/1. We are very grateful to the reviewers M. Crucifix and D. De Vleeschouwer, and to the editor A. Winguth, for their thorough and constructive comments which have helped to improve the manuscript.

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

**Table 1**   Uniform ranges for forcing and dummy parameters

|  | min | max |
| --- | --- | --- |
| pCO$_2$ (ppm) | 280 | 3000 |
| Precession ( ° ) | 0 | 360 |
| Obliquity ( ° ) | 22.0 | 24.5 |
| Eccentricity ( - ) | 0.00 | 0.06 |
| Dummy ( - ) | 0 | 1 |

**Table 2**   Forcing factors and dummy values for each member in the ensemble.  Precession = $\omega$, the angle between the moving vernal equinox and the longitude of perihelion.

| Member (-) | CO$_2$ (ppm) | Eccentricity (-) | Precession (°) | Obliquity (°) | Dummy (-) |
| --- | --- | --- | --- | --- | --- |
| 1 | 975.6 | 0.0022 | 142.5 | 22.37 | 0.822 |
| 2 | 2418.7 | 0.0256 | 165.2 | 23.95 | 0.907 |
| 3 | 1259.4 | 0.0007 | 307.1 | 23.91 | 0.323 |
| 4 | 801.3 | 0.0163 | 270.4 | 23.50 | 0.276 |
| 5 | 1720.1 | 0.0559 | 206.7 | 23.82 | 0.402 |
| 6 | 327.1 | 0.0595 | 135.9 | 23.53 | 0.681 |

| | | | | |
|---|---|---|---|---|
| 7 | 2937.7 | 0.0418 | 287.1 | 22.53 | 0.650 |
| 8 | 1200.3 | 0.0237 | 313.2 | 24.12 | 0.978 |
| 9 | 1420.7 | 0.0158 | 297.1 | 23.86 | 0.931 |
| 10 | 2157.6 | 0.0432 | 100.6 | 23.74 | 0.661 |
| 11 | 1791.7 | 0.0241 | 247.2 | 23.43 | 0.429 |
| 12 | 2369.0 | 0.0425 | 78.9 | 22.65 | 0.167 |
| 13 | 2502.9 | 0.0296 | 0.5 | 22.69 | 0.122 |
| 14 | 2149.2 | 0.0405 | 249.9 | 24.23 | 0.347 |
| 15 | 1061.7 | 0.0394 | 40.9 | 23.94 | 0.189 |
| 16 | 711.3 | 0.0199 | 274.6 | 22.08 | 0.913 |
| 17 | 1817.1 | 0.0578 | 291.4 | 23.08 | 0.888 |
| 18 | 722.1 | 0.0463 | 195.8 | 24.38 | 0.865 |
| 19 | 2988.5 | 0.0039 | 110.1 | 24.40 | 0.049 |
| 20 | 539.4 | 0.0251 | 212.5 | 23.29 | 0.234 |
| 21 | 450.6 | 0.0335 | 96.1 | 22.28 | 0.674 |
| 22 | 2700.1 | 0.0049 | 165.9 | 23.66 | 0.630 |
| 23 | 2025.4 | 0.0320 | 189.4 | 23.63 | 0.087 |
| 24 | 2268.7 | 0.0308 | 233.3 | 22.86 | 0.461 |
| 25 | 1447.2 | 0.0364 | 62.0 | 23.40 | 0.541 |
| 26 | 1168.3 | 0.0300 | 147.4 | 22.97 | 0.947 |
| 27 | 1317.6 | 0.0377 | 12.4 | 23.04 | 0.714 |
| 28 | 1639.5 | 0.0265 | 150.9 | 22.98 | 0.524 |
| 29 | 399.0 | 0.0589 | 262.7 | 23.46 | 0.028 |
| 30 | 2876.3 | 0.0411 | 203.0 | 22.05 | 0.608 |
| 31 | 2611.1 | 0.0170 | 54.3 | 22.84 | 0.746 |
| 32 | 2831.7 | 0.0564 | 187.2 | 23.72 | 0.696 |
| 33 | 1998.5 | 0.0372 | 278.8 | 24.19 | 0.805 |
| 34 | 1465.0 | 0.0439 | 38.9 | 23.50 | 0.376 |
| 35 | 1660.0 | 0.0109 | 85.3 | 22.88 | 0.896 |
| 36 | 2393.7 | 0.0587 | 127.9 | 24.27 | 0.191 |
| 37 | 286.3 | 0.0004 | 27.1 | 23.99 | 0.391 |
| 38 | 667.4 | 0.0509 | 116.5 | 22.71 | 0.569 |
| 39 | 2246.8 | 0.0450 | 317.4 | 22.90 | 0.103 |
| 40 | 2334.2 | 0.0096 | 294.7 | 23.61 | 0.532 |
| 41 | 2968.2 | 0.0346 | 329.8 | 22.51 | 0.314 |
| 42 | 768.2 | 0.0085 | 218.3 | 23.00 | 0.000 |
| 43 | 925.8 | 0.0450 | 327.2 | 24.32 | 0.753 |
| 44 | 384.5 | 0.0081 | 60.6 | 22.59 | 0.436 |
| 45 | 850.7 | 0.0551 | 322.9 | 23.21 | 0.459 |
| 46 | 1112.8 | 0.0150 | 356.7 | 23.27 | 0.579 |
| 47 | 1255.8 | 0.0116 | 212.2 | 22.31 | 0.487 |
| 48 | 1124.1 | 0.0530 | 343.7 | 22.40 | 0.065 |
| 49 | 2113.9 | 0.0276 | 9.9 | 22.19 | 0.856 |
| 50 | 1681.0 | 0.0354 | 175.5 | 22.45 | 0.287 |

1 **Table 3**                **$R^2$ correlation between PC scores from SVD and PC scores emulated with the linear models.**

| | PC1 | PC2 | PC3 |
|---|---|---|---|
| DJF_temperature | 0.95 | 0.58 | 0.75 |
| JJA_temperature | 0.97 | 0.97 | 0.72 |
| DJF_precipitation | 0.97 | 0.92 | 0.64 |
| JJA_precipitation | 0.99 | 0.99 | 0.89 |

4 **Table 4**                **Total effects of forcing parameters on simple scalar metrics.**

| | $CO_2$ | Eccentricity | Obliquity | Precession |
|---|---|---|---|---|
| MAT | 0.993 | 0.002 | 0.000 | 0.005 |
| N. seasonality | 0.766 | 0.003 | 0.011 | 0.220 |
| N. winter TPTD | 0.939 | 0.006 | 0.039 | 0.017 |
| N. summer TPTD | 0.144 | 0.000 | 0.673 | 0.183 |
| S. winter POLC | 0.979 | 0.004 | 0.005 | 0.012 |
| N. winter POLC | 0.088 | 0.000 | 0.789 | 0.122 |
| Asian monsoon index | 0.094 | 0.004 | 0.063 | 0.840 |
| African monsoon index | 0.017 | 0.001 | 0.001 | 0.981 |
| American monsoon index | 0.490 | 0.004 | 0.020 | 0.486 |

1    **Table 5**               R correlation values for PC scores for temperature and precipitation in DJF and JJA. Values where $R^2 \geq 0.5$
2    are shown in red.

| | | DJF_precipitation | | |
|---|---|---|---|---|
| | | PC1 | PC2 | PC3 |
| | PC1 | **0.993** | -0.004 | -0.080 |
| DJF_temperature | PC2 | -0.067 | -0.364 | **-0.864** |
| | PC3 | 0.005 | **0.783** | -0.354 |

| | | JJA_precipitation | | |
|---|---|---|---|---|
| | | PC1 | PC2 | PC3 |
| | PC1 | **0.976** | 0.091 | 0.157 |
| JJA_temperature | PC2 | 0.098 | **-0.947** | 0.082 |
| | PC3 | -0.180 | -0.049 | **0.795** |

5    **Table 6**               R correlation values for forcing factors and PC scores. Values where $R^2 \geq 0.5$ are shown in red.

| | | $CO_2$ | precession index | obliquity |
|---|---|---|---|---|
| | PC1 | **-0.859** | -0.018 | -0.057 |
| DJF_temperature | PC2 | 0.381 | -0.087 | -0.354 |
| | PC3 | 0.038 | **-0.924** | 0.311 |
| | PC1 | **-0.899** | 0.178 | -0.066 |
| JJA_temperature | PC2 | -0.018 | **-0.875** | 0.362 |
| | PC3 | 0.342 | 0.056 | -0.239 |
| | PC1 | **-0.867** | 0.003 | -0.025 |
| DJF_precipitation | PC2 | -0.198 | **-0.82** | 0.044 |
| | PC3 | -0.278 | 0.465 | 0.164 |
| | PC1 | **-0.953** | 0.065 | 0.008 |
| JJA_precipitation | PC2 | -0.07 | **0.96** | -0.131 |
| | PC3 | 0.219 | 0.191 | -0.029 |

1 **Table 7**     **Linear models derived from normalised forcing functions and monsoon indices**

| Terms | Asia | Africa | America |
|---|---|---|---|
| intercept | -0.096 | 0.200 | -0.273 |
| $CO_2$ | 0.187 | -0.089 | -0.422 |
| $\varepsilon$ | 0.189 | 0.027 | 0.065 |
| $e$ | 0.049 | -0.091 | -0.070 |
| $\sin(\omega)$ | -0.577 | 0.510 | 0.309 |
| $\cos(\omega)$ | -0.114 | -0.064 | -0.105 |
| $CO_2^2$ | - | 0.150 | 0.278 |
| $e^2$ | - | -0.115 | - |
| $e \times \sin(\omega)$ | -0.468 | 0.501 | 0.240 |
| $CO_2 \times \sin(\omega)$ | -0.214 | 0.215 | -0.085 |
| $\varepsilon \times \sin(\omega)$ | - | -0.069 | -0.071 |
| $e \times \cos(\omega)$ | -0.100 | - | - |
| $\sin(\omega) \times \cos(\omega)$ | 0.118 | - | - |
| $\varepsilon \times \cos(\omega)$ | - | -0.121 | - |
| $CO_2 \times \varepsilon$ | 0.121 | - | - |
| $CO_2 \times \cos(\omega)$ | - | 0.098 | - |
| $CO_2 \times e$ | - | 0.096 | - |

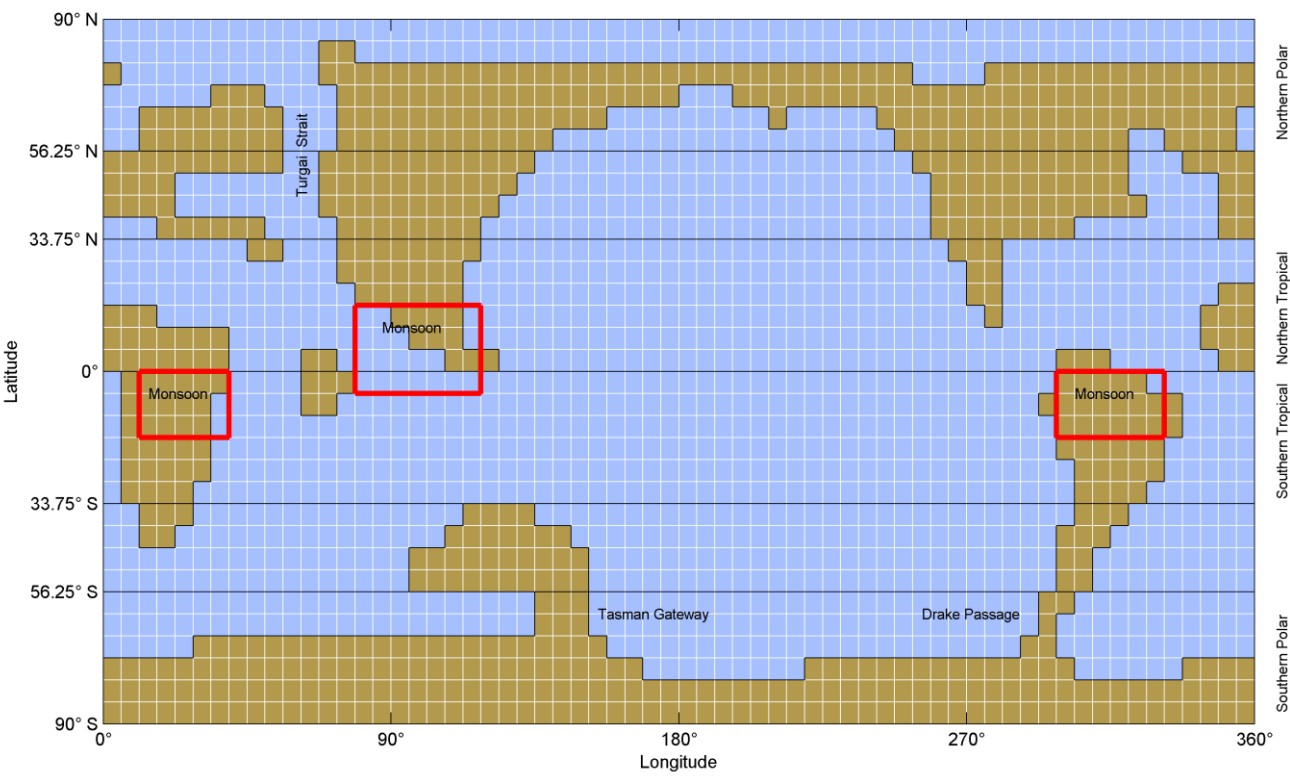

**Figure 1: Eocene palaeogeography and geographic areas used to determine simple metric values**

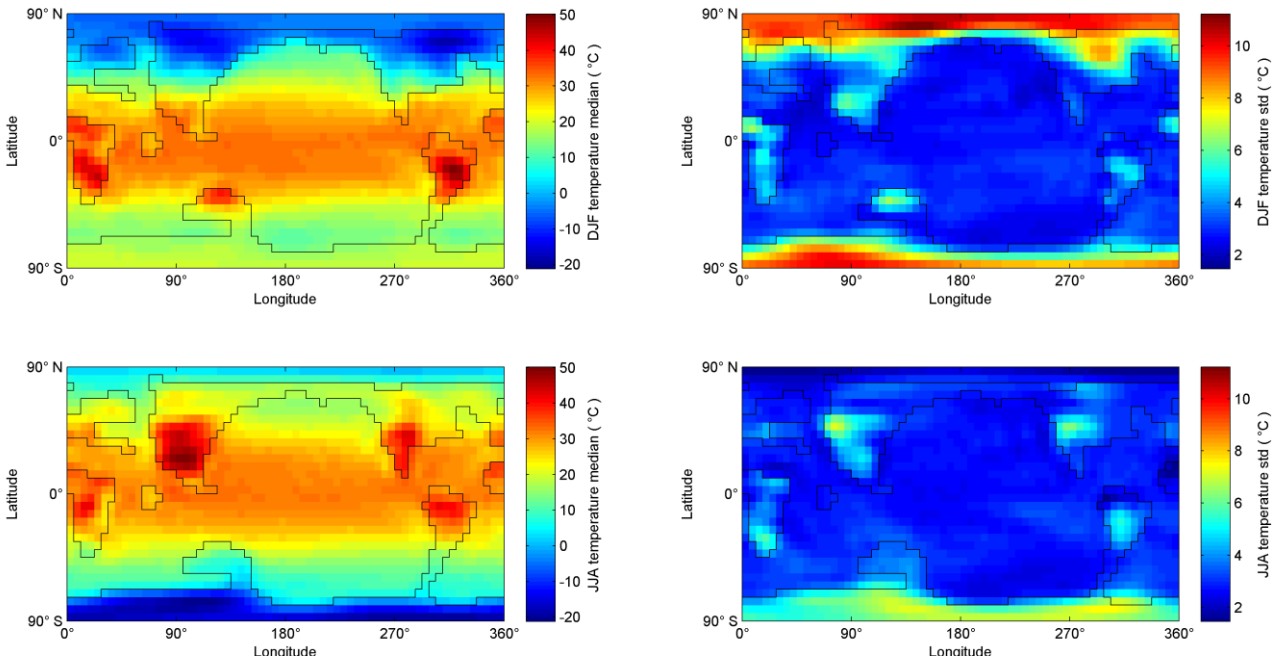

Figure 2: Ensemble temperature medians (left column) and standard deviations (right column) in DJF (top row) and JJA (bottom row).

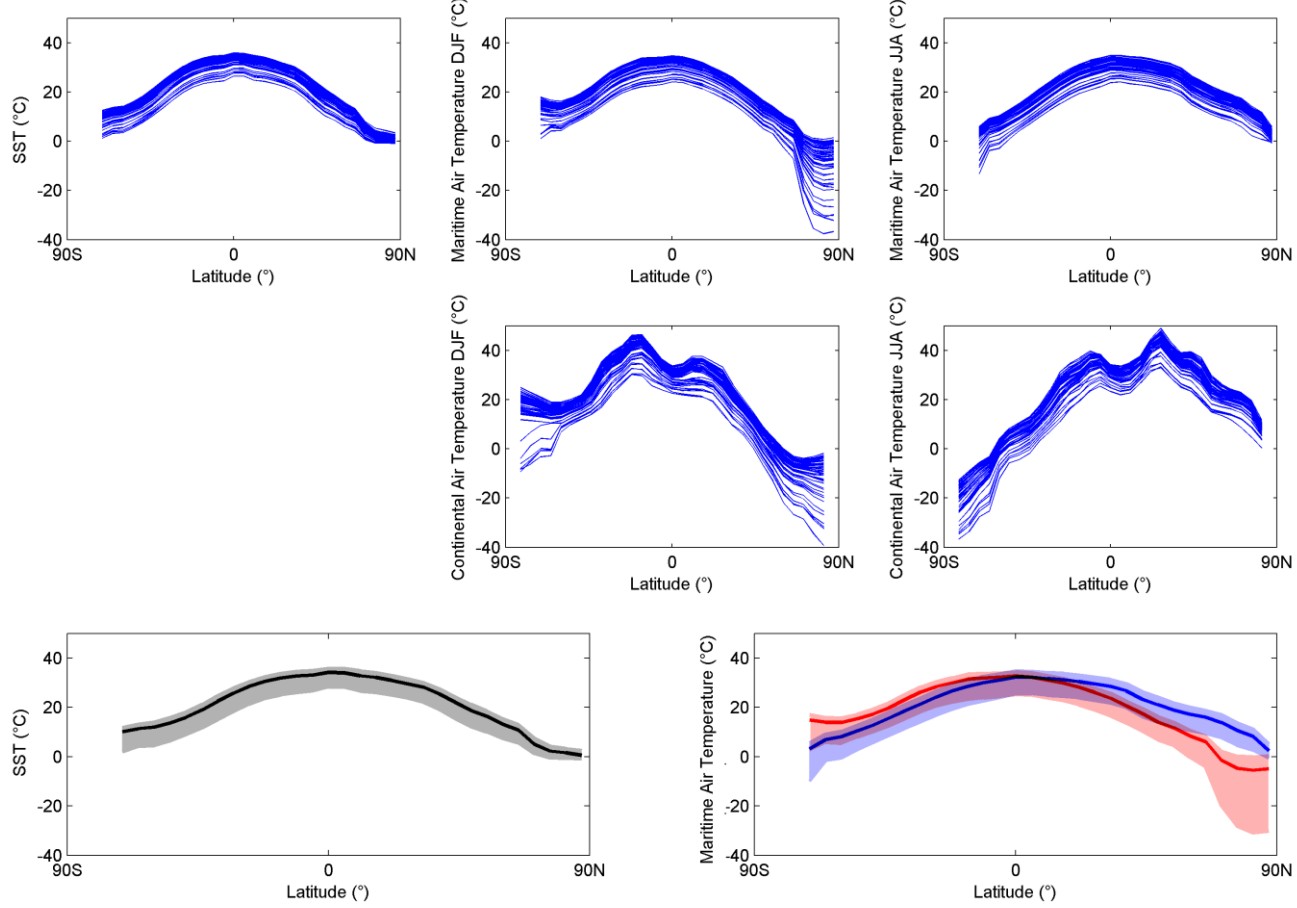

Figure 3: Top: full ensemble distributions of mean latitude values of global annual mean sea surface temperature (SST), with mean latitude maritime surface air temperature in DJF and JJA.

Middle: mean latitude continental surface air temperature in DJF and JJA.

Bottom: ensemble medians and 5% and 95% percentiles of global annual mean SST, and maritime surface air temperature in DJF (red) and JJA (blue).

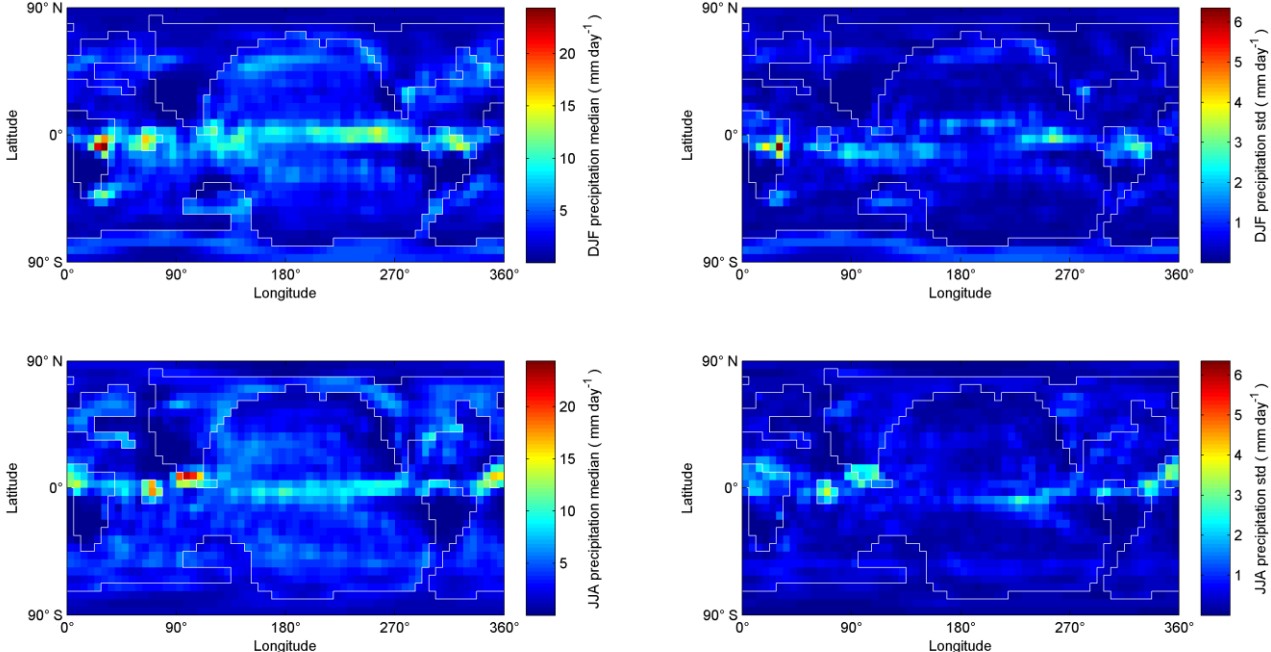

Figure 4: Ensemble precipitation medians (left column) and standard deviations (right column) in DJF (top row) and JJA (bottom row).

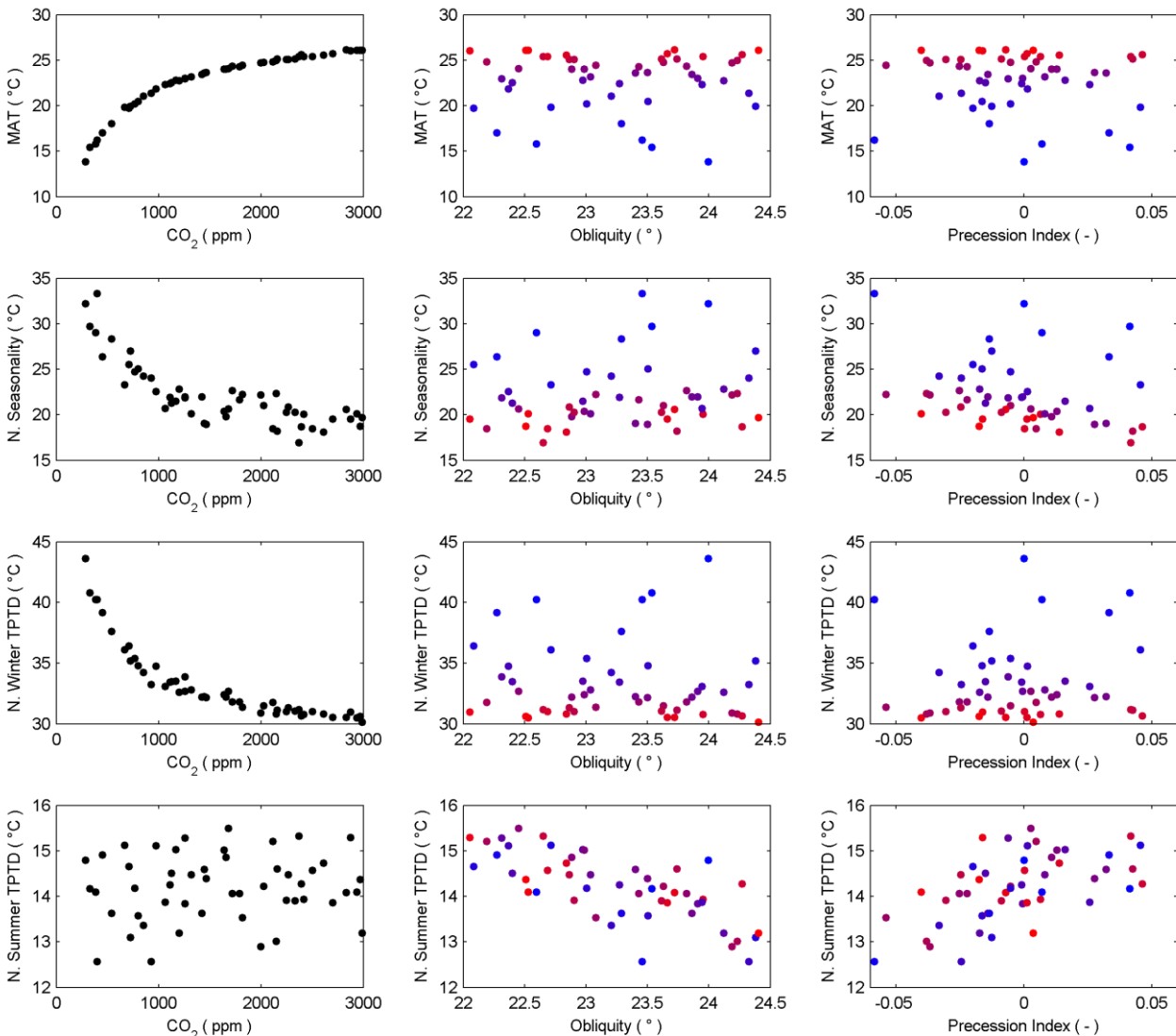

Figure 5: Correlation between three forcing factors $CO_2$, obliquity and precession index (in columns from left to right), and the simple metrics MAT, northern seasonality, northern winter tropical-polar temperature difference and northern summer tropical-polar temperature difference (in rows from top to bottom). $CO_2$ is plotted in colour in the obliquity and precession plots (blue = low, red = high)

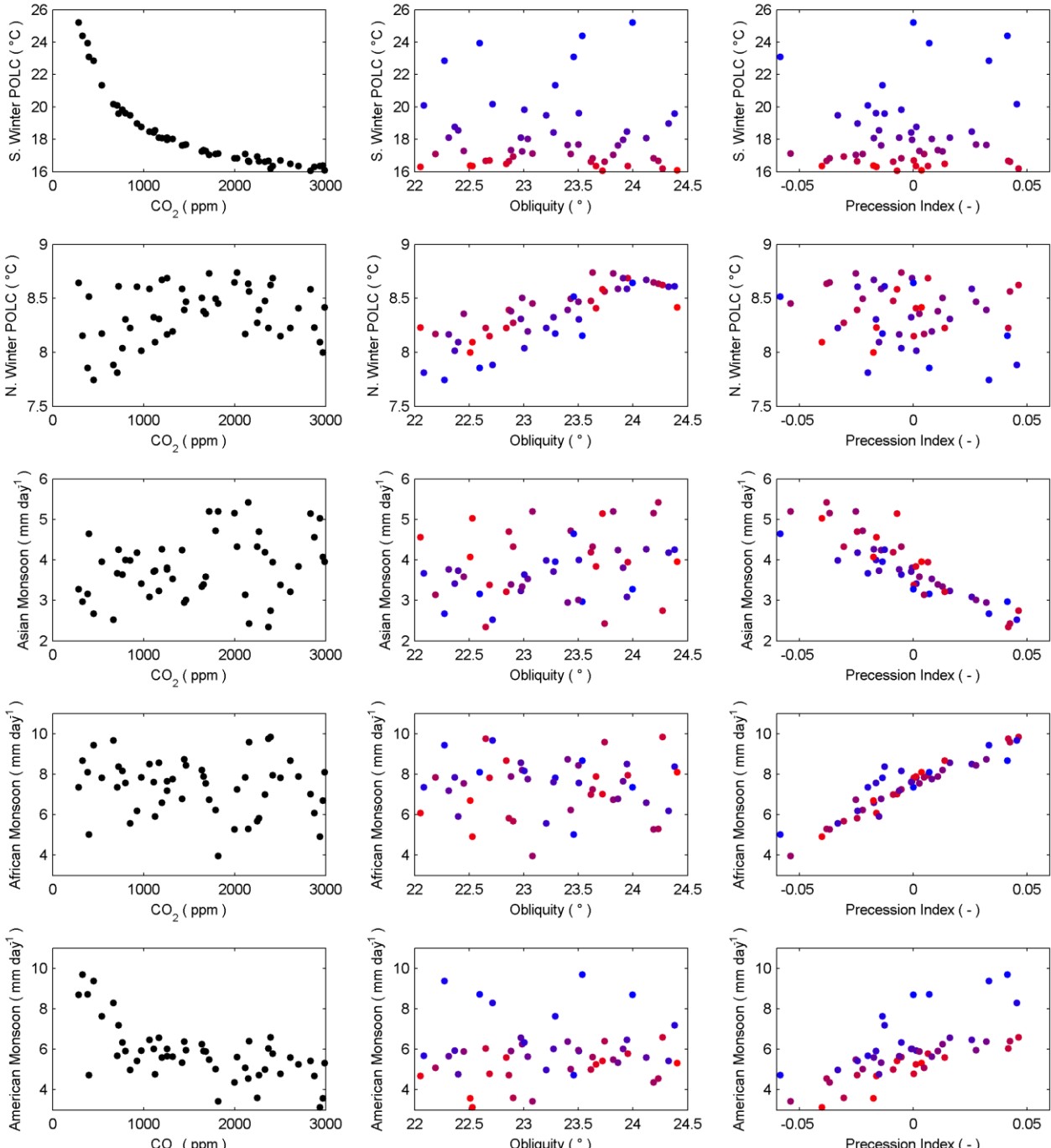

**Figure 6: Correlation between three forcing factors $CO_2$, obliquity and precession index (in columns from left to right), and the simple metrics southern winter polar OLC, northern winter polar OLC, Asian monsoon index African monsoon index and American monsoon index (in rows from top to bottom). $CO_2$ is plotted in colour in the obliquity and precession plots (blue = low, red = high)**

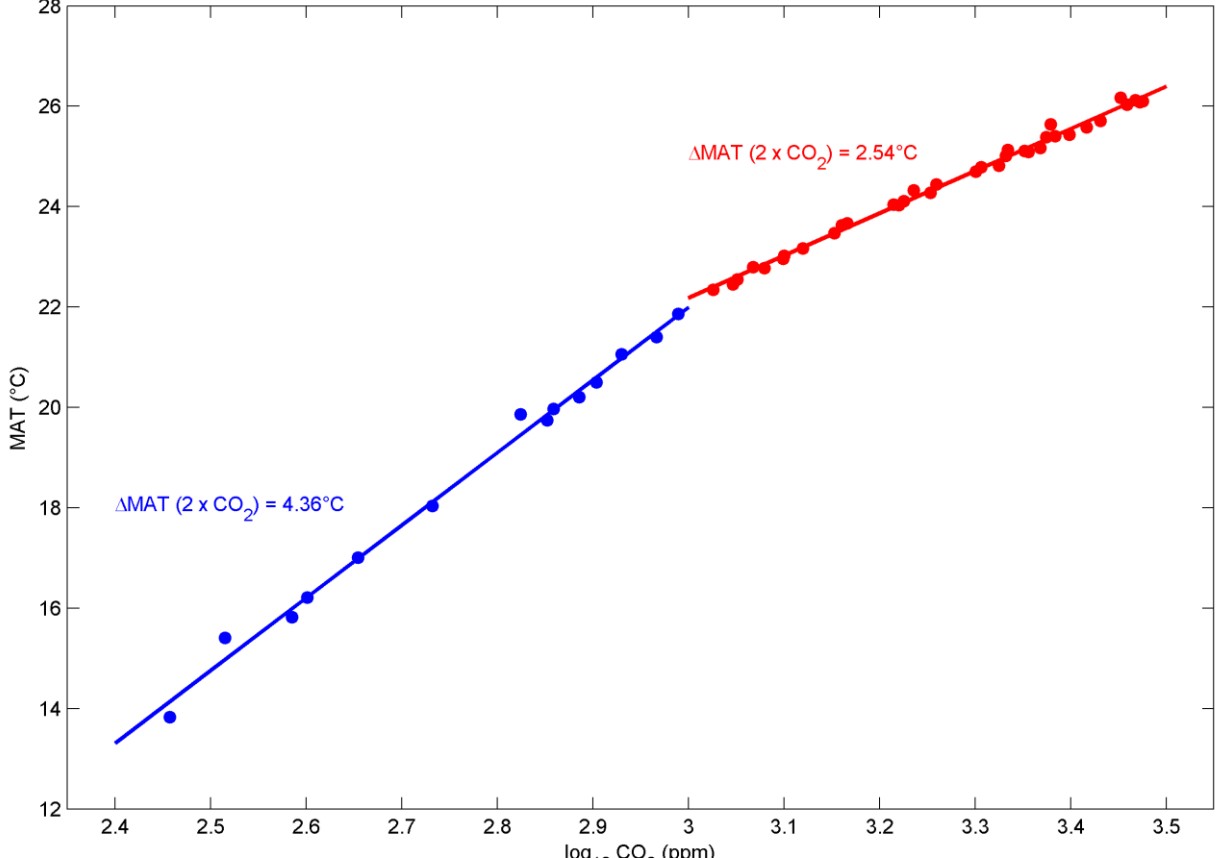

Figure 7: Mean air temperature plotted against $CO_2$ on a logarithmic scale, with regression lines plotted for $CO_2 < 1000$ ppm (blue), and $CO_2 > 1000$ ppm (red), with climate sensitivities for a doubling of $CO_2$ from both of the regressions.

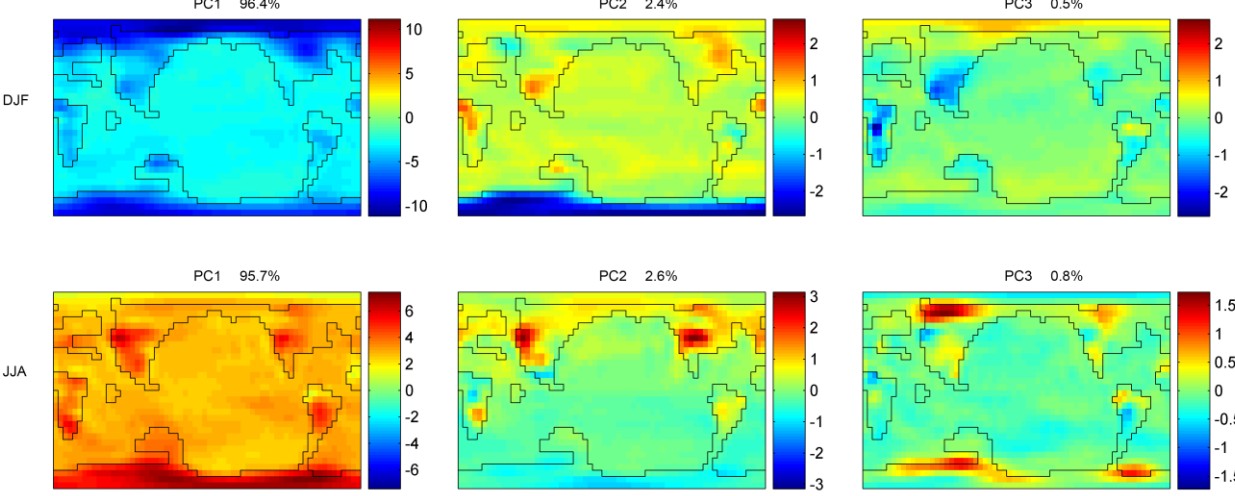

3 **Figure 8: The first three principal components of DJF_temperature (top row) and JJA_temperature (bottom row). Percentages of**
4 **variance explained by each principal component are shown above each plot.**

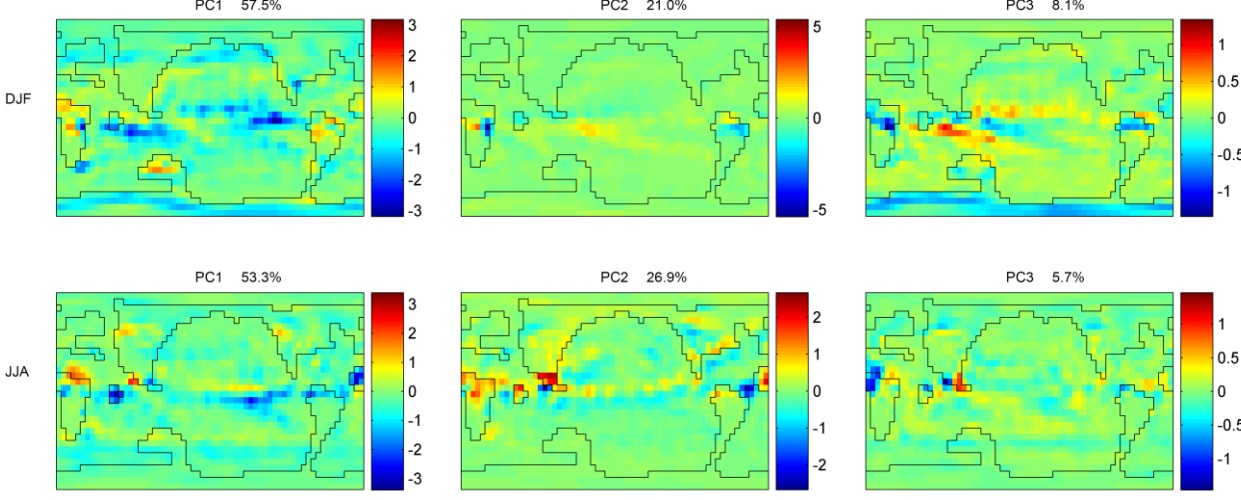

8 **Figure 9: The first three principal components of DJF_precipitation (top row) and JJA_precipitation (bottom row). Percentages**
9 **of variance explained by each principal component are shown above each plot.**

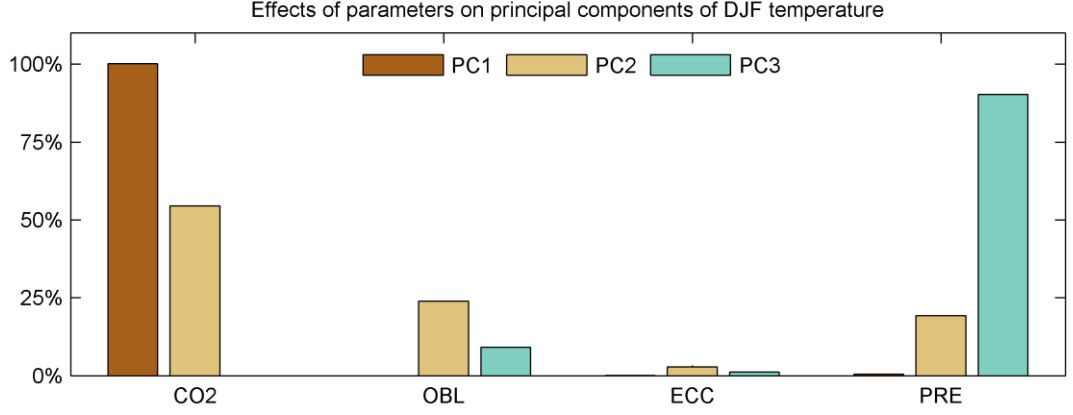

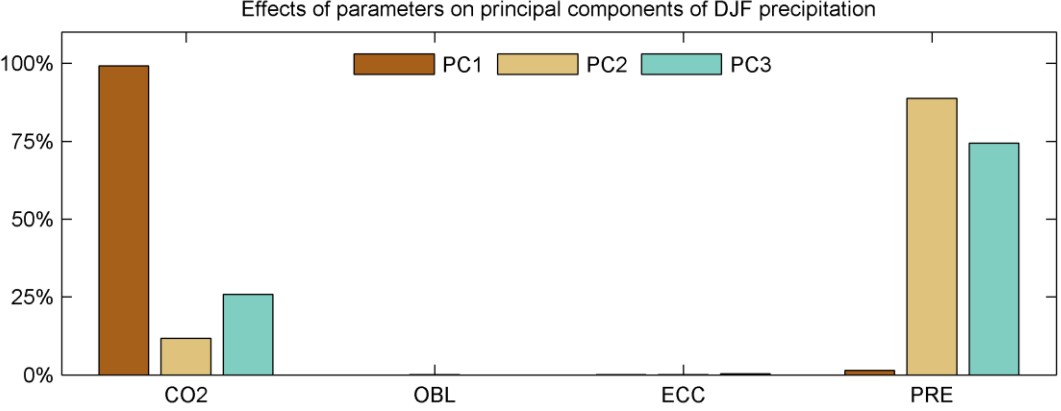

**Figure 10: Main effects of forcing parameters on the first three principal components of DJF_temperature (top row) and DJF_precipitation (bottom row).**

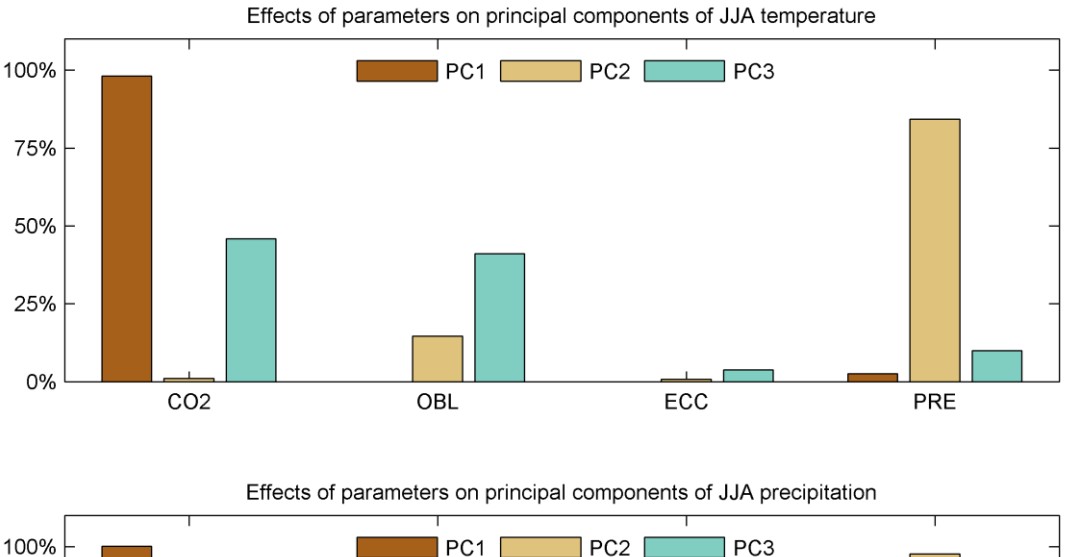

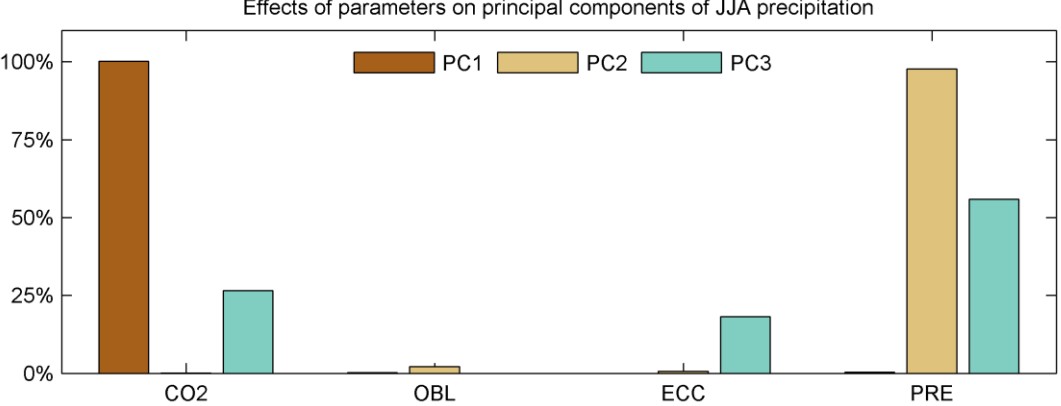

Figure 11: Main effects of forcing parameters on the first three principal components of JJA_temperature (top row) and JJA_precipitation (bottom row).

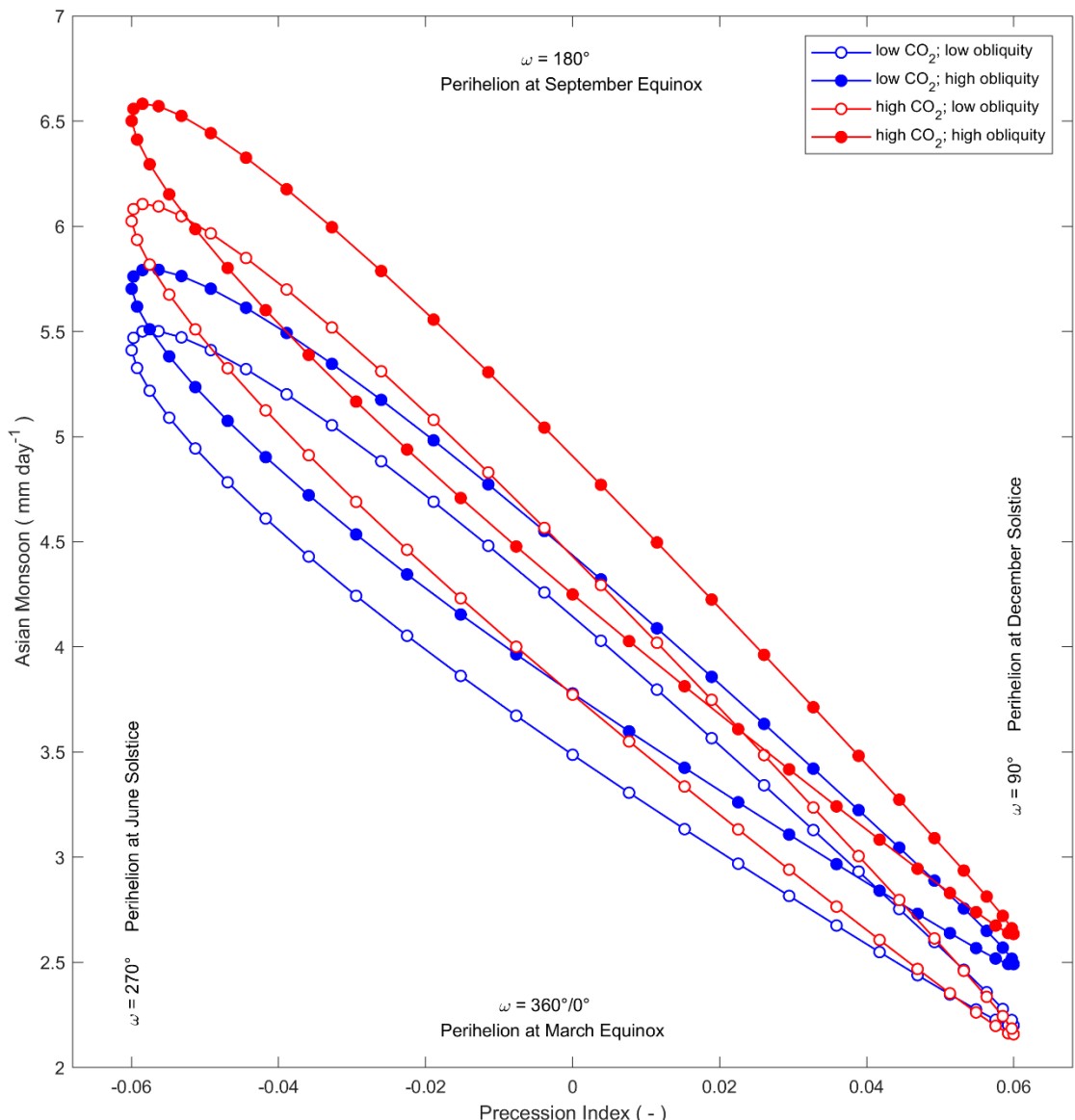

2 **Figure 12: Emulated values of the Asian monsoon index, for the full range of the precession index ($e\sin\varpi$), at low and high values**

3 **of $CO_2$ and obliquity ($\varepsilon$).**

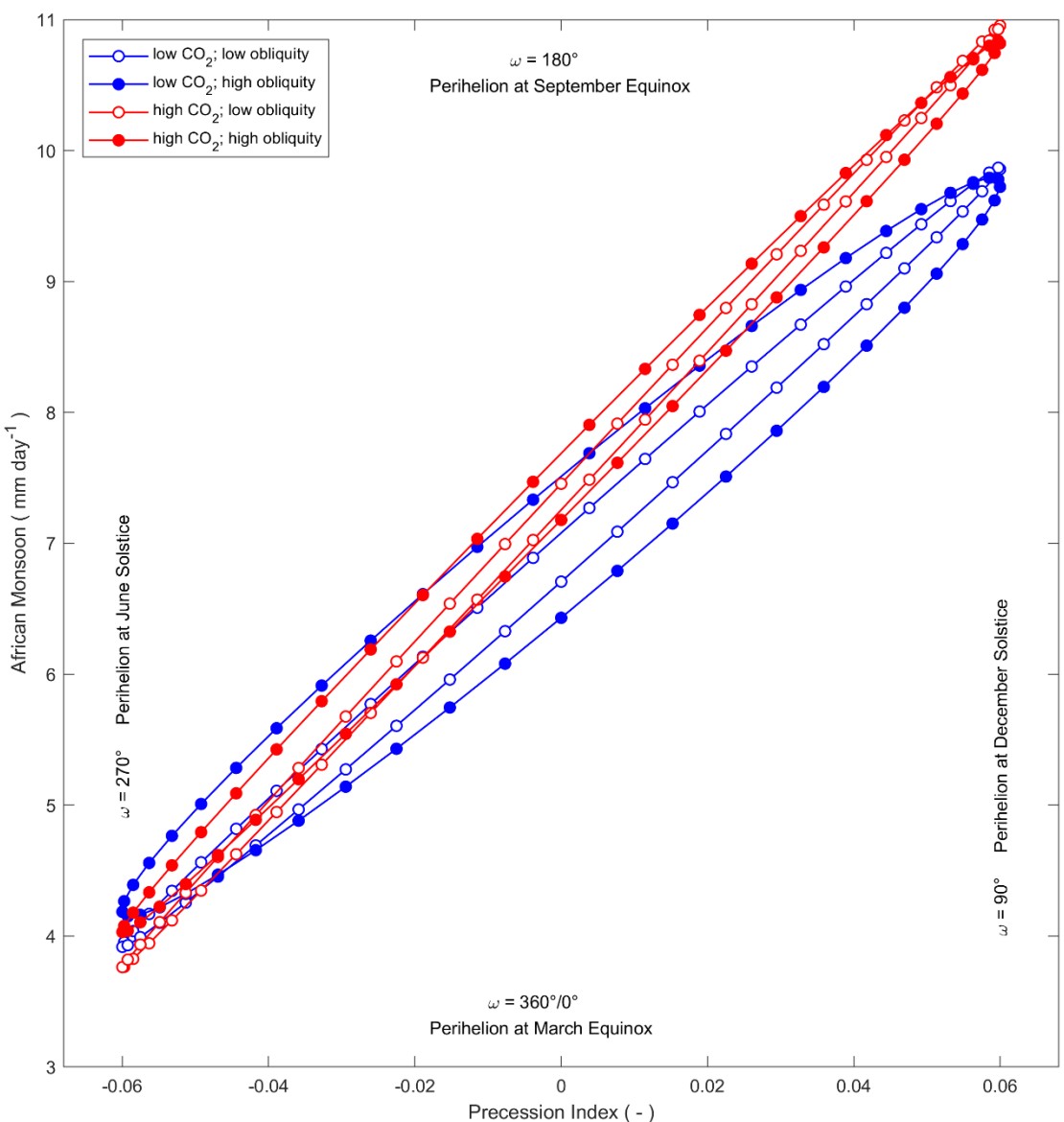

**Figure 13: Emulated values of the African monsoon index, for the full range of the precession index ($e\sin\omega$), at low and high values**
**of $CO_2$ and obliquity ($\varepsilon$).**

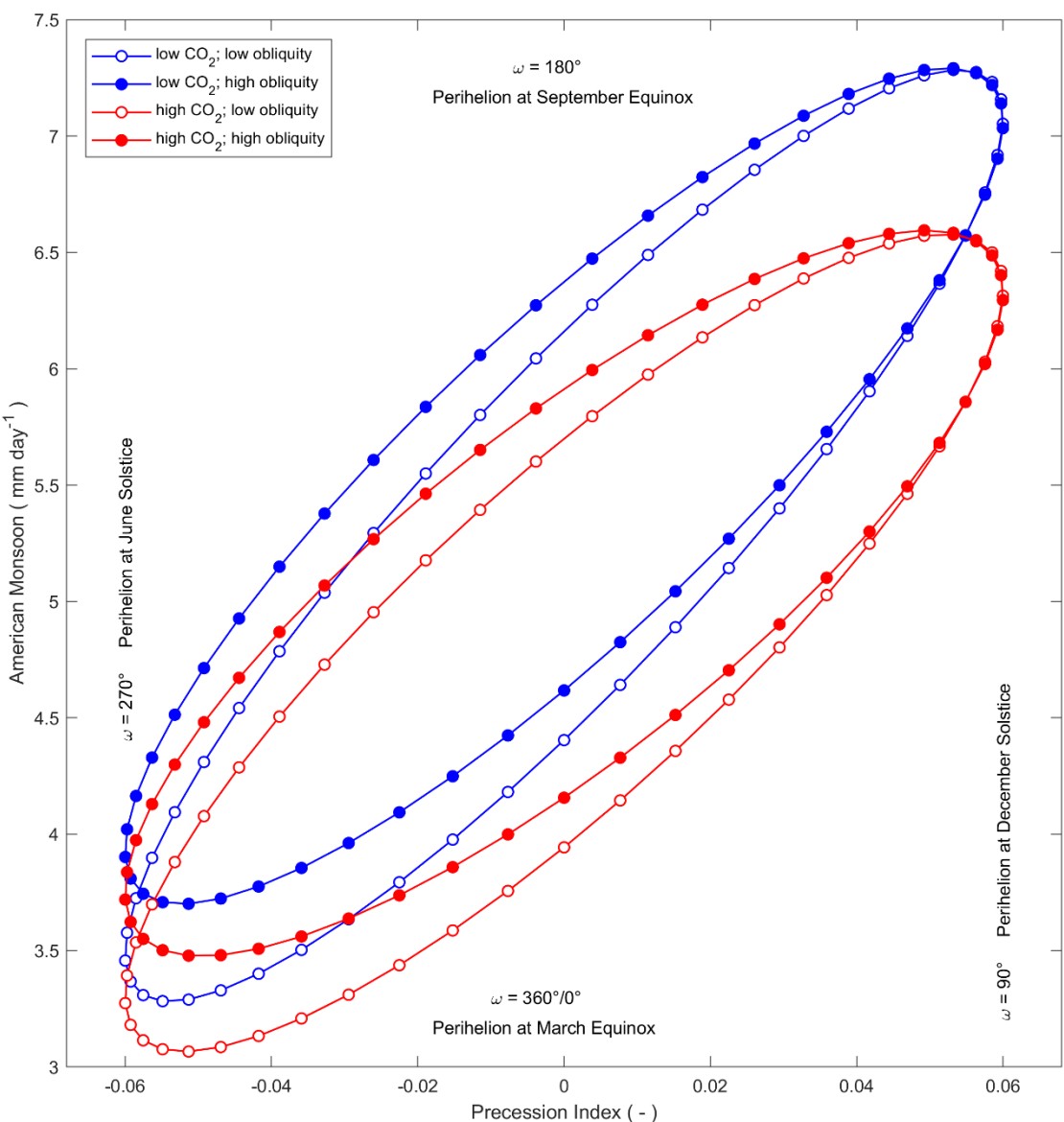

Figure 14: Emulated values of the Amerian monsoon index, for the full range of the precession index (*e*sin$\varpi$), at low and high values of $CO_2$ and obliquity ($\varepsilon$).