# Peer review of "Sensitivity of the Eocene Climate to CO2 and Orbital Variability"

_Climate of the Past, 2017_

## Referee Comment (RC1) · M. Crucifix (Referee) · 7 Jun 2017

**1  Summary**

Keery et al. present a sensitivity analysis of the Eocene climate to four factors: CO$_2$ concentration, eccentricity, obliquity, and precession angle. They use, to this end, the PLASIM-GENIE model (details in their section 3) with suitable palaeogeography. The methodology relies on a 50-member hyper-cube sample of a 5-d space (one extra dummy variable was added), and linear modelling with a Information Criteria for model selection. Experiment output are summarised using fit-for-purpose summaries like "tropical-polar temperature difference" and monsoon indices, as well as principal components obtained from a singular value decomposition. The authors conclude on

the importance of $CO_2$ for global mean temperature, and of the orbital elements for the spatial distribution and regional weather systems such as monsoons.

**2   Main comments**

1. The paper is in the line of a number of recent studies attempting to estimate the relative sensitivity of the climate system to $CO_2$ and orbital forcing, using a methodology founded on ensemble of experiments. This includes, in addition to the Holden et al. (2015) and Bounceur et al. (2015) cited, Araya-Melo et al. (2015) and Lord et al. (2017)[1]. Keery et al. is the only article to focus on the Eocene, which makes it an original contribution. It also uses a much simpler methodology than Araya-Melo et al. (2015), Bounceur et al. (2015), and Lord et al. (2017) because it uses linear regression instead of a Gaussian process emulator. In fact, the authors reference to the word "emulator" is slightly unusual because emulation is, in the climate literature, often used to designate statistical meta-modelling with a focus on uncertainty quantification. Claiming (p. 8) that a "similar emulator approach has been applied by Bounceur et al. 2015" is therefore somewhat misleading. Bounceur et al. and Araya-Melo et al. applied the developments of Oakley and O'Hagan (2004) with, in the case of Bounceur, the additional complication of the PCA emulator. In passing, Araya-Melo et Lord used HadCM3 which shows that ensemble-based sensitivity analysis to orbital forcing is doable with GCMs (this qualifies the author's comment on line 15, p.2). Of course, the fact that other authors have adopted a more sophisticated methodology invalidates by no means the approach used by Keery et al.: there may be no need to use a sledgehammer to crack a nut. It remains that the methodological set up used here is a step backwards compared to recent studies, and this arguably
* * *
[1]the later was submitted after Keery et al. and could not of course be cited by these authors.

requires some justification. How much do we lose with the linearity assumption, and which impact does it have on the uncertainties of the quantification of main effects? (see comment 3. more specifically on main effects).

2. Experiment design. The authors do not say much about the ensemble design, except that this is a latin hypercube. There are many ways to do a latin hypercube, and it usually involves additional constraints. In fact this experiment design raises some doubts. For example, why are some secondary structures (periodic up and downs) apparent in the response to obliquity, Figure 5, middle column? Is this just a subjective visual impression? One potentially problematic element is the definition of the sampled astronomical space. It seems that latin hypercube sampling is made on axes along $e$, $\varpi$ (longitude of perihelion) and $\varepsilon$. If this is what the authors have been doing then this is non-physical. We know that the astronomical forcing generates effects through seasonal and daily insolation, which are very well approximated by linear functions of $e \sin \varpi$ (which the authors call the precession index on Fig. 6) and $e \cos \varpi$. This is the reason why several authors have chosen to sample the astronomical space following the axes $e \sin \varpi$ and $e \cos \varpi$ and regress against these components. Presumably the regression analysis by Keery is indeed done against these indices but the text is not always clear. Lines 1-2 p. 8. rather suggest that the explanatory variables where $\sin \varpi$ and $\cos \varpi$ (instead of their multiplication by $e$) and the lines 4-5 p. 11 are quite confusing. Hopefully the choice of regression variables is mainly matter of text clarification, but the design of the latin hypercube may have a more fundamental problem.

3. There may be some confusion about the meaning of the *main effects*. Saltelli does not use the phrase "first order" to mean linear approximation. In a case where only one factor would matter (be the relationship linear on not), the main and total effects would match (Saltelli et al. (2004), ch. 1 states clearly the definitions; or refer again to Oakley and O'Hagan (2004)). More generally, computing

main and total effects is not trivial and always involves some approximations. More details on their computation would be welcome.

4. Singular value decomposition is a great dimensionality reduction methodology, but how much is learned by analysing the behaviour of principal components separately is a more contentious subject. Identification of principal components can be fragile to some implementation details, such as, e.g. grid area weighting and experiment design, and the physical phenomena which give rise to climate variability need not be orthogonal. In fact physical modes may project poorly on the orthogonal vectors (Monahan and Fyfe, 2006). These caveats implicitly acknowledged by the authors (p. 11, ll. 20-21) but this state-of-affairs poses some questions about the emphasis on principal components in this article.

**3  Minor (scientific) comments**

- How Fig. 2 should be interpreted is not entirely clear since the ensemble was not explicitly designed so that the ensemble mean is an estimate of the Eocene climate mean.

**4  Minor (editorial) comments**

- Introduce subtitle after section 2.

- Material about cyclostratigraphy under section 2.1.2. may possibly be considered for shortening as slightly out of scope of the article. This said this is an interesting read.

- PLASIM-GENIE does not need a specific section: it can fall under section 3. Methods.

- p. 6 reference Gough (1981) is mistakenly repeated.

- p. 7, the sentence "We apply the linear algebraic tool SVD" sounds unnecessarily sophisticated. Why not "We perform a singular value decomposition to identify principal components"

- p. 10, l. 27 : define the word "precession" precisely.

- p. 12, ll. 13-17 : introducing new results so close to the closing words is usually not encouraged.

**5 Digital material**

- Relevant data of the Eocene runs (at least the summaries and experiment input data) could be provided.

**References**

P. A. Araya-Melo, M. Crucifix, and N. Bounceur. Global sensitivity analysis of the indian monsoon during the pleistocene. *Climate of the Past*, 11:45–61, 2015. doi: 10.5194/cp-11-45-2015. URL http://www.clim-past.net/11/45/2015/.

N. Bounceur, M. Crucifix, and R. D. Wilkinson. Global sensitivity analysis of the climate-vegetation system to astronomical forcing: an emulator-based approach. *Earth System Dynamics*, 6:205–224, 2015. doi: 10.5194/esd-6-205-2015. URL http://www.earth-syst-dynam.net/6/205/2015/.

P. B. Holden, N. R. Edwards, P. H. Garthwaite, and R. D. Wilkinson. Emulation and interpretation of high-dimensional climate model outputs. *Journal of Applied Statistics*, pages 1–18, 2015. doi: 10.1080/02664763.2015.1016412. URL http://dx.doi.org/10.1080/02664763.2015.1016412.

N. S. Lord, M. Crucifix, D. J. Lunt, M. C. Thorne, N. Bounceur, H. Dowsett, C. L. O'Brien, and A. Ridgwell. Emulation of long-term changes in global climate: Application to the late pliocene and future. *Climate of the Past Discussions*, page 1–47, 2017. ISSN 1814-9359. doi: 10.5194/cp-2017-57. URL http://dx.doi.org/10.5194/cp-2017-57.

A. H. Monahan and J. C. Fyfe. On the nature of zonal jet eofs. *Journal of Climate*, 19: 6409–6424, 2006. ISSN 1520-0442. doi: 10.1175/jcli3960.1. URL http://dx.doi.org/10.1175/JCLI3960.1.

J. E. Oakley and A. O'Hagan. Probabilistic sensitivity analysis of complex models: a bayesian approach. *Journal of the Royal Statistical Society: Series B (Statistical Methodology)*, 66: 751–769, 2004. doi: 10.1111/j.1467-9868.2004.05304.x.

A. Saltelli, S. Tarentola, F. Campolongo and M. Ratto, Sensitivity Analysis in Practice: A guide to assessing scientific methods *John Wiley and Sons Ltd.*, 219pp., 2004.

---

## Referee Comment (RC2) · D. De Vleeschouwer (Referee) · 8 Jun 2017

This paper reports on an ensemble of 50 Eocene climate-model simulations, each of which characterized by a different combination of eccentricity, obliquity, precession and atmospheric CO2 concentration. The climate model is the PLASIM-GENIE model, a new model of intermediate complexity, recently introduced by Holden et al. (2016). The study aims to summarize the ensemble of paleoclimate simulations by looking at what-they-call *"simple metrics"*, principal component analysis and an emulator approach.

This study provides a couple of interesting results. The first is the existence of a sea-ice-related threshold mechanism in the northern hemispheric high latitudes. From Figure 2 and 3, it seems that when a certain threshold in the extent of DJF-sea-ice is

exceeded, temperatures (both sea-surface and maritime air temperatures) drop significantly. It would be interesting to read the author's opinion how this compares to the recent findings of modeling work by Zeebe et al. (2017), who found that *"High-latitude mechanisms are unlikely drivers of orbitally paced changes in the late Paleocene-early Eocene"*. The interesting role of (seasonal) sea-ice in the climate system of the early Eocene aspect remains, however, rather underdeveloped in the present version of the paper. The second interesting aspect is the distinct response to precession of monsoonal precipitation and temperature in the different monsoonal systems (e.g. Figure 6). The description and discussion of these Eocene paleoclimate simulations is useful and perfectly fits the scope of the journal. The current version of the manuscript is, however, unsatisfactory for publication in Climate of the Past for the reasons listed below.

**Major Comments**

1. One of the major conclusions in the current version of the manuscript, is that 95The emulator approach adopted in this study allows for estimating the response of different aspects of the climate system (e.g. wet-season monsoonal precipitation) over the full input space. It would -for example- be interesting to see the response of precipitation and temperatures in the different monsoonal systems to astronomical forcing for specific pCO2 levels. This could be an elegant way to circumvent the disparity in time-scales between CO2 and orbital variability.

2. The authors do not provide their 50-simulation experimental design. It is essential to have an overview of the parameter settings for each simulation that was run in the framework of this study. The details on the settings of the 50 simulations could be given either in the form of a Table, or in the form of a figure, or in both forms. For good examples, please check Figure 2 and Table 1 in Araya-Melo et al. (2015, cp-11-45-2015), Figure 2 and Table 2 in Lord et al. (2017, cp-2017-57), and Figure 1 in Bounceur et al. (2015, esd-6-205-2015).
3. From Figure 6, it is very clear that precession has an important influence on the Asian Monsoon intensity, with higher rainfall when the index is minimum (i.e. Earth in perihelion during JJA, maximum northern hemisphere summer insolation). However, if I interpret PC2 in JJA temperature and PC2 in JJA precipitation correctly (Table 5 and Figures 7 and 8), it seems that a precession-driven increase in monsoonal rainfall coincides with a decrease in JJA temperature in the Asian Monsoon region. Such a decrease in temperature is remarkable, given that it occurs when northern hemisphere JJA insolation is maximum. This observation can either be explained by the consumption of incoming solar radiation as latent heat, or by a negative influence of the increased cloud cover on the radiation balance. Indeed, the reflective character of clouds contributes to the planetary albedo. In the revised version of the manuscript, I would like to read more discussion of paleoclimate mechanisms like this one.

4. Page 7, lines 23-25 and Figure 6: When I was first interpreting Figure 6, I was confused by the fact that the Asian Monsoon and the African monsoon seemed to respond to precession in the same way, despite the fact that they are located on opposite sides of the equator. It took me quite a while to realize that both monsoonal systems are responding to precession in the expected way: with intensified wet-season precipitation in the Asian Monsoon system when the Earth reaches perihelion in JJA (negative precession index), and intensified wet-season precipitation in the African Monsoon system when the Earth reaches perihelion in DFF (positive precession index). I only understood this after reading lines 23-25 (page 7) several times. Indeed, the authors define their monsoon-related "simple scalar metric" by the difference in rainfall in DJF and JJA, regardless of whether DJF is the wet or the dry season in the monsoonal system considered. This also explains why the panel of Figure 6 that is related to the African Monsoon shows negative values, whereas the panel that is related to the Asian Monsoon exhibits positive values. I would strongly advise the authors to think about ways to illustrate the monsoonal response to precession in a more intuitive way. Maybe the paper by Tuenter et al (2003) could provide some inspiration as to how to best present the response of a summer monsoon to precessional (and obliquity?) forcing. Also, why

is the South American monsoon system missing from Figure 6?

**Additional comments and recommendations**

Abstract line 5 and p. 2 lines 1-3: I would recommend being a little bit more conservative on the possible analogy between the PETM and the ongoing anthropogenic disturbance of the global carbon cycle. Also cite Zeebe et al. (2016, Nature Geoscience) here.

Abstract: The abstract reads too technical and vague. I find the following sentence particularly vague: "Two dimensional model output fields are reduced to scalar values through simple summarizing algorithms and by singular value decomposition." The reader gets very little information from this sentence. I would recommend rewriting the abstract, making it more results-oriented.

Page 2, line 30: suggestion: "The Earth resided in a greenhouse state"

Page 3, line 4: What do you mean with "high levels of radiative forcing"? Only eccentricity influences the total amount of solar energy received by the Earth. . . but the amplitude of that variability is only 0.15

Page 2, line 9: Either you provide the reader with information on which kind of evidence exists. Or you rewrite like: "During the PETM, the emission of organic carbon was initially in the form of methane, which later oxidized to CO2".

Page 2, line 23: "broadly similar" is quite a subjective, interpretative qualification. I find the Eocene paleogeography quite different from todays, given that the Tethys Ocean was still open. If you want to point to the similarity with the present-day, you could state that the majority of the continents were located in the northern hemisphere.

Page 4, line 10 and many other occurrences: "dominant periods of  100 kyr and 405 kyr". In an eccentricity power spectrum there are 4 peaks around 100 kyr, but only a single one at 405 kyr. Therefore, I would suggest the above notation.

[Figure]

Page 4, line 16: Jacques Laskar does not calculate time scales. He calculates astronomical solutions.

Page 5: Why is Section 3 not a subsection of Section 4 "Methods"?

Page 5, line 3: What is "T21"?

Page 6, lines 9-11: An injection of carbon into the atmosphere is measured in tons of C, whereas the concentration of $CO_2$ in the atmosphere is measured in ppm. These are thus two different things, with two different units. You have to rephrase this sentence to correct for that.

Page 6, lines 13-16: It's not immediately clear to me how knowledge on the phase relationship between carbon isotope excursions and the astronomical parameters would influence the experimental design of your study. If you would know these phase relationships, would you then have designed your experiments differently?

Page 6, line 26: What do you mean with "quasi-steady state"?

Page 7, line 7-8: The atmospheric circulation patterns during the Eocene were most definitely different from those in the modern world. I think you can remove the "are likely to".

Page 7 line 27: Spell out SVD

Page 8 line 9: Please provide the appropriate references where these criteria are defined.

Page 8 lines 23-24: The Figure 3 that you are referring to, only contains global annual mean SST's, not the Arctic winter SST's you are discussing.

Page 9, line 1: It is unclear to me what exactly you mean with "parametric uncertainty"

Page 10, line 17: JJA instead of JJF.

Page 10, line 15: Shouldn't this be Table 4?

The paper contains a few important shortcomings when it comes to appropriately referencing pre-existing work.

For example, the authors do not refer to the Deep-time Model Intercomparison Project (Deep-MIP, Lunt et al., 2017, gmd-10-889-2017). The authors do not frame their study within that project, nor do they differentiate their study from that project. A statement on this topic is indiscernible, given that both this study and the Deep-MIP project explicitly focus on simulating (early) Eocene warm climates and that both are using the same paleogeographic configuration from Herold et al. (2014).

The authors refer to Bounceur et al. (2015), who applied a "similar emulator approach" (p. 8 line 13). First of all, I am unsure whether that statement is technically correct. Secondly, this reference is missing from the reference list.

On page 4, line 28, the authors give credit to Ruddiman (2006, cp-2-43-2006) for noting "a relationship between obliquity and the extent of northern ice sheets". First of all, this is a Pleistocene-focused paper, of which I don't really see the relevance when discussing orbital configurations during the Eocene and possible influence on climate. Moreover, the relationship between obliquity-induced minima in NH summer insolation and ice age cycles was already suggested by Milutin Milankovitch in 1941.

**References**

Bounceur, Nabila, Michel Crucifix, and R. D. Wilkinson. "Global sensitivity analysis of the climate-vegetation system to astronomical forcing: an emulator-based approach." Earth System Dynamics 6.1 (2015): 205.

Herold, N., Buzan, J., Seton, M., Goldner, A., Green, J. A. M., Müller, R. D., Markwick, P., and Huber, M.: A suite of early Eocene (âĹij 55 Ma) climate model boundary conditions, Geosci. Model Dev., 7, 2077–2090, doi:10.5194/gmd-7-2077-2014, 2014.

Tuenter, Erik, et al. "The response of the African summer monsoon to remote and local forcing due to precession and obliquity." Global and Planetary Change 36.4 (2003):

219-235.

Zeebe, Richard E., Andy Ridgwell, and James C. Zachos. "Anthropogenic carbon release rate unprecedented during the past 66 million years." Nature Geoscience 9.4 (2016): 325-329.

Zeebe, R. E., T. Westerhold, K. Littler, and J. C. Zachos (2017), Orbital forcing of the Paleocene and Eocene carbon cycle, Paleoceanography, 32, doi:10.1002/2016PA003054

---

## Author Comment (AC1) · 15 Jul 2017

**Keery et al., Sensitivity of the Eocene Climate to CO2 and Orbital Variability**

Response to M. Crucifix (Referee)

Referee comments in black
Author responses in red

We are very grateful for this thorough review.

1 Summary

Keery et al. present a sensitivity analysis of the Eocene climate to four factors: $CO_2$ concentration, eccentricity, obliquity, and precession angle. They use, to this end, the PLASIM-GENIE model (details in their section 3) with suitable palaeogeography. The methodology relies on a 50-member hyper-cube sample of a 5-d space (one extra dummy variable was added), and linear modelling with a Information Criteria for model selection. Experiment output are summarised using fit-for-purpose summaries like "tropical-polar temperature difference" and monsoon indices, as well as principal components obtained from a singular value decomposition. The authors conclude on the importance of $CO_2$ for global mean temperature, and of the orbital elements for the spatial distribution and regional weather systems such as monsoons.

2 Main comments

1. The paper is in the line of a number of recent studies attempting to estimate the relative sensitivity of the climate system to $CO_2$ and orbital forcing, using a methodology founded on ensemble of experiments. This includes, in addition to the Holden et al. (2015) and Bounceur et al. (2015) cited, Araya-Melo et al. (2015) and Lord et al. (2017). Keery et al. is the only article to focus on the Eocene, which makes it an original contribution. It also uses a much simpler methodology than Araya-Melo et al. (2015), Bounceur et al. (2015), and Lord et al. (2017) because it uses linear regression instead of a Gaussian process emulator. In fact, the authors reference to the word "emulator" is slightly unusual because emulation is, in the climate literature, often used to designate statistical meta-modelling with a focus on uncertainty quantification. Claiming (p. 8) that a "similar emulator approach has been applied by Bounceur et al. 2015" is therefore somewhat misleading. Bounceur et al. and Araya-Melo et al. applied the developments of Oakley and O'Hagan (2004) with, in the case of Bounceur, the additional complication of the PCA emulator.

We agree that the comparison of our emulator to the emulators developed by Araya-Melo et al. (2015) and Bounceur et al. (2015) was misleading, and we will amend this section:

> Our emulator approach uses linear regression, rather than a Gaussian process, and is therefore simpler than the methods applied by Bounceur et al. (2015) in a study of the response of the climate-vegetation system in interglacial conditions to astronomical forcing, and by Araya-Melo et al. (2015) in their study of the Indian monsoon in the Pleistocene.

In spite of its simplicity, we are confident that our approach may be correctly described as an emulator, as it fulfills the criteria described by O'Hagan (2006), and cited by Araya-Melo et al. (2015):

- it is derived from a small number of model runs filling the entire multidimensional input space
- once the emulator is built, it is not necessary to perform any additional runs with the model

In passing, Araya-Melo et Lord used HadCM3 which shows that ensemble-based sensitivity analysis to orbital forcing is doable with GCMs (this qualifies the author's comment on line 15, p.2).

We will amend this paragraph to acknowledge recent ensemble studies using GCMs:

Climate simulations with high temporal and spatial resolution can be obtained from General Circulation Models (GCMs), but the requirement of GCMs for powerful computers and long run-times makes them difficult to deploy for large ensembles of model simulations and restricts their ability to investigate the large uncertainties in forcings and model parameterisations. Such ensembles are more practical with more heavily parameterised and hence more computationally efficient Earth system Models of Intermediate Complexity (EMICs), (Weber, 2010), although we note that Araya-Melo et al. (2015) and Lord et al. (2017) have deployed the GCM HadCM3 in ensemble-based studies of orbital forcing effects on climates of the Pleistocene and late Pliocene respectively.

Of course, the fact that other authors have adopted a more sophisticated methodology invalidates by no means the approach used by Keery et al.: there may be no need to use a sledgehammer to crack a nut. It remains that the methodological set up used here is a step backwards compared to recent studies, and this arguably requires some justification. How much do we lose with the linearity assumption, and which impact does it have on the uncertainties of the quantification of main effects? (see comment 3. more specifically on main effects).

As we have noted in our methods section, we have demonstrated that the linear models can be used to emulate PC scores with very high correlations to the PC scores derived directly through SVD, with examples from temperature and precipitation shown in Table 3.  We can therefore be confident that main effects derived from the linear models are robust.  We will amend the text:

Unlike linear models, GP models are intrinsically stochastic and give a more accurate quantification of their own error in emulating the input data. However, GP models can become computationally demanding in high dimensional space, and their results can be more difficult to interpret.

2. Experiment design. The authors do not say much about the ensemble design, except that this is a latin hypercube. There are many ways to do a latin hypercube, and it usually involves additional constraints.

We will add a detailed description of the method used to generate the latin hypercube in an appendix, include forcing factor values for the full ensemble in a new Table, and we will amend the main text:

The present study has been designed to facilitate direct comparison between the results for specific ensemble members and their direct counterparts in a related study using the EMIC GENIE-1 (Edwards and Marsh, 2005), which will include additional forcing parameters not used by this PLASIM-GENIE study. We have applied an iterative method to generate a pair of corresponding hypercubes with five and eleven dimensions for the PLASIM-GENIE and GENIE-1 studies respectively, in which the minimum Euclidean distance between any two points is maximised, and linear correlation between any two parameters is minimised. Details of the steps taken to generate the hypercubes are provided in Appendix A.  The absolute value of the correlation coefficient r did not exceed 0.1 for any pair of input (forcing and dummy) parameters.  Uniform ranges for each of the PLASIM-GENIE forcing parameters and the dummy parameter are shown in Table 1, and the values applied in all 50 PLASIM-GENIE ensemble members are shown in Table 2.

Table 2 Forcing factors and dummy values for each member in the ensemble.  Precession = $\omega$, the angle between the moving vernal equinox and the longitude of perihelion.

| Member (-) | $CO_2$ (ppm) | Eccentricity (-) | Precession (°) | Obliquity (°) | Dummy (-) |
|---|---|---|---|---|---|
| 1 | 975.6 | 0.0022 | 142.5 | 22.37 | 0.822 |
| 2 | 2418.7 | 0.0256 | 165.2 | 23.95 | 0.907 |
| 3 | 1259.4 | 0.0007 | 307.1 | 23.91 | 0.323 |
| 4 | 801.3 | 0.0163 | 270.4 | 23.50 | 0.276 |
| 5 | 1720.1 | 0.0559 | 206.7 | 23.82 | 0.402 |
| 6 | 327.1 | 0.0595 | 135.9 | 23.53 | 0.681 |
| 7 | 2937.7 | 0.0418 | 287.1 | 22.53 | 0.650 |
| 8 | 1200.3 | 0.0237 | 313.2 | 24.12 | 0.978 |
| 9 | 1420.7 | 0.0158 | 297.1 | 23.86 | 0.931 |
| 10 | 2157.6 | 0.0432 | 100.6 | 23.74 | 0.661 |
| 11 | 1791.7 | 0.0241 | 247.2 | 23.43 | 0.429 |
| 12 | 2369.0 | 0.0425 | 78.9 | 22.65 | 0.167 |
| 13 | 2502.9 | 0.0296 | 0.5 | 22.69 | 0.122 |
| 14 | 2149.2 | 0.0405 | 249.9 | 24.23 | 0.347 |
| 15 | 1061.7 | 0.0394 | 40.9 | 23.94 | 0.189 |
| 16 | 711.3 | 0.0199 | 274.6 | 22.08 | 0.913 |
| 17 | 1817.1 | 0.0578 | 291.4 | 23.08 | 0.888 |
| 18 | 722.1 | 0.0463 | 195.8 | 24.38 | 0.865 |
| 19 | 2988.5 | 0.0039 | 110.1 | 24.40 | 0.049 |
| 20 | 539.4 | 0.0251 | 212.5 | 23.29 | 0.234 |
| 21 | 450.6 | 0.0335 | 96.1 | 22.28 | 0.674 |
| 22 | 2700.1 | 0.0049 | 165.9 | 23.66 | 0.630 |
| 23 | 2025.4 | 0.0320 | 189.4 | 23.63 | 0.087 |
| 24 | 2268.7 | 0.0308 | 233.3 | 22.86 | 0.461 |
| 25 | 1447.2 | 0.0364 | 62.0 | 23.40 | 0.541 |
| 26 | 1168.3 | 0.0300 | 147.4 | 22.97 | 0.947 |
| 27 | 1317.6 | 0.0377 | 12.4 | 23.04 | 0.714 |
| 28 | 1639.5 | 0.0265 | 150.9 | 22.98 | 0.524 |
| 29 | 399.0 | 0.0589 | 262.7 | 23.46 | 0.028 |
| 30 | 2876.3 | 0.0411 | 203.0 | 22.05 | 0.608 |
| 31 | 2611.1 | 0.0170 | 54.3 | 22.84 | 0.746 |
| 32 | 2831.7 | 0.0564 | 187.2 | 23.72 | 0.696 |
| 33 | 1998.5 | 0.0372 | 278.8 | 24.19 | 0.805 |
| 34 | 1465.0 | 0.0439 | 38.9 | 23.50 | 0.376 |
| 35 | 1660.0 | 0.0109 | 85.3 | 22.88 | 0.896 |
| 36 | 2393.7 | 0.0587 | 127.9 | 24.27 | 0.191 |
| 37 | 286.3 | 0.0004 | 27.1 | 23.99 | 0.391 |
| 38 | 667.4 | 0.0509 | 116.5 | 22.71 | 0.569 |
| 39 | 2246.8 | 0.0450 | 317.4 | 22.90 | 0.103 |
| 40 | 2334.2 | 0.0096 | 294.7 | 23.61 | 0.532 |
| 41 | 2968.2 | 0.0346 | 329.8 | 22.51 | 0.314 |
| 42 | 768.2 | 0.0085 | 218.3 | 23.00 | 0.000 |
| 43 | 925.8 | 0.0450 | 327.2 | 24.32 | 0.753 |
| 44 | 384.5 | 0.0081 | 60.6 | 22.59 | 0.436 |
| 45 | 850.7 | 0.0551 | 322.9 | 23.21 | 0.459 |
| 46 | 1112.8 | 0.0150 | 356.7 | 23.27 | 0.579 |
| 47 | 1255.8 | 0.0116 | 212.2 | 22.31 | 0.487 |
| 48 | 1124.1 | 0.0530 | 343.7 | 22.40 | 0.065 |
| 49 | 2113.9 | 0.0276 | 9.9 | 22.19 | 0.856 |
| 50 | 1681.0 | 0.0354 | 175.5 | 22.45 | 0.287 |

In fact this experiment design raises some doubts. For example, why are some secondary structures (periodic up and downs) apparent in the response to obliquity, Figure 5, middle column? Is this just a subjective visual impression?

We have created an additional plot of the two forcing factors obliquity and $CO_2$, for discussion, but not for inclusion in the paper, and this shows a very similar pattern to the obliquity-MAT subplot in Figure 5, with corresponding clusters and the same slight impression of periodicity. We can therefore be confident that the apparent periodicity noted by the reviewer in the model output is an artefact of randomly generated structure in the model input.

[Figure]

Figure R1   Obliquity plotted against $CO_2$.

One potentially problematic element is the definition of the sampled astronomical space. It seems that latin hypercube sampling is made on axes along e, ω (longitude of perihelion) and ε. If this is what the authors have been doing then this is non-physical. We know that the astronomical forcing generates effects through seasonal and daily insolation, which are very well approximated by linear functions of e sinω (which the authors call the precession index on Fig. 6) and e cosω. This is the reason why several authors have chosen to sample the astronomical space following the axes e sinω and e cosω and regress against these components. Presumably the regression analysis by Keery is indeed done against these indices but the text is not always clear. Lines 1-2 p. 8. rather suggest that the explanatory variables where sinω and cosω (instead of their multiplication by e) and the lines 4-5 p. 11 are quite confusing. Hopefully the choice of regression variables is mainly matter of text clarification, but the design of the latin hypercube may have a more fundamental problem.

We have indeed constructed our hypercube by sampling independently on *e*, $\omega$ (longitude of perihelion) and $\varepsilon$, but we do not agree that this is non-physical, as there are no combinations of these parameters which can be excluded for the early Eocene period.  If we have ignored any information which would imply that some combinations are less likely to have occurred than others (we are not aware of any), then this would only result in a minor reduction in the efficiency with which we fill our state space.  We note that precessional effects are well approximated by *e*sin$\omega$ and *e*cos$\omega$, and that several authors have chosen to sample and regress against these components, but we have chosen not to take this approach, as it would not allow any climatic effects of eccentricity which may exist independently of precession to be identified.

We will amend our description of the forcing factors:

In order to investigate the sensitivity of the Eocene climate to variation in atmospheric $CO_2$ and orbital parameters, we have constructed an ensemble of 50 model configurations, each with a unique set of forcing parameters comprising atmospheric $CO_2$, eccentricity (*e*), obliquity ($\varepsilon$) and precession ($\omega$), the angle on the Earth's orbit around the Sun between the moving vernal equinox and the longitude of perihelion (Berger et al., 1993).  When *e* is zero, the Earth's distance from the Sun is constant at all points

on the orbit, so there is no precessional effect.  The magnitude of precessional effects is controlled by $e$, while phase is controlled by $\omega$, so precessional effects are commonly described by the precession index given by $e\sin\omega$.  The only orbital parameter which alters the total annual solar radiation received by the Earth is $e$, although the range of variation is very small.  We include $e$ and $\omega$ as separate and independent forcing parameters, rather than combined as the precession index, or in the form $e\cos\omega$.  This approach does not make the assumption that the only effect of eccentricity on the Earth's climate is through its effect on the amplitude of the precession cycle, but allows experimental results to be examined for effects of $e$ and $\omega$ either separately or in combination.   An additional dummy parameter is included to test for possible overfitting of relationships between forcing parameters and model output fields.

We will also amend our description of our preparation of the forcing factors for linear modelling:

Values of the forcing parameters $CO_2$, $e$ and $\varepsilon$ (with its very small angular range considered to be approximately linear) were normalised to the range [-1, 1] and combined with $\sin\omega$ and $\cos\omega$ to form 50-element column vectors representing the forcing factors.

3. There may be some confusion about the meaning of the main effects. Saltelli does not use the phrase "first order" to mean linear approximation. In a case where only one factor would matter (be the relationship linear on not), the main and total effects would match (Saltelli et al. (2004), ch. 1 states clearly the definitions; or refer again to Oakley and O'Hagan (2004)). More generally, computing main and total effects is not trivial and always involves some approximations.More details on their computation would be welcome.

We will amend the text to provide more details on the computation of the main effects and total effects:

In order to analyse the results of each of our linear models, we apply the method described in detail by Holden et al. (2015) to derive the main effects (Oakley and O'Hagan, 2004), which provide a measure of the variation in the linear model output due to each of the terms (first order, second order and cross products), derived from their coefficients, and total effects (Homma and Saltelli, 1996), which separate the effect of each forcing parameter on the variation in the model output.  Since the forcing factors are scaled within the range [-1, 1], the variances of the first order, second order and cross product terms can be approximated as $\frac{1}{3}$, $\frac{1}{9}$ and $\frac{4}{45}$ respectively, and we have applied these values as scaling factors in calculating the main effects and total effects.

4. Singular value decomposition is a great dimensionality reduction methodology, but how much is learned by analysing the behaviour of principal components separately is a more contentious subject. Identification of principal components can be fragile to some implementation details, such as, e.g. grid area weighting and experiment design, and the physical phenomena which give rise to climate variability need not be orthogonal. In fact physical modes may project poorly on the orthogonal vectors (Monahan and Fyfe, 2006). These caveats implicitly acknowledged by the authors (p. 11, ll. 20-21) but this state-of-affairs poses some questions about the emphasis on principal components in this article.

We will amend the text to acknowledge these caveats:

We perform a singular value decomposition to identify the PCs and empirical orthogonal functions (EOFs) of temperature and precipitation fields in the full ensemble, although we note that climate variability may not be due to physical processes which vary orthogonally, and identification of PCs can be influenced by aspects of the experimental design.

3 Minor (scientific) comments

• How Fig. 2 should be interpreted is not entirely clear since the ensemble was not explicitly designed so that the ensemble mean is an estimate of the Eocene climate mean.

Figures 2 and 4 are included to provide an illustrative summary of the spatial distribution and variation of temperature and precipitation in the full ensemble output, without implying that the ensemble mean is an estimate of the Eocene climate mean.  We will amend the text:

> Analysis of the model results has focused on variation in surface air temperature and precipitation in both winter and summer in each hemisphere, although it should be noted that our experiment has not been designed such that mean values in our ensemble output represent direct estimates of the Eocene climate mean.

4 Minor (editorial) comments

• Introduce subtitle after section 2.

We will introduce the subtitle 'Climate of the Early Eocene'

• Material about cyclostratigraphy under section 2.1.2. may possibly be considered for shortening as slightly out of scope of the article. This said this is an interesting read.

We would prefer to retain the section on cyclostratigraphy in full, as we believe it provides important details which are relevant to our experimental design, particularly our selection of independent orbital values, and the separation of $e$ and $\omega$.

• PLASIM-GENIE does not need a specific section: it can fall under section 3.Methods.
This section will be moved to the Methods as suggested by both reviewers.

• p. 6 reference Gough (1981) is mistakenly repeated.

The duplicated reference will be removed.

• p. 7, the sentence "We apply the linear algebraic tool SVD" sounds unnecessarily sophisticated. Why not "We perform a singular value decomposition to identifyprincipal components"

We will amend this sentence:

> We perform a singular value decomposition to identify the PCs and empirical orthogonal functions (EOFs) of temperature and precipitation fields in the full ensemble.

• p. 10, l. 27 : define the word "precession" precisely.

We will make amendments to the text to define precession ($\omega$), and the precession index ($e\sin\omega$).  See our response to an earlier comment.

• p. 12, ll. 13-17 : introducing new results so close to the closing words is usually not encouraged.

We will delete these results, as further analysis suggests it is difficult to draw any very useful conclusions from the extra experiment, and we will amend the text to include the reference to Anagnostou et al. (2016):

> If atmospheric $CO_2$ remained within a narrower range throughout the period, for example in the range 700 to 1800 ppm indicated for the early Eocene by Anagnostou et al. (2016) in a recent study using boron isotopes, then outside of short-lived hyperthermals, the relative influence of $CO_2$ and orbital inputs might have been more evenly balanced.

5 Digital material

• Relevant data of the Eocene runs (at least the summaries and experiment input data) could be provided.

We will include the values of forcing factors for the 50 member ensemble in a new Table.

References

P. A. Araya-Melo, M. Crucifix, and N. Bounceur. Global sensitivity analysis of the indian monsoon during the pleistocene. Climate of the Past, 11:45–61, 2015. doi: 10.5194/cp-11-45-2015. URL http://www.clim-past.net/11/45/2015/.

N. Bounceur, M. Crucifix, and R. D. Wilkinson. Global sensitivity analysis of the climatevegetation system to astronomical forcing: an emulator-based approach. Earth System Dynamics, 6:205–224, 2015. doi: 10.5194/esd-6-205-2015. URL http://www.earth-syst-dynam.net/6/205/2015/.

P. B. Holden, N. R. Edwards, P. H. Garthwaite, and R. D. Wilkinson. Emulation and interpretation of high-dimensional climate model outputs. Journal of Applied Statistics, pages 1–18, 2015. doi: 10.1080/02664763.2015.1016412. URL http://dx.doi.org/10.1080/02664763.2015. 1016412.

N. S. Lord, M. Crucifix, D. J. Lunt, M. C. Thorne, N. Bounceur, H. Dowsett, C. L. O'Brien, and A. Ridgwell. Emulation of long-term changes in global climate: Application to the late pliocene and future. Climate of the Past Discussions, page 1–47, 2017. ISSN 1814-9359. doi: 10.5194/cp-2017-57. URL http://dx.doi.org/10.5194/cp-2017-57.

A. H. Monahan and J. C. Fyfe. On the nature of zonal jet eofs. Journal of Climate, 19: 6409–6424, 2006. ISSN 1520-0442. doi: 10.1175/jcli3960.1. URL http://dx.doi.org/10.1175/ JCLI3960.1.

J. E. Oakley and A. O'Hagan. Probabilistic sensitivity analysis of complex models: a bayesian approach. Journal of the Royal Statistical Society: Series B (Statistical Methodology), 66: 751–769, 2004. doi: 10.1111/j.1467-9868.2004.05304.x.

A. Saltelli, S. Tarentola, F. Campolongo and M. Ratto, Sensitivity Analysis in Practice: A guide to assessing scientific methods John Wiley and Sons Ltd., 219pp., 2004.

---

## Author Comment (AC2) · 15 Jul 2017

**Keery et al., Sensitivity of the Eocene Climate to CO2 and Orbital Variability**

Response to D. De Vleeschouwer (Referee)

Referee comments in black
Author responses in red

We are very grateful for this thorough review.

This paper reports on an ensemble of 50 Eocene climate-model simulations, each of which characterized by a different combination of eccentricity, obliquity, precession and atmospheric CO2 concentration. The climate model is the PLASIM-GENIE model, a new model of intermediate complexity, recently introduced by Holden et al. (2016). The study aims to summarize the ensemble of paleoclimate simulations by looking at what-they-call "simple metrics", principal component analysis and an emulator approach. This study provides a couple of interesting results. The first is the existence of a seaice-related threshold mechanism in the northern hemispheric high latitudes. From Figure 2 and 3, it seems that when a certain threshold in the extent of DJF-sea-ice is exceeded, temperatures (both sea-surface and maritime air temperatures) drop significantly. It would be interesting to read the author's opinion how this compares to the recent findings of modeling work by Zeebe et al. (2017), who found that "High-latitude mechanisms are unlikely drivers of orbitally paced changes in the late Paleocene-early Eocene". The interesting role of (seasonal) sea-ice in the climate system of the early Eocene aspect remains, however, rather underdeveloped in the present version of the paper.

In our discussion of Figs. 2 & 3 [page 9, line 10] we have stated: "The variation in TPTD across the ensemble thus appears to be essentially driven by the strength of snow and ice albedo feedback", and a little further on, in our discussion of Fig. 5 [page 9, line 22], in particular the plot of $CO_2$ v northern winter TPTD we have declared: "and it can also be seen that $CO_2$ strongly affects the northern TPTD in the winter, but not in the summer, when the combined influence of obliquity and precession index is discernible, suggesting that temperature proxies with seasonal bias may have a significant orbital imprint. The plot of atmospheric $CO_2$ against N. Winter TPTD shows a change in gradient at approximately 1000 ppm CO2 and 32°C. This may be related to the logarithmic dependence of radiative forcing on CO2 concentration, as well as the disappearance of ice above some threshold level, cf Fig. 3."

We will add the additional comment:

A possible sea ice related threshold mechanism influencing both SST and maritime air temperature in high northern latitudes may be observed in Fig. 3, and this is strongly associated with the increase in northern winter TPTD at low $CO_2$ levels. Zeebe et al. (2017) have analysed a high resolution benthic isotope record covering the late Palaeocene - early Eocene, and have concluded that orbitally paced cycles are unlikely to have been driven by high latitude mechanisms. Our PLASIM-GENIE modelling suggests that northern TPTD is not orbitally paced in the winter, being controlled by $CO_2$, but is orbitally paced in the summer, by a combination of obliquity and precession.

The second interesting aspect is the distinct response to precession of monsoonal precipitation and temperature in the different monsoonal systems (e.g. Figure 6). The description and discussion of these Eocene paleoclimate simulations is useful and perfectly fits the scope of the journal. The current version of the manuscript is, however, unsatisfactory for publication in Climate of the Past for the reasons listed below.

**Major Comments**

1. One of the major conclusions in the current version of the manuscript, is that 95The emulator approach adopted in this study allows for estimating the response of different aspects of the climate system (e.g. wet-season monsoonal precipitation) over the full input space. It would -for example- be interesting to see the response of precipitation and temperatures in the different monsoonal systems to astronomical forcing for specific pCO2 levels. This could be an elegant way to circumvent the disparity in time-scales between CO2 and orbital variability.

We have amended the subplots for obliquity and precession index in Figures 5 and 6 to denote the $CO_2$ level on a continuous colour scale.  This approach gives a simple visual indication of which relationships between the astronomical forcing factors and the temperature and precipitation simple metrics are influenced by $CO_2$. Figure 6 also now includes an additional row of subplots for the American monsoon index.

We have applied emulators derived from linear modelling of the forcing factors and monsoon indices, to estimate values of each of the monsoon indices over the full range of precession ($\omega$), with fixed high eccentricity ($e$), for low and high values of $CO_2$, and low and high values of obliquity ($\varepsilon$).

We will make amendments to the abstract:

> The results demonstrate the importance of orbital variation as an agent of change in climates of the past, and we demonstrate that emulators derived from our modelling output can be used as rapid and efficient surrogates of the full complexity model, to provide estimates of early Eocene climate conditions from any set of forcing parameters.

and to the final paragraph of the introduction:

> By applying the linear modelling and emulation methods of Holden et al. (2015), we regress both the simple scalar metrics and the SVD reduced dimension model outputs onto the forcing parameters, and from the derived relationships, we infer main effects denoting the effect of each explanatory term in the linear model, and total effects denoting the effect of each forcing parameter, on the variation in the scalar metrics and on the temperature and precipitation output fields.  We demonstrate that emulators derived in respect of tropical precipitation metrics can be used to estimate Eocene monsoonal responses to any combination of GHG and orbital forcing parameter values.

We will add new Figures 11, 12 and 13, plotting emulated values of the Asian, African and American monsoon indices.

We will add a paragraph to the Results section:

> We apply the linear models derived from the forcing factors and monsoon indices as emulators to estimate values of monsoon indices corresponding to the full range of precession ($\omega$), with eccentricity fixed at its high limit of 0.06, low and high values of $CO_2$ (300 ppm and 3000 ppm), and low and high values of obliquity (22.0° and 24.5°).  Precession index ($e\sin\omega$) and emulated values of the Asian, African and American monsoon indices are plotted in Figures 11, 12 and 13 respectively. Relationships between the precession index and the monsoon indices which are visually suggested in Figure 6 are shown with clear structure in Figures 11, 12 and 13.  In each of the monsoon areas, the increase in precipitation due to precession effects is more pronounced at high atmospheric concentration of $CO_2$, and also at high obliquity.

We will add a paragraph to the Summary;

> We have demonstrated that emulators derived from linear modelling of the PLASIM-GENIE ensemble results can be used as a rapid and efficient method of estimating early Eocene climate conditions from any set of forcing parameters, without the need for further deployment of the EMIC.

[Figure]

Figure 5 reworked with $CO_2$ plotted in colour in obliquity and precession plots (blue = low, red = high)

[Figure]

Figure 6 reworked with $CO_2$ plotted in colour in obliquity and precession plots (blue = low, red = high). Additional (bottom) row plots the forcing factors against the American monsoon index.

[Figure]

Figure 11: Emulated values of the Asian monsoon index, for the full range of the precession index ($e\sin\omega$), at low and high values of $CO_2$ and obliquity ($\varepsilon$).

[Figure]

Figure 12: Emulated values of the African monsoon index, for the full range of the precession index ($e\sin\omega$), at low and high values of $CO_2$ and obliquity ($\varepsilon$).

[Figure]

Figure 13: Emulated values of the American monsoon index, for the full range of the precession index ($e\sin\omega$), at low and high values of $CO_2$ and obliquity ($\varepsilon$).

2. The authors do not provide their 50-simulation experimental design. It is essential to have an overview of the parameter settings for each simulation that was run in the framework of this study. The details on the settings of the 50 simulations could be given either in the form of a Table, or in the form of a figure, or in both forms. For good examples, please check Figure 2 and Table 1 in Araya-Melo et al. (2015, cp-11-45-2015), Figure 2 and Table 2 in Lord et al. (2017, cp-2017-57), and Figure 1 in Bounceur et al. (2015, esd-6-205-2015).

We will include the values of the forcing factors and the dummy variable for the ensemble in a new table (Table 2).

We note that Araya-Melo et al. (2015) constrained their experiment to exclude non-physical combinations of $CO_2$ and sea ice, and their Figure 2 includes an informative subplot showing fairly strong inverse correlation between $CO_2$ and sea ice. In our study, however, we do not have a priori information with which to constrain any combinations of our forcing factors, each of which is sampled independently to maximise state space coverage and to minimise correlations between the forcing factors. We include in this response a new figure showing cross-plots and r coefficients of all of the forcing factors and the dummy parameter, which illustrate both the coverage of the state space, and the very low correlation between any of the factors. We do not

consider that this figure, or a variation, could add significant information to that included in the text, which will be amended to include the statement:

The absolute value of the correlation coefficient r did not exceed 0.1 for any pair of input (forcing and dummy) parameters.

Table 2 Forcing factors and dummy values for each member in the ensemble.  Precession = $\omega$, the angle between the moving vernal equinox and the longitude of perihelion.

| Member (-) | $CO_2$ (ppm) | Eccentricity (-) | Precession (°) | Obliquity (°) | Dummy (-) |
|---|---|---|---|---|---|
| 1 | 975.6 | 0.0022 | 142.5 | 22.37 | 0.822 |
| 2 | 2418.7 | 0.0256 | 165.2 | 23.95 | 0.907 |
| 3 | 1259.4 | 0.0007 | 307.1 | 23.91 | 0.323 |
| 4 | 801.3 | 0.0163 | 270.4 | 23.50 | 0.276 |
| 5 | 1720.1 | 0.0559 | 206.7 | 23.82 | 0.402 |
| 6 | 327.1 | 0.0595 | 135.9 | 23.53 | 0.681 |
| 7 | 2937.7 | 0.0418 | 287.1 | 22.53 | 0.650 |
| 8 | 1200.3 | 0.0237 | 313.2 | 24.12 | 0.978 |
| 9 | 1420.7 | 0.0158 | 297.1 | 23.86 | 0.931 |
| 10 | 2157.6 | 0.0432 | 100.6 | 23.74 | 0.661 |
| 11 | 1791.7 | 0.0241 | 247.2 | 23.43 | 0.429 |
| 12 | 2369.0 | 0.0425 | 78.9 | 22.65 | 0.167 |
| 13 | 2502.9 | 0.0296 | 0.5 | 22.69 | 0.122 |
| 14 | 2149.2 | 0.0405 | 249.9 | 24.23 | 0.347 |
| 15 | 1061.7 | 0.0394 | 40.9 | 23.94 | 0.189 |
| 16 | 711.3 | 0.0199 | 274.6 | 22.08 | 0.913 |
| 17 | 1817.1 | 0.0578 | 291.4 | 23.08 | 0.888 |
| 18 | 722.1 | 0.0463 | 195.8 | 24.38 | 0.865 |
| 19 | 2988.5 | 0.0039 | 110.1 | 24.40 | 0.049 |
| 20 | 539.4 | 0.0251 | 212.5 | 23.29 | 0.234 |
| 21 | 450.6 | 0.0335 | 96.1 | 22.28 | 0.674 |
| 22 | 2700.1 | 0.0049 | 165.9 | 23.66 | 0.630 |
| 23 | 2025.4 | 0.0320 | 189.4 | 23.63 | 0.087 |
| 24 | 2268.7 | 0.0308 | 233.3 | 22.86 | 0.461 |
| 25 | 1447.2 | 0.0364 | 62.0 | 23.40 | 0.541 |
| 26 | 1168.3 | 0.0300 | 147.4 | 22.97 | 0.947 |
| 27 | 1317.6 | 0.0377 | 12.4 | 23.04 | 0.714 |
| 28 | 1639.5 | 0.0265 | 150.9 | 22.98 | 0.524 |
| 29 | 399.0 | 0.0589 | 262.7 | 23.46 | 0.028 |
| 30 | 2876.3 | 0.0411 | 203.0 | 22.05 | 0.608 |
| 31 | 2611.1 | 0.0170 | 54.3 | 22.84 | 0.746 |
| 32 | 2831.7 | 0.0564 | 187.2 | 23.72 | 0.696 |
| 33 | 1998.5 | 0.0372 | 278.8 | 24.19 | 0.805 |
| 34 | 1465.0 | 0.0439 | 38.9 | 23.50 | 0.376 |
| 35 | 1660.0 | 0.0109 | 85.3 | 22.88 | 0.896 |
| 36 | 2393.7 | 0.0587 | 127.9 | 24.27 | 0.191 |
| 37 | 286.3 | 0.0004 | 27.1 | 23.99 | 0.391 |
| 38 | 667.4 | 0.0509 | 116.5 | 22.71 | 0.569 |
| 39 | 2246.8 | 0.0450 | 317.4 | 22.90 | 0.103 |
| 40 | 2334.2 | 0.0096 | 294.7 | 23.61 | 0.532 |
| 41 | 2968.2 | 0.0346 | 329.8 | 22.51 | 0.314 |
| 42 | 768.2 | 0.0085 | 218.3 | 23.00 | 0.000 |
| 43 | 925.8 | 0.0450 | 327.2 | 24.32 | 0.753 |
| 44 | 384.5 | 0.0081 | 60.6 | 22.59 | 0.436 |
| 45 | 850.7 | 0.0551 | 322.9 | 23.21 | 0.459 |
| 46 | 1112.8 | 0.0150 | 356.7 | 23.27 | 0.579 |
| 47 | 1255.8 | 0.0116 | 212.2 | 22.31 | 0.487 |
| 48 | 1124.1 | 0.0530 | 343.7 | 22.40 | 0.065 |
| 49 | 2113.9 | 0.0276 | 9.9 | 22.19 | 0.856 |
| 50 | 1681.0 | 0.0354 | 175.5 | 22.45 | 0.287 |

[Figure]

Figure R2  Correlation plots and r coefficients between all forcing factors.

3. From Figure 6, it is very clear that precession has an important influence on the Asian Monsoon intensity, with higher rainfall when the index is minimum (i.e. Earth in perihelion during JJA, maximum northern hemisphere summer insolation). However, if I interpret PC2 in JJA temperature and PC2 in JJA precipitation correctly (Table 5 and Figures 7 and 8), it seems that a precession-driven increase in monsoonal rainfall coincides with a decrease in JJA temperature in the Asian Monsoon region. Such a decrease in temperature is remarkable, given that it occurs when northern hemisphere JJA insolation is maximum. This observation can either be explained by the consumption of incoming solar radiation as latent heat, or by a negative influence of the increased cloud cover on the radiation balance. Indeed, the reflective character of clouds contributes to the planetary albedo. In the revised version of the manuscript, I would like to read more discussion of paleoclimate mechanisms like this one.

This temperature decrease is indeed observed in the model results for the Asian monsoon.  We will augment the text to describe this effect more clearly:

An increase in the second PC scores for JJA precipitation in the Asian monsoon region (Fig. 8) corresponds to a decrease in the second PC scores for JJA temperature (Fig. 7), and as already noted, the second PC scores for both temperature and precipitation in JJA are strongly correlated to the precession

index. This temperature reduction during the Asian monsoon was also observed by Holden et al. (2014), and attributed to a reduction in incoming solar radiation due to increased cloud cover, and an increase in energy lost as latent heat with an increase in evaporation.

4. Page 7, lines 23-25 and Figure 6: When I was first interpreting Figure 6, I was confused by the fact that the Asian Monsoon and the African monsoon seemed to respond to precession in the same way, despite the fact that they are located on opposite sides of the equator. It took me quite a while to realize that both monsoonal systems are responding to precession in the expected way: with intensified wet-season precipitation in the Asian Monsoon system when the Earth reaches perihelion in JJA (negative precession index), and intensified wet-season precipitation in the African Monsoon system when the Earth reaches perihelion in DFF (positive precession index). I only understood this after reading lines 23-25 (page 7) several times. Indeed, the authors define their monsoon-related "simple scalar metric" by the difference in rainfall in DJF and JJA, regardless of whether DJF is the wet or the dry season in the monsoonal system considered. This also explains why the panel of Figure 6 that is related to the African Monsoon shows negative values, whereas the panel that is related to the Asian Monsoon exhibits positive values. I would strongly advise the authors to think about ways to illustrate the monsoonal response to precession in a more intuitive way. Maybe the paper by Tuenter et al (2003) could provide some inspiration as to how to best present the response of a summer monsoon to precessional (and obliquity?) forcing. Also, why is the South American monsoon system missing from Figure 6?

We have amended our monsoon indices so that each is now derived by subtracting winter precipitation from summer precipitation, as suggested. Figure 6 has been altered accordingly, and now also includes a row for the American monsoon index, an entry for which will be added to the table of total effects of forcing parameters on simple scalar metrics (presently Table 3; will be Table 4).

We will amend the text to reflect the changes to the monsoon indices:

In this study, we derive simple scalar metrics to denote indices for monsoons for Asia, Africa and South America by subtracting winter rainfall from summer rainfall for defined geographical regions, denoted on Fig. 1, and selected for their similarity to monsoonal regions in the modern continental configuration.

We will amend our comments on Figures 5 and 6 in the Results section, following addition of the American monsoon index, and the use of colour in these Figures:

In Figs. 5 and 6, $CO_2$, obliquity ($\varepsilon$) and precession index ($e\sin\omega$) are plotted against MAT, northern seasonality, northern winter TPTD and northern summer TPTD (Fig. 5), and southern winter polar OLC, northern winter polar OLC, Asian monsoon index, African monsoon index and American monsoon index (Fig. 6). Subplots for obliquity and precession index in Figures 5 and 6 denote the $CO_2$ level on a continuous colour scale.

and we will add the comment:

The American monsoon index is fairly strongly correlated with the precession index at high levels of $CO_2$, and negatively correlated with $CO_2$ at low levels of $CO_2$.

We note that the study by Tuenter et al. (2003) included six experimental setups, with each one comprising either maximum or minimum values of obliquity, and maximum, minimum or zero values of precession. They were therefore able to illustrate their results in the form of spatial patterns of the differences in output values for pairs of experiments with contrasting values of one or both forcing factors. This approach is not appropriate for our 50 member ensemble, with uncorrelated forcing factor values, in which no pairs of experiments can be identified for this type of comparison.

**Additional comments and recommendations**

Abstract line 5 and p. 2 lines 1-3: I would recommend being a little bit more conservative on the possible analogy between the PETM and the ongoing anthropogenic disturbance of the global carbon cycle. Also cite Zeebe et al. (2016, Nature Geoscience) here.

We will amend the text to clarify the importance of the PETM, particularly its importance as the closest, if not perfect, analogue to anthropogenic climate change:

Since the PETM is the most recent period in Earth's history for which estimated atmospheric GHG concentrations are similar in magnitude to those of the present-day, and expected to arise from fossil fuel burning, the PETM may provide a valuable analogue for anthropogenic climate change.

We will also cite Zeebe et al (2016) in the first paragraph of the introduction.

Abstract: The abstract reads too technical and vague. I find the following sentence particularly vague: "Two dimensional model output fields are reduced to scalar values through simple summarizing algorithms and by singular value decomposition." The reader gets very little information from this sentence. I would recommend rewriting the abstract, making it more results-oriented.

We will delete this sentence, and make amendments to the abstract to make it less vague, with more focus on the results, including our additional work using the emulators.

Page 2, line 30: suggestion: "The Earth resided in a greenhouse state"

We don't understand the reason behind this suggestion. Our intention was to emphasise that the greenhouse state had been continuous since the early Cretaceous, so we will leave the sentence unchanged.

Page 3, line 4: What do you mean with "high levels of radiative forcing"? Only eccentricity influences the total amount of solar energy received by the Earth: : : but the amplitude of that variability is only 0.15

Huber & Caballero (2011) used $CO_2$ as a proxy for all changes to incoming and outgoing radiation. They commented "We have not addressed whether the enhanced radiative forcing was due to $pCO_2$, methane, other greenhouse gases, novel cloud feedbacks, or other "missing" factors. We have also not established whether large forcing is actually necessary, the alternative being high values of climate sensitivity as in the study of Heinemann et al. (2009) and only moderate increases in forcing."

We will amend the text to clarify this:

Huber and Caballero (2011), hereafter HC11, have demonstrated that with sufficiently high levels of $CO_2$ (as a proxy for all forms of radiative forcing), climate models can generate global air temperature distributions in broad agreement with the proxy temperature measurements.

Page 2, line 9: Either you provide the reader with information on which kind of evidence exists. Or you rewrite like: "During the PETM, the emission of organic carbon was initially in the form of methane, which later oxidized to CO2".

We will amend the text (Page 3, line 9), to give brief details of the evidence, and we will include an additional citation:

There is some evidence from analysis and modelling of the timing and duration of variations in $\delta^{13}C$ and $\delta^{13}O$ observed in nannoplankton fossils that some of the GHG emissions were initially in the form of $CH_4$ (Dickens, 2011; Lunt et al., 2011; Thomas et al., 2002), which is rapidly oxidised in the atmosphere to $CO_2$.

Page 2, line 23: "broadly similar" is quite a subjective, interpretative qualification. I find the Eocene paleogeography quite different from todays, given that the Tethys Ocean was still open. If you want to point to

the similarity with the present-day, you could state that the majority of the continents were located in the northern hemisphere.

We have used the phrase "broadly similar" in the sense that the continental configuration is instantly recognisable, unlike for example, the Triassic period, with a single supercontinent just starting to break up into those that we're familiar with today.  We will amend this paragraph:

> The arrangement of the continents and oceans in the Early Eocene was broadly similar to that of the present, with the Earth's land mass divided into the same major continents, and with most of the land mass in the northern hemisphere.  India had not yet collided with the Eurasian continent, and the closure of the Tethys Ocean was not yet complete.  Such tectonic movements may have effected some changes to the climate system.  In particular, the configuration of ocean gateways strongly influences modes of ocean circulation, and hence affects energy transport throughout the climate system (Lunt et al., 2016; Sijp et al., 2014).

Page 4, line 10 and many other occurrences: "dominant periods of 100 kyr and 405 kyr". In an eccentricity power spectrum there are 4 peaks around 100 kyr, but only a single one at 405 kyr. Therefore, I would suggest the above notation.

We note that there are multiple peaks in the power spectra for eccentricity, equivalent to a single peak with a period of approximately 100 ka, together with an isolated peak for eccentricity with a period of 405 ka.  There are similar clusters of peaks around 40 ka for obliquity, and around 20 ka for precession.  We will amend the text to use the approximation symbol '~' in respect of the obliquity, precession and 100 ka eccentricity cycles, but not in respect of the 405 ka eccentricity cycle:

> The main oscillations are the eccentricity of the Earth's orbit around the Sun, with periods of ~100 ka and 405 ka, the obliquity or tilt of the Earth's axis of rotation, with a period of ~40 ka, and precession, the relative timing between perihelion and the seasons, with a period of ~20 ka (Berger et al., 1993).

Page 4, line 16: Jacques Laskar does not calculate time scales. He calculates astronomical solutions.

We will replace "astronomical time scale" with "astronomical solution".

Page 5: Why is Section 3 not a subsection of Section 4 "Methods"?

This section will be moved to the Methods as suggested by both reviewers.

Page 5, line 3: What is "T21"?

We will amend this sentence to clarify that T21 denotes the resolution obtained through spectral modelling:

> We apply the model at a spectral T21 atmospheric resolution, which corresponds to a triangular truncation applied at wave number 21 and a horizontal resolution of 5.625°, with 10 layers, and a matching ocean grid with 32 depth levels.

Page 6, lines 9-11: An injection of carbon into the atmosphere is measured in tons of C, whereas the concentration of CO2 in the atmosphere is measured in ppm. These are thus two different things, with two different units. You have to rephrase this sentence to correct for that.
We will amend the sentence as follows:

> Although the maximum mass of $CO_2$ injected into the atmosphere during CIEs, and in particular the PETM, remains uncertain, there is broad agreement that the atmospheric concentration of $CO_2$ did not exceed 3000 ppm (e.g. Gehler et al., 2016), and that it did not fall below the pre-industrial level of 280 ppm at any time during the early Eocene.

Page 6, lines 13-16: It's not immediately clear to me how knowledge on the phase relationship between carbon isotope excursions and the astronomical parameters would influence the experimental design of your study. If you would know these phase relationships, would you then have designed your experiments differently?

If these relationships were known, we would have been able to concentrate our investigation on combinations of the orbital forcing parameters of particular interest, i.e. those considered to be important in respect of the CIEs. We will amend this paragraph:

> Since the absolute astronomical time scale for the early Eocene has an uncertainty which is greater than the periods of the obliquity and precession cycles, and there remains disagreement as to which phases of the eccentricity cycles are related to CIEs, there are no combinations of the orbital forcing parameters which can be known a priori to be of greater importance in their effects on the Eocene climate in general, and on their contributions to the initiation, duration and termination of the CIEs in particular. We therefore select values of orbital parameters independently, and from the full range of each parameter's variation during the early Eocene.

Page 6, line 26: What do you mean with "quasi-steady state"?

We will add the phrase "a spin-up period of" to clarify that the "quasi-steady" state is the state of approximate equilibration of the model after the model has run for long enough such that the initial conditions have been 'forgotten'.

Page 7, line 7-8: The atmospheric circulation patterns during the Eocene were most definitely different from those in the modern world. I think you can remove the "are likely to".

We agree, and we will replace "are likely to have differed" with "will have differed".

Page 7 line 27: Spell out SVD
We will amend this sentence to accommodate suggestions from both reviewers:

> We perform a singular value decomposition to identify the PCs and empirical orthogonal functions (EOFs) of temperature and precipitation fields in the full ensemble.

Page 8 line 9: Please provide the appropriate references where these criteria are defined.

We will provide the appropriate references for the Akaike information criterion (Akaike, 1974), and Bayes information criterion (Schwarz, 1978), and since these are of a highly technical nature, we will add a reference to a much cited textbook on model selection:

> Burnham and Anderson (2003) provide a detailed discussion of the application of information criteria in model selection.

Page 8 lines 23-24: The Figure 3 that you are referring to, only contains global annual mean SST's, not the Arctic winter SST's you are discussing.

We will amend the text:

> We note that the Arctic winter median air temperature is below freezing over both land and sea in the PLASIM-GENIE ensemble, (see Fig 3) and the Arctic does not remain ice-free throughout the year in any of the 50 simulations in our study.

Page 9, line 1: It is unclear to me what exactly you mean with "parametric uncertainty"

We will amend the text for clarification:

> Quantification of model-related uncertainty is beyond the scope of the present study.

Page 10, line 17: JJA instead of JJF.

We will correct this error.

Page 10, line 15: Shouldn't this be Table 4?

We will correct this error – it will now be Table 5, following earlier insertion of an additional table.

The paper contains a few important shortcomings when it comes to appropriately referencing pre-existing work.
For example, the authors do not refer to the Deep-time Model Intercomparison Project (Deep-MIP, Lunt et al., 2017, gmd-10-889-2017). The authors do not frame their study within that project, nor do they differentiate their study from that project. A statement on this topic is indiscernible, given that both this study and the Deep-MIP project explicitly focus on simulating (early) Eocene warm climates and that both are using the same paleogeographic configuration from Herold et al. (2014).

This paper was at the final stages of preparation when Lunt et al. (2017) was published online (on 23 February 2017).  We are pleased to note that their recommended palaeogeography is that of Herold et al. (2014) which we have used as the basis for the palaeogeography in our study.  We will amend the first sentence in the description of our model configuration:

> This study was designed before Lunt et al. (2017) presented their 'DeepMIP' guidelines for model simulations of the latest Paleocene and early Eocene.  However, our palaeogeography is based on the high-resolution digital reconstruction of the early Eocene published by Herold et al. (2014), and which Lunt et al. (2017) recommended should be used as the standard for all palaeoclimate simulations within the DeepMIP framework.  We have used the dataset of Herold et al. (2014) as an initial configuration for the tectonic layout, topography and bathymetric boundary conditions in our study.

We will also add a comment on the solar constant:

> We note that Lunt et al. (2017) have recommended that a modern value of 1361.0 W m$^{-2}$ should be applied to studies within the DeepMIP framework, in order to facilitate comparison between simulations with modern and pre-industrial levels of $CO_2$, and to offset the absence of elevated levels of $CH_4$.

The authors refer to Bounceur et al. (2015), who applied a "similar emulator approach" (p. 8 line 13). First of all, I am unsure whether that statement is technically correct. Secondly, this reference is missing from the reference list.

We will ensure that Bounceur et al. (2015) are included in the reference list, and we will amend the text to clarify our comparison with their approach:

> Our emulator approach uses linear regression, rather than a Gaussian process, and is therefore simpler than the methods applied by Bounceur et al. (2015) in a study of the response of the climate-vegetation system in interglacial conditions to astronomical forcing, and by Araya-Melo et al. (2015) in their study of the Indian monsoon in the Pleistocene.

On page 4, line 28, the authors give credit to Ruddiman (2006, cp-2-43-2006) for noting "a relationship between obliquity and the extent of northern ice sheets". First of all, this is a Pleistocene-focused paper, of which I don't really see the relevance when discussing orbital configurations during the Eocene and possible influence on climate.  Moreover, the relationship between obliquity-induced minima in NH summer insolation and ice age cycles was already suggested by Milutin Milankovitch in 1941.

We agree that this is misleading, and adds little to the paper.  We will delete it.

**References**

Bounceur, Nabila, Michel Crucifix, and R. D. Wilkinson. "Global sensitivity analysis of the climate-vegetation system to astronomical forcing: an emulator-based approach." Earth System Dynamics 6.1 (2015): 205.

Herold, N., Buzan, J., Seton, M., Goldner, A., Green, J. A. M., Müller, R. D., Markwick, P., and Huber, M.: A suite of early Eocene (â´Lij 55 Ma) climate model boundary conditions, Geosci. Model Dev., 7, 2077–2090, doi:10.5194/gmd-7-2077-2014, 2014.

Tuenter, Erik, et al. "The response of the African summer monsoon to remote and local forcing due to precession and obliquity." Global and Planetary Change 36.4 (2003): 219-235.

Zeebe, Richard E., Andy Ridgwell, and James C. Zachos. "Anthropogenic carbon release rate unprecedented during the past 66 million years." Nature Geoscience 9.4 (2016): 325-329.

Zeebe, R. E., T. Westerhold, K. Littler, and J. C. Zachos (2017), Orbital forcing of the Paleocene and Eocene carbon cycle, Paleoceanography, 32, doi:10.1002/2016PA003054

Our additional references:

Akaike, H.: A new look at the statistical model identification, IEEE transactions on automatic control, 19, 716-723, 1974

Burnham, K. P. and Anderson, D. R.: Model selection and multimodel inference: a practical information-theoretic approach, Springer, New York, 2003.

Schwarz, G.: Estimating the dimension of a model, The annals of statistics, 6, 461-464, 1978

Thomas, D. J., Zachos, J. C., Bralower, T. J., Thomas, E., and Bohaty, S.: Warming the fuel for the fire: Evidence for the thermal dissociation of methane hydrate during the Paleocene-Eocene thermal maximum, Geology, 30, 1067-1070, 2002

---

## Author Response (AR1)

**Keery et al., Sensitivity of the Eocene Climate to CO2 and Orbital Variability**

Responses to Reviewers

We have amended the manuscript to address the issues raised by the reviewers. We have included comments on the limitations and advantages of models of intermediate complexity in the Methods and Summary sections. We have expanded our discussion of the responses of the monsoons, including the American monsoon, to orbital cycle, and clarified comparisons or our results with temperature proxy measurements.

Details are provided below in our responses to each of the two reviewers, with reviewers' comments in black, and our responses in red.

In addition to our responses to the reveiwers' comments, we have recalculated main effects and total effects without using an approximation in the calculation of variances, resulting in only very minor numerical differences, and no change to our main findings. We have prepared amended versions of Table 4 and Figures 10 and 11, and made very minor changes to the text, where appropriate.

We have included calculations of climate sensitivity for a doubling of atmospheric $CO_2$, added a new figure to illustrate this, and amended the Results section to include our findings with respect to a dependency of climate sensitivity on low or high states of $CO_2$ concentration:

> Figure 7 shows the relationship between $CO_2$ (plotted on a logarithmic scale), and MAT, with an abrupt change of gradient clearly visible at a $CO_2$ concentration of 1000 ppm. From the two gradients, we derive climate sensitivity values for a doubling of $CO_2$ concentration at $CO_2$ levels below 1000 ppm, and at $CO_2$ levels above 1000 ppm, of 4.36°C and 2.54°C respectively. We note that our modelled values of carbon in vegetation in the ENTS module remain low outside of the tropics at low $CO_2$ concentration, but as $CO_2$ concentration increases, land areas at higher latitudes reach maximum values of carbon in vegetation, with all land areas showing no further capacity for increased carbon in vegetation at an atmospheric concentration of ~1000 ppm. The increase in land vegetation cover, with corresponding reduction in albedo, acts as a positive feedback to rising temperature caused by increasing $CO_2$, but this feedback mechanism ceases to operate when all available land is at its maximum vegetation capacity, with a consequent reduction in the climate sensitivity.

and we have included the following text in the Summary:

> Our modelling results suggest that climate sensitivity is state dependent, with a value of 4.36°C in a low $CO_2$ state, and 2.54°C in a high $CO_2$ state, due to a positive feedback mechanism in which albedo reduces as vegetation increases to its maximum value when $CO_2$ concentration reaches 1000 ppm.

Responses to each of the reviewers, and the amended manuscript with markup, are included below.

**Keery et al., Sensitivity of the Eocene Climate to CO2 and Orbital Variability**

Response to M. Crucifix (Referee)

Referee comments in black

Author responses in red

We are very grateful for this thorough review.

Summary

Keery et al. present a sensitivity analysis of the Eocene climate to four factors: CO2 concentration, eccentricity, obliquity, and precession angle. They use, to this end, the PLASIM-GENIE model (details in their section 3) with suitable palaeogeography. The methodology relies on a 50-member hyper-cube sample of a 5-d space (one extra dummy variable was added), and linear modelling with a Information Criteria for model selection. Experiment output are summarised using fit-for-purpose summaries like "tropical-polar temperature difference" and monsoon indices, as well as principal components obtained from a singular value decomposition. The authors conclude on the importance of CO2 for global mean temperature, and of the orbital elements for the spatial distribution and regional weather systems such as monsoons.

Main comments

1. The paper is in the line of a number of recent studies attempting to estimate the relative sensitivity of the climate system to CO2 and orbital forcing, using a methodology founded on ensemble of experiments. This includes, in addition to the Holden et al. (2015) and Bounceur et al. (2015) cited, Araya-Melo et al. (2015) and Lord et al. (2017). Keery et al. is the only article to focus on the Eocene, which makes it an original contribution. It also uses a much simpler methodology than Araya-Melo et al. (2015), Bounceur et al. (2015), and Lord et al. (2017) because it uses linear regression instead of a Gaussian process emulator. In fact, the authors reference to the word "emulator" is slightly unusual because emulation is, in the climate literature, often used to designate statistical meta-modelling with a focus on uncertainty quantification. Claiming (p. 8) that a "similar emulator approach has been applied by Bounceur et al. 2015" is therefore somewhat misleading. Bounceur et al. and Araya-Melo et al. applied the developments of Oakley and O'Hagan (2004) with, in the case of Bounceur, the additional complication of the PCA emulator.

We agree that the comparison of our emulator to the emulators developed by Araya-Melo et al. (2015) and Bounceur et al. (2015) was misleading, and we have amended this section:

> Our emulator approach uses linear regression, rather than a Gaussian process (GP), and is therefore simpler than the methods applied by Bounceur et al. (2015) in a study of the response of the climate-vegetation system in interglacial conditions to astronomical forcing, and by Araya-Melo et al. (2015) in their study of the Indian monsoon in the Pleistocene.

In spite of its simplicity, we are confident that our approach may be correctly described as an emulator, as it fulfills the criteria described by O'Hagan (2006), and cited by Araya-Melo et al. (2015):

- it is derived from a small number of model runs filling the entire multidimensional input space
- once the emulator is built, it is not necessary to perform any additional runs with the model

In passing, Araya-Melo et Lord used HadCM3 which shows that ensemble-based sensitivity analysis to orbital forcing is doable with GCMs (this qualifies the author's comment on line 15, p.2).

We have amended this paragraph to acknowledge recent ensemble studies using GCMs:

Climate simulations with high temporal and spatial resolution can be obtained from General Circulation Models (GCMs), but the requirement of GCMs for powerful computers and long run-times makes them difficult to deploy for large ensembles of model simulations and restricts their ability to investigate the large uncertainties in forcings and model parameterisations. Such ensembles are more practical with more heavily parameterised and hence more computationally efficient Earth system Models of Intermediate Complexity (EMICs), (Weber, 2010), although we note that Araya-Melo et al. (2015) and Lord et al. (2017) have deployed the GCM HadCM3 in ensemble-based studies of orbital forcing effects on climates of the Pleistocene and late Pliocene respectively.

Of course, the fact that other authors have adopted a more sophisticated methodology invalidates by no means the approach used by Keery et al.: there may be no need to use a sledgehammer to crack a nut. It remains that the methodological set up used here is a step backwards compared to recent studies, and this arguably requires some justification. How much do we lose with the linearity assumption, and which impact does it have on the uncertainties of the quantification of main effects? (see comment 3. more specifically on main effects).

As we have noted in our methods section, we have demonstrated that the linear models can be used to emulate PC scores with very high correlations to the PC scores derived directly through SVD, with examples from temperature and precipitation shown in Table 3. We can therefore be confident that main effects derived from the linear models are robust. We have amended the text:

Unlike linear models, GP models are intrinsically stochastic and give a more accurate quantification of their own error in emulating the input data. However, GP models can become computationally demanding in high dimensional space, and their results can be more difficult to interpret.

2. Experiment design. The authors do not say much about the ensemble design, except that this is a latin hypercube. There are many ways to do a latin hypercube, and it usually involves additional constraints.

We have added a detailed description of the method used to generate the latin hypercube in an appendix, include forcing factor values for the full ensemble in a new Table, and we have amended the main text:

The present study has been designed to facilitate direct comparison between the results for specific ensemble members and their direct counterparts in a future study using the EMIC model GENIE-1 (Edwards and Marsh, 2005), which will include additional forcing parameters not used by this PLASIM-GENIE study. We have applied an iterative method to generate a pair of corresponding hypercubes with five and eleven dimensions for the PLASIM-GENIE and GENIE-1 studies respectively, in which the minimum Euclidean distance between any two points is maximised, and linear correlation between any two parameters is minimised. Details of the steps taken to generate the hypercubes are provided in Appendix A. The absolute value of the r correlation coefficient does not exceed 0.1 for any pair of input (forcing and dummy) parameters. Uniform ranges for each of the forcing parameters and the dummy parameter are shown in Table 1, and the values applied in all 50 PLASIM-GENIE ensemble members are shown in Table 2.

In fact this experiment design raises some doubts. For example, why are some secondary structures (periodic up and downs) apparent in the response to obliquity, Figure 5, middle column? Is this just a subjective visual impression?

We have created an additional plot of the two forcing factors obliquity and $CO_2$, for discussion, but not for inclusion in the paper, and this shows a very similar pattern to the obliquity-MAT subplot in Figure 5, with corresponding clusters and the same slight impression of periodicity. We can therefore be confident that the apparent periodicity noted by the reviewer in the model output is an artefact of randomly generated structure in the model input.

[Figure]

Figure R1   Obliquity plotted against $CO_2$.

One potentially problematic element is the definition of the sampled astronomical space. It seems that latin hypercube sampling is made on axes along e, ω (longitude of perihelion) and ε. If this is what the authors have been doing then this is non-physical. We know that the astronomical forcing generates effects through seasonal and daily insolation, which are very well approximated by linear functions of e sinω (which the authors call the precession index on Fig. 6) and e cosω. This is the reason why several authors have chosen to sample the astronomical space following the axes e sinω and e cosω and regress against these components. Presumably the regression analysis by Keery is indeed done against these indices but the text is not always clear. Lines 1-2 p. 8. rather suggest that the explanatory variables where sinω and cosω (instead of their multiplication by e) and the lines 4-5 p. 11 are quite confusing. Hopefully the choice of regression variables is mainly matter of text clarification, but the design of the latin hypercube may have a more fundamental problem.

We have indeed constructed our hypercube by sampling independently on $e$, $\omega$ (longitude of perihelion) and $\varepsilon$, but we do not agree that this is non-physical, as there are no combinations of these parameters which can be excluded for the early Eocene period. If we have ignored any information which would imply that some combinations are less likely to have occurred than others (we are not aware of any), then this would only result in a minor reduction in the efficiency with which we fill our state space. We note that precessional effects are well approximated by $e\sin\omega$ and $e\cos\omega$, and that several authors have chosen to sample and regress against these components, but we have chosen not to take this approach, as it would not allow any climatic effects of eccentricity which may exist independently of precession to be identified. We have amended our description of the forcing factors:

> In order to investigate the sensitivity of the Eocene climate to variation in atmospheric $CO_2$ and orbital parameters, we have constructed an ensemble of 50 model configurations, each with a unique set of forcing parameters comprising atmospheric $CO_2$, eccentricity ($e$), obliquity ($\varepsilon$) and precession ($\omega$), the angle on the Earth's orbit around the Sun between the moving vernal equinox and the longitude of perihelion (Berger et al., 1993). When $e$ is zero, the Earth's distance from the Sun is constant at all points on the orbit, so there is no precessional effect. The magnitude of precessional effects is controlled by $e$, while phase is controlled by $\omega$, so precessional effects are commonly described by the precession index given by $e\sin\omega$. The only orbital parameter which alters the total annual solar radiation received by the Earth is $e$, although the range of variation is very small. We include $e$ and $\omega$ as separate and independent forcing parameters, rather than combined as the precession index, or in the form $e\cos\omega$. This approach does not make the assumption that the only effect of eccentricity on the Earth's climate is through its effect on the amplitude of the precession cycle, but allows experimental results to be examined for effects of $e$ and $\omega$ either separately or in combination. An additional dummy parameter is included to test for possible overfitting of relationships between forcing parameters and model output fields.

We have also amended our description of our preparation of the forcing factors for linear modelling:

Values of the forcing parameters $CO_2$, $e$ and $\varepsilon$ (with its very small angular range considered to be approximately linear) were normalised to the range [-1, 1] and combined with $\sin\omega$ and $\cos\omega$ to form 50-element column vectors representing the forcing factors.

3. There may be some confusion about the meaning of the main effects. Saltelli does not use the phrase "first order" to mean linear approximation. In a case where only one factor would matter (be the relationship linear on not), the main and total effects would match (Saltelli et al. (2004), ch. 1 states clearly the definitions; or refer again to Oakley and O'Hagan (2004)). More generally, computing main and total effects is not trivial and always involves some approximations.More details on their computation would be welcome.

We have amended the text to provide more details on the computation of the main effects and total effects:

In order to analyse the results of each of our linear models, we apply the method described in detail by Holden et al. (2015) to derive the main effects (Oakley and O'Hagan, 2004), which provide a measure of the variation in the linear model output due to each of the terms (first order, second order and cross products), derived from their coefficients, and total effects (Homma and Saltelli, 1996), which separate the effect of each forcing parameter on the variation in the model output. Although the forcing factors are all scaled within the range [-1, 1], the trigonometrical precession terms are not uniformly distributed across this range. We have therefore computed the variances of the first order, second order and cross product terms directly for all parameters, rather than applying the respective approximations of $\frac{1}{3}$, $\frac{1}{9}$ and $\frac{4}{45}$, and we have applied these values as scaling factors in calculating the main effects and total effects.

4. Singular value decomposition is a great dimensionality reduction methodology, but how much is learned by analysing the behaviour of principal components separately is a more contentious subject. Identification of principal components can be fragile to some implementation details, such as, e.g. grid area weighting and experiment design, and the physical phenomena which give rise to climate variability need not be orthogonal. In fact physical modes may project poorly on the orthogonal vectors (Monahan and Fyfe, 2006). These caveats implicitly acknowledged by the authors (p. 11, ll. 20-21) but this state-of-affairs poses some questions about the emphasis on principal components in this article.

We have amended the text to acknowledge these caveats:

We perform a singular value decomposition to identify the PCs and empirical orthogonal functions (EOFs) of temperature and precipitation fields in the full ensemble, although we note that climate variability may not be due to physical processes which vary orthogonally, and identification of PCs can be influenced by aspects of the experimental design.

Minor (scientific) comments

• How Fig. 2 should be interpreted is not entirely clear since the ensemble was not explicitly designed so that the ensemble mean is an estimate of the Eocene climate mean.

Figures 2 and 4 are included to provide an illustrative summary of the spatial distribution and variation of temperature and precipitation in the full ensemble, without implying that the ensemble mean is an estimate of the Eocene climate mean. We have amended the text:

Analysis of the model results has focused on variation in surface air temperature and precipitation in both winter and summer in each hemisphere, although it should be noted that our experiment has not been designed such that mean values in our ensemble output represent direct estimates of the Eocene climate mean.

Minor (editorial) comments

• Introduce subtitle after section 2.

We have introduced the subtitle 'Climate of the Early Eocene'

• Material about cyclostratigraphy under section 2.1.2. may possibly be considered for shortening as slightly out of scope of the article. This said this is an interesting read.

We would prefer to retain the section on cyclostratigraphy in full, as we believe it provides important details which are relevant to our experimental design, particularly our selection of independent orbital values, and the separation of $e$ and $\omega$.

• PLASIM-GENIE does not need a specific section: it can fall under section 3.Methods.

This section has been moved to the Methods as suggested by both reviewers.

• p. 6 reference Gough (1981) is mistakenly repeated.

The duplicated reference has been be removed.

• p. 7, the sentence "We apply the linear algebraic tool SVD" sounds unnecessarily sophisticated. Why not "We perform a singular value decomposition to identifyprincipal components"

We have amended this sentence:

> We perform a singular value decomposition to identify the PCs and empirical orthogonal functions (EOFs) of temperature and precipitation fields in the full ensemble.

• p. 10, l. 27 : define the word "precession" precisely.

We have made amendments to the text to define precession ($\omega$), and the precession index ($e\sin\omega$). See our response to an earlier comment.

• p. 12, ll. 13-17 : introducing new results so close to the closing words is usually not encouraged.

We have deleted these results, as further analysis suggests it is difficult to draw any very useful conclusions from the extra experiment, and we have amended the text to include the reference to Anagnostou et al. (2016):

> If atmospheric $CO_2$ remained within a narrower range throughout the period, for example in the range 700 to 1800 ppm indicated for the early Eocene by Anagnostou et al. (2016) in a recent study using boron isotopes, then outside of short-lived hyperthermals, the relative influence of $CO_2$ and orbital inputs might have been more evenly balanced.

Digital material

• Relevant data of the Eocene runs (at least the summaries and experiment input data) could be provided.

We have included the values of forcing factors for the 50 member ensemble in a new Table.

Response to D. De Vleeschouwer (Referee)

Referee comments in black

Author responses in red

We are very grateful for this thorough review.

This paper reports on an ensemble of 50 Eocene climate-model simulations, each of which characterized by a different combination of eccentricity, obliquity, precession and atmospheric CO2 concentration. The climate model is the PLASIM-GENIE model, a new model of intermediate complexity, recently introduced by Holden et al. (2016). The study aims to summarize the ensemble of paleoclimate simulations by looking at what-they-call "simple metrics", principal component analysis and an emulator approach. This study provides a couple of interesting results. The first is the existence of a seaice-related threshold mechanism in the northern hemispheric high latitudes. From Figure 2 and 3, it seems that when a certain threshold in the extent of DJF-sea-ice is exceeded, temperatures (both sea-surface and maritime air temperatures) drop significantly. It would be interesting to read the author's opinion how this compares to the recent findings of modeling work by Zeebe et al. (2017), who found that "High-latitude mechanisms are unlikely drivers of orbitally paced changes in the late Paleocene-early Eocene". The interesting role of (seasonal) sea-ice in the climate system of the early Eocene aspect remains, however, rather underdeveloped in the present version of the paper.

In our discussion of Figs. 2 & 3 [page 9, line 10] we have stated: "The variation in TPTD across the ensemble thus appears to be essentially driven by the strength of snow and ice albedo feedback", and a little further on, in our discussion of Fig. 5 [page 9, line 22], in particular the plot of $CO_2$ v northern winter TPTD we have declared: "and it can also be seen that $CO_2$ strongly affects the northern TPTD in the winter, but not in the summer, when the combined influence of obliquity and precession index is discernible, suggesting that temperature proxies with seasonal bias may have a significant orbital imprint. The plot of atmospheric $CO_2$ against N. Winter TPTD shows a change in gradient at approximately 1000 ppm CO2 and 32°C. This may be related to the logarithmic dependence of radiative forcing on CO2 concentration, as well as the disappearance of ice above some threshold level, cf Fig. 3."

We have added the additional comment:

> A possible sea ice related threshold mechanism influencing both SST and maritime air temperature in high northern latitudes may be observed in Fig. 3, and this is strongly associated with the increase in northern winter TPTD at low $CO_2$ levels. Zeebe et al. (2017) have analysed a high resolution benthic isotope record covering the late Palaeocene - early Eocene, and have concluded that orbitally paced cycles are unlikely to have been driven by high latitude mechanisms, but our PLASIM-GENIE modelling suggests that while northern TPTD is not orbitally paced in the winter, being controlled by $CO_2$, it is orbitally paced in the summer, by a combination of obliquity and precession.

The second interesting aspect is the distinct response to precession of monsoonal precipitation and temperature in the different monsoonal systems (e.g. Figure 6). The description and discussion of these Eocene paleoclimate simulations is useful and perfectly fits the scope of the journal. The current version of the manuscript is, however, unsatisfactory for publication in Climate of the Past for the reasons listed below.

**Major Comments**

1. One of the major conclusions in the current version of the manuscript, is that 95The emulator approach adopted in this study allows for estimating the response of different aspects of the climate system (e.g. wet-season monsoonal precipitation) over the full input space. It would -for example- be interesting to see the response of precipitation and temperatures in the different monsoonal systems to astronomical forcing for specific pCO2 levels. This could be an elegant way to circumvent the disparity in time-scales between CO2 and orbital variability.

We have amended the subplots for obliquity and precession index in Figures 5 and 6 to denote the $CO_2$ level on a continuous colour scale. This approach gives a simple visual indication of which relationships between the astronomical forcing factors and the temperature and precipitation simple metrics are influenced by $CO_2$. Figure 6 also now includes an additional row of subplots for the American monsoon index.

We have applied emulators derived from linear modelling of the forcing factors and monsoon indices, to estimate values of each of the monsoon indices over the full range of precession ($\omega$), with fixed high eccentricity ($e$), for low and high values of $CO_2$, and low and high values of obliquity ($\varepsilon$).

We have made amendments to the abstract:

> The results demonstrate the importance of orbital variation as an agent of change in climates of the past, and we demonstrate that emulators derived from our modelling output can be used as rapid and efficient surrogates of the full complexity model, to provide estimates of early Eocene climate conditions from any set of forcing parameters.

and to the final paragraph of the introduction:

> By applying the linear modelling and emulation methods of Holden et al. (2015), we regress both the simple scalar metrics and the SVD reduced dimension model outputs onto the forcing parameters, and from the derived relationships, we infer main effects denoting the effect of each explanatory term in the linear model, and total effects denoting the effect of each forcing parameter, on the variation in the scalar metrics and on the temperature and precipitation output fields. We demonstrate that emulators derived in respect of tropical precipitation metrics can be used to estimate Eocene monsoonal responses to any combination of GHG and orbital forcing parameter values.

We have added new Figures 12, 13 and 14, plotting emulated values of the Asian, African and American monsoon indices.

We have added a paragraph to the Results section:

> We apply the linear models derived from the forcing factors and monsoon indices as emulators to estimate values of monsoon indices corresponding to the full range of precession ($\omega$), with eccentricity fixed at its high limit of 0.06, low and high values of $CO_2$ (300 ppm and 3000 ppm), and low and high values of obliquity (22.0° and 24.5°). Precession index ($e\sin\omega$) and emulated values of the Asian, African and American monsoon indices are plotted in Figures 12, 13 and 14 respectively. Relationships between the precession index and the monsoon indices which are visually suggested in Figure 6 are shown with clear structure in Figures 12, 13 and 14. In each of the monsoon areas, the increase in precipitation due to precession effects is more pronounced at high atmospheric concentration of $CO_2$, and also at high obliquity.

We have added a paragraph to the Summary;

> We have demonstrated that emulators derived from linear modelling of the PLASIM-GENIE ensemble results can be used as a rapid and efficient method of estimating climate conditions from any set of forcing parameters, without the need for further deployment of the EMIC.

2. The authors do not provide their 50-simulation experimental design. It is essential to have an overview of the parameter settings for each simulation that was run in the framework of this study. The details on the settings of the 50 simulations could be given either in the form of a Table, or in the form of a figure, or in both forms. For good examples, please check Figure 2 and Table 1 in Araya-Melo et al. (2015, cp-11-45-2015), Figure 2 and Table 2 in Lord et al. (2017, cp-2017-57), and Figure 1 in Bounceur et al. (2015, esd-6-205-2015).

We have included the values of the forcing factors and the dummy variable for the ensemble in a new table (Table 2).

We note that Araya-Melo et al. (2015) constrained their experiment to exclude non-physical combinations of $CO_2$ and sea ice, and their Figure 2 includes an informative subplot showing fairly strong inverse correlation between $CO_2$ and sea ice. In our study, however, we do not have a priori information with which to constrain any combinations of our forcing factors, each of which is sampled independently to maximise state space coverage and to minimise correlations between the forcing factors. We include in this response a new figure showing cross-plots and r coefficients of all of the forcing factors and the dummy parameter, which illustrate both the coverage of the state space, and the very low correlation between any of the factors. We do not consider that this figure, or a variation, could add significant information to that included in the text, which has been amended to include the statement:

> The absolute value of the correlation coefficient r does not exceed 0.1 for any pair of input (forcing and dummy) parameters.

[Figure]

Figure R2  Correlation plots and r coefficients between all forcing factors.

3. From Figure 6, it is very clear that precession has an important influence on the Asian Monsoon intensity, with higher rainfall when the index is minimum (i.e. Earth in perihelion during JJA, maximum northern hemisphere summer insolation). However, if I interpret PC2 in JJA temperature and PC2 in JJA precipitation correctly (Table 5 and Figures 7 and 8), it seems that a precession-driven increase in monsoonal rainfall coincides with a decrease in JJA temperature in the Asian Monsoon region. Such a decrease in temperature is remarkable, given that it occurs when northern hemisphere JJA insolation is maximum. This observation can either be explained by the consumption of incoming solar radiation as latent heat, or by a negative influence of the increased cloud cover on the radiation balance. Indeed, the reflective character of clouds contributes to the planetary albedo. In the revised version of the manuscript, I would like to read more discussion of paleoclimate mechanisms like this one.

This temperature decrease is indeed observed in the model results for the Asian monsoon. We have augmented the text to describe this effect more clearly:

> An increase in the second PC scores for JJA precipitation in the Asian monsoon region (Fig. 9) corresponds to a decrease in the second PC scores for JJA temperature (Fig. 8), and as already noted, the second PC scores for both temperature and precipitation in JJA are strongly correlated to the precession index. This temperature reduction during the Asian monsoon was also observed by Holden et al. (2014), and attributed to a reduction in incoming solar radiation associated with increased cloud cover and surface evaporation.

4. Page 7, lines 23-25 and Figure 6: When I was first interpreting Figure 6, I was confused by the fact that the Asian Monsoon and the African monsoon seemed to respond to precession in the same way, despite the fact that they are located on opposite sides of the equator. It took me quite a while to realize that both monsoonal systems are responding to precession in the expected way: with intensified wet-season precipitation in the Asian Monsoon system when the Earth reaches perihelion in JJA (negative precession index), and intensified wet-season precipitation in the African Monsoon system when the Earth reaches perihelion in DFF (positive precession index). I only understood this after reading lines 23-25 (page 7) several times. Indeed, the authors define their monsoon-related "simple scalar metric" by the difference in rainfall in DJF and JJA, regardless of whether DJF is the wet or the dry season in the monsoonal system considered. This also explains why the panel of Figure 6 that is related to the African Monsoon shows negative values, whereas the panel that is related to the Asian Monsoon exhibits positive values. I would strongly advise the authors to think about ways to illustrate the monsoonal response to precession in a more intuitive way. Maybe the paper by Tuenter et al (2003) could provide some inspiration as to how to best present the response of a summer monsoon to precessional (and obliquity?) forcing. Also, why is the South American monsoon system missing from Figure 6?

We have amended our monsoon indices so that each is now derived by subtracting winter precipitation from summer precipitation, as suggested. Figure 6 has been altered accordingly, and now also includes a row for the American monsoon index, an entry for which will be added to the table of total effects of forcing parameters on simple scalar metrics (presently Table 3; will be Table 4).

We have amended the text to reflect the changes to the monsoon indices:

> In this study, we derive simple scalar metrics to denote indices for monsoons for Asia, Africa and South America by subtracting winter rainfall from summer rainfall, for defined geographical regions, denoted on Fig. 1, and selected for their similarity to monsoonal regions in the modern continental configuration.

We have amended our comments on Figures 5 and 6 in the Results section, following addition of the American monsoon index, and the use of colour in these Figures:

> In Figs. 5 and 6, $CO_2$, obliquity ($\varepsilon$) and precession index ($e\sin\omega$) are plotted against MAT, northern seasonality, northern winter TPTD and northern summer TPTD (Fig. 5), and southern winter polar OLC, northern winter polar OLC, Asian monsoon index, African monsoon index and American monsoon index (Fig. 6). Subplots for obliquity and precession index in Figures 5 and 6 denote the $CO_2$ level on a continuous colour scale.

and we have added the comment:

> The American monsoon index is fairly strongly correlated with the precession index at high levels of $CO_2$, and negatively correlated with $CO_2$ at low levels of $CO_2$.

We note that the study by Tuenter et al. (2003) included six experimental setups, with each one comprising either maximum or minimum values of obliquity, and maximum, minimum or zero values of precession. They were therefore able to illustrate their results in the form of spatial patterns of the differences in output values for pairs of experiments with contrasting values of one or both forcing factors. This approach is not appropriate for our 50 member ensemble, with uncorrelated forcing factor values, in which no pairs of experiments can be identified for this type of comparison.

**Additional comments and recommendations**

Abstract line 5 and p. 2 lines 1-3: I would recommend being a little bit more conservative on the possible analogy between the PETM and the ongoing anthropogenic disturbance of the global carbon cycle. Also cite Zeebe et al. (2016, Nature Geoscience) here.

We have amended the text to clarify the importance of the PETM, particularly its importance as the closest, if not perfect, analogue to anthropogenic climate change, and to include a citation of Zeebe et al (2016):

Since the PETM is the most recent period in Earth's history for which estimated atmospheric GHG concentrations are similar in magnitude to those of the present-day, and expected to arise from fossil fuel burning, the PETM may provide a valuable analogue for anthropogenic climate change (e.g. McInerney and Wing, 2011; Zeebe et al., 2016; Zeebe and Zachos, 2013).

Abstract: The abstract reads too technical and vague. I find the following sentence particularly vague: "Two dimensional model output fields are reduced to scalar values through simple summarizing algorithms and by singular value decomposition." The reader gets very little information from this sentence. I would recommend rewriting the abstract, making it more results-oriented.

We have deleted this sentence, and made amendments to the abstract to make it less vague, with more focus on the results, including our additional work on climate sensitivity, and using the emulators.

Page 2, line 30: suggestion: "The Earth resided in a greenhouse state"

We don't understand the reason behind this suggestion. Our intention was to emphasise that the greenhouse state had been continuous since the early Cretaceous, so we will leave the sentence unchanged.

Page 3, line 4: What do you mean with "high levels of radiative forcing"? Only eccentricity influences the total amount of solar energy received by the Earth: : : but the amplitude of that variability is only 0.15

Huber & Caballero (2011) used $CO_2$ as a proxy for all changes to incoming and outgoing radiation. They commented "We have not addressed whether the enhanced radiative forcing was due to $pCO_2$, methane, other greenhouse gases, novel cloud feedbacks, or other "missing" factors. We have also not established whether large forcing is actually necessary, the alternative being high values of climate sensitivity as in the study of Heinemann et al. (2009) and only moderate increases in forcing."

We have amended the text to clarify this:

Huber and Caballero (2011), hereafter HC11, have demonstrated that with sufficiently high levels of $CO_2$ (as a proxy for all forms of radiative forcing), climate models can generate global air temperature distributions in broad agreement with the proxy temperature measurements.

Page 2, line 9: Either you provide the reader with information on which kind of evidence exists. Or you rewrite like: "During the PETM, the emission of organic carbon was initially in the form of methane, which later oxidized to CO2".

We have amended the text to give brief details of the evidence, and we will include an additional citation:

There is some evidence from analysis and modelling of the timing and duration of variations in $\delta^{13}C$ and $\delta^{13}O$ observed in nannoplankton fossils that some of the GHG emissions were initially in the form of $CH_4$ (Dickens, 2011; Lunt et al., 2011; Thomas et al., 2002), which is rapidly oxidised in the atmosphere to $CO_2$.

Page 2, line 23: "broadly similar" is quite a subjective, interpretative qualification. I find the Eocene paleogeography quite different from todays, given that the Tethys Ocean was still open. If you want to point to the similarity with the present-day, you could state that the majority of the continents were located in the northern hemisphere.

We have used the phrase "broadly similar" in the sense that the continental configuration is instantly recognisable, unlike for example, the Triassic period, with a single supercontinent just starting to break up into those that we're familiar with today. We have amended this paragraph:

> The arrangement of the continents and oceans in the Early Eocene was broadly similar to that of the present, with the Earth's land mass divided into the same major continents, and with most of the land mass in the northern hemisphere. India had not yet collided with the Eurasian continent, and the closure of the Tethys Ocean was not yet complete. Such tectonic movements may have effected some changes to the climate system. In particular, the configuration of ocean gateways strongly influences modes of ocean circulation, and hence affects energy transport throughout the climate system (Lunt et al., 2016; Sijp et al., 2014).

Page 4, line 10 and many other occurrences: "dominant periods of 100 kyr and 405 kyr". In an eccentricity power spectrum there are 4 peaks around 100 kyr, but only a single one at 405 kyr. Therefore, I would suggest the above notation.

We note that there are multiple peaks in the power spectra for eccentricity, equivalent to a single peak with a period of approximately 100 ka, together with an isolated peak for eccentricity with a period of 405 ka. There are similar clusters of peaks around 40 ka for obliquity, and around 20 ka for precession. We have amended the text to use the approximation symbol '~' in respect of the obliquity, precession and 100 ka eccentricity cycles, but not in respect of the 405 ka eccentricity cycle:

> The main oscillations are the eccentricity of the Earth's orbit around the Sun, with periods of ~100 ka and 405 ka, the obliquity or tilt of the Earth's axis of rotation, with a period of ~40 ka, and precession, the relative timing between perihelion and the seasons, with a period of ~20 ka (Berger et al., 1993).

Page 4, line 16: Jacques Laskar does not calculate time scales. He calculates astronomical solutions.

We have replaced "astronomical time scale" with "astronomical solution".

Page 5: Why is Section 3 not a subsection of Section 4 "Methods"?

This section has been moved to the Methods as suggested by both reviewers.

Page 5, line 3: What is "T21"?

We have amended this sentence to clarify that T21 denotes the resolution obtained through spectral modelling:

> We apply the model at a spectral T21 atmospheric resolution, which corresponds to a triangular truncation applied at wave number 21 and a horizontal resolution of 5.625°, with 10 layers, and a matching ocean grid with 32 depth levels.

Page 6, lines 9-11: An injection of carbon into the atmosphere is measured in tons of C, whereas the concentration of CO2 in the atmosphere is measured in ppm. These are thus two different things, with two different units. You have to rephrase this sentence to correct for that.

We have amended the sentence as follows:

> Although the maximum mass of $CO_2$ injected into the atmosphere during CIEs, and in particular the PETM, remains uncertain, there is broad agreement that the atmospheric concentration of $CO_2$ did not exceed 3000 ppm (e.g. Gehler et al., 2016), and that it did not fall below the pre-industrial level of 280 ppm at any time during the early Eocene.

Page 6, lines 13-16: It's not immediately clear to me how knowledge on the phase relationship between carbon isotope excursions and the astronomical parameters would influence the experimental design of your study. If you would know these phase relationships, would you then have designed your experiments differently?

If these relationships were known, we would have been able to concentrate our investigation on combinations of the orbital forcing parameters of particular interest, i.e. those considered to be important in respect of the CIEs. We have amended this paragraph:

> Since the absolute astronomical time scale for the early Eocene has an uncertainty which is greater than the periods of the obliquity and precession cycles, and there remains disagreement as to which phases of the eccentricity cycles are related to CIEs, there are no combinations of the orbital forcing parameters which can be known a priori to be of greater importance in their effects on the Eocene climate in general, and on their contributions to the initiation, duration and termination of the CIEs in particular. We therefore select values of orbital parameters independently, and from the full range of each parameter's variation during the early Eocene.

Page 6, line 26: What do you mean with "quasi-steady state"?

We have added the phrase "a spin-up period of" to clarify that the "quasi-steady" state is the state of approximate equilibrium of the model after the model has run for long enough such that the initial conditions have been 'forgotten'.

Page 7, line 7-8: The atmospheric circulation patterns during the Eocene were most definitely different from those in the modern world. I think you can remove the "are likely to".

We agree, and we have replaced "are likely to have differed" with "will have differed".

Page 7 line 27: Spell out SVD
We have amended this sentence to accommodate suggestions from both reviewers:

> We perform a singular value decomposition to identify the PCs and empirical orthogonal functions (EOFs) of temperature and precipitation fields in the full ensemble

Page 8 line 9: Please provide the appropriate references where these criteria are defined.

We have provided the appropriate references for the Akaike information criterion (Akaike, 1974), and Bayes information criterion (Schwarz, 1978), and since these are of a highly technical nature, we have added a reference to a much cited textbook on model selection:

> Burnham and Anderson (2003) provide a detailed discussion of the application of information criteria in model selection.

Page 8 lines 23-24: The Figure 3 that you are referring to, only contains global annual mean SST's, not the Arctic winter SST's you are discussing.

We have amended the text:

> We note that the Arctic winter median air temperature is below freezing over both land and sea in the PLASIM-GENIE ensemble, (see Fig 3) and the Arctic does not remain ice-free throughout the year in any of the 50 simulations in our study.

Page 9, line 1: It is unclear to me what exactly you mean with "parametric uncertainty"

We have amended the text for clarification:

> Quantification of model-related uncertainty is beyond the scope of the present study.

Page 10, line 17: JJA instead of JJF.

We have corrected this error.

Page 10, line 15: Shouldn't this be Table 4?

We have corrected this error – it is now Table 5, following earlier insertion of an additional table.

The paper contains a few important shortcomings when it comes to appropriately referencing pre-existing work.

For example, the authors do not refer to the Deep-time Model Intercomparison Project (Deep-MIP, Lunt et al., 2017, gmd-10-889-2017). The authors do not frame their study within that project, nor do they differentiate their study from that project. A statement on this topic is indiscernible, given that both this study and the Deep-MIP project explicitly focus on simulating (early) Eocene warm climates and that both are using the same paleogeographic configuration from Herold et al. (2014).

This paper was at the final stages of preparation when Lunt et al. (2017) was published online (on 23 February 2017). We are pleased to note that their recommended palaeogeography is that of Herold et al. (2014) which we have used as the basis for the palaeogeography in our study. We have amended the first sentence in the description of our model configuration:

> This study was designed before Lunt et al. (2017) presented their 'DeepMIP' guidelines for model simulations of the latest Paleocene and early Eocene. However, our palaeogeography is based on the high-resolution digital reconstruction of the early Eocene published by Herold et al. (2014), and which Lunt et al. (2017) recommended should be used as the standard for all palaeoclimate simulations within the DeepMIP framework. We have used the dataset of Herold et al. (2014) as an initial configuration for the tectonic layout, topography and bathymetric boundary conditions in our study.

We have also added a comment on the solar constant:

> We note that Lunt et al. (2017) have recommended that a modern value of 1361.0 W m$^{-2}$ should be applied to studies within the DeepMIP framework, in order to facilitate comparison between simulations with modern and pre-industrial levels of $CO_2$, and to offset the absence of elevated levels of $CH_4$.

The authors refer to Bounceur et al. (2015), who applied a "similar emulator approach" (p. 8 line 13). First of all, I am unsure whether that statement is technically correct. Secondly, this reference is missing from the reference list.

We have ensured that Bounceur et al. (2015) are included in the reference list, and we have amended the text to clarify our comparison with their approach:

> Our emulator approach uses linear regression, rather than a Gaussian process (GP), and is therefore simpler than the methods applied by Bounceur et al. (2015) in a study of the response of the climate-vegetation system in interglacial conditions to astronomical forcing, and by Araya-Melo et al. (2015) in their study of the Indian monsoon in the Pleistocene.

On page 4, line 28, the authors give credit to Ruddiman (2006, cp-2-43-2006) for noting "a relationship between obliquity and the extent of northern ice sheets". First of all, this is a Pleistocene-focused paper, of which I don't really see the relevance when discussing orbital configurations during the Eocene and possible influence on climate. Moreover, the relationship between obliquity-induced minima in NH summer insolation and ice age cycles was already suggested by Milutin Milankovitch in 1941.

We agree that this is misleading, and adds little to the paper. We have deleted it.

Our additional references:

[revised manuscript text omitted]

---

## Referee Report (RR1)

**Review "Sensitivity of the Eocene Climate to CO2 and orbital variability", by John S. Keery et al. (2017, Climate of the Past, cp-2017-60). Second Round.**

The authors satisfactory answered my major concerns, raised in my review of the first version of this manuscript. I particularly appreciate the way the authors tackled the disparity in time-scales between $CO_2$ and orbital variability, for example by adding the color scale to Figs. 5 and 6, and by integrating new Figures 12, 13 and 14 into the revised version of the manuscript.

However, I have two small questions regarding the latter three Figures.
1. In the text, the authors write "*In each of the monsoon areas, the increase in precipitation … is more pronounced at high atmospheric concentration of CO2*". But, according to Figure 14, the American monsoon is "*wettest*" under low $CO_2$ forcing. Is this a mistake in color-coding, or does the American monsoon system really respond differently to CO2 forcing? If the latter is true, this should be discussed in much more detail.
2. Figures 12, 13, 14 all show some degree of non-stationarity: the monsoonal index seems to depend not only on the precession index, but also on whether the precession index is increasing or decreasing. In other words, the emulated values of the monsoon index do not represent a line, but an ellipse. This is a very interesting result, but I do not fully understand how the authors came to this result, and I would like to read more about these values were obtained. Please discuss which limb of the ellipse represents increasing precession index (and vice versa), and please discuss why the American monsoon system seems to be more "non-stationary", compared to the African monsoon system.

Once the authors will have addressed these points of discussion regarding Figs. 12 – 14, I consider this manuscript suitable for publication in *Climate of the Past*.

**Technical corrections**
> **Page 3, line 18:** $^{18}O$ instead of $^{13}O$
> **Page 5 and throughout the paper:** There are a lot of different opinions on how to write 1000 years in a geologic scientific context. However, the "k" shown in upper case is an incorrect usage. I prefer the use of "kyr" for durations and relative time, and "ka" for thousand years ago (i.e. absolute time). See http://www.ldeo.columbia.edu/~ncb/Selected_Articles_all_files/25_Stratigraphy.6.100.pdf
> **Page 4, lines 25 – 30:** I still do not understand why a discussion of cyclostratigraphic tuning techniques is relevant for this work.
> **Table 6**: Draw horizontal lines under the PC3 rows, so to visually separate the three rows that correspond to a single climatic measure (e.g. DJF_temperature).

---

## Author Response (AR2)

**Keery et al., Sensitivity of the Eocene Climate to CO2 and Orbital Variability**

Response to M. Crucifix (Referee)

Referee comments in black

Author responses in red

Report #2
Submitted on 02 Nov 2017
Referee #1: Michel Crucifix, michel.crucifix@uclouvain.be

The authors have provided a revised version of their manuscript. My review has concentrated mainly on the responses to questions. Even though the authors made a case that the climatological results they obtain are robust, there are still some contentious aspects on the experiment design and I will attempt here to make my case more explicitly.

The authors rebutted my comment on the 'a-physical' character of their experiment design as follows:

1. there are no combinations of these parameters which can be excluded for the early Eocene period
2. This approach does not make the assumption that the only effect of eccentricity on the Earth's climate is through its effect on the amplitude of the precession cycle

They argument one is a useful condition to avoid wasting computing time, but not sufficient to guarantee a good design. Condition 2. is perhaps true but in fact comes at the price of wrong assumptions.

I agree that my condemnation of the design as 'a-physical' was not the most explicit, so I will rephrase my argument differently.

Latin Hypercube and related techniques (maxi-min, etc.) are mathematically justified by the hypothesis that distances in the input space (here $e$, $\varpi$, $\varepsilon$) translate _a priori_ into distances in the output space. Of course this is never a posteriori quite true (hence the interest of actually doing the experiment), but this hypothesis is the prior assumption which precisely justifies the interest of maximising distances in the input space (this is, _a priori_, the best way to make experiments maximally informative). See textbooks on this (e.g. Santner, Williams and Notz).

A LHS with maximisation of Euclidian distances in the $\{e,\varpi,\varepsilon\}$ space, as done here, therefore implies two wrong assumptions: there is a non-zero distance between two experiments with different $\varpi$ but zero eccentricity (while the insolation input is rigorously the same), and there is a twice greater distance between $\varpi$=359 degrees and $\varpi$=0 degrees, than between, say $\varpi$=100 degrees and $\varpi$=280 degrees. Both are wrong, not because of assumptions on model response, but because $\varpi$ is (1) an angle which (2) loses meaning at $e=0$.

On the other hand, as Devleeschouwer or Araya-Melo have shown, an $\{e\sin\varpi,e\cos\varpi\}$ input space does not prevent at all to detect effects of eccentricity. However, because of the geometry of insolation, we can never expect the same climate in two experiments with both high eccentricity but two opposite phases of $\varpi$. Net effects of eccentricity tend to come as the result of a non-symmetrical (non-linear) effects of positive and negative climatic precession anomalies (a form of rectification), or possibly more subtle effects in the tropics which could also be detected with a $\{e\sin\varpi,e\cos\varpi\}$ experiment design.

Of course I do not want to block a nice paper for that reason alone, but I cannot accept the line of defense that LHS sampling a $\{e,\varpi\}$ space is a good option, out of the fear of spreading bad practice.

In his responses to our paper, Michel Crucifix has drawn attention to a weakness in our method. Our selection of values for precession, an angular parameter, is from the linear range 0-360°, with the consequence that the maximin criterion within the Latin hypercube algorithm is incorrectly calculated. We accept this criticism, and have added a sentence to the text to clarify this point, and our belief that the effect of this is minor.

> We note that our selection of values for $\omega$, an angular parameter, is from 0-360°, treated as a linear range, with the consequence that the maximin criterion within the Latin hypercube algorithm is incorrectly calculated. However, given the dimensionality of our experimental design, this is unlikely to result in a significant reduction in the efficiency with which design points are distributed throughout the very sparsely populated state-space.

We thank Crucifix for clarifying that the approach presented in works which he has co-authored (Araya-Melo et al., 2015; Bounceur et al., 2015), in which independent values of $e\sin\omega$, $e\cos\omega$ and $\varepsilon$ are sampled, with rejection of absolute values of $e\sin\omega$ and $e\cos\omega$ which equal or exceed the maximum value of $e$, *does* allow values of $e$ and $\omega$ for any design point to be identified by trigonometric analysis, while efficiently sampling the state space. We have amended the paper to clarify this point. We have deleted the second sentence in the following paragraph:

> We include $e$ and $\omega$ as separate and independent forcing parameters, rather than combined as the precession index, or in the form $e\cos\omega$.

We have also added two sentences:

> We draw readers' attention to an approach presented by Bounceur et al. (2015), in which independent values of $e\sin\omega$, $e\cos\omega$ and $\varepsilon$ are sampled, with rejection of absolute values of $e\sin\omega$ and $e\cos\omega$ which equal or exceed the maximum value of $e$. This experimental design allows values of $e$ and $\omega$ for any design point to be identified by trigonometric analysis, while efficiently sampling the state space.

We are pleased that Crucifix does not wish to block our paper on this issue alone.

**Keery et al., Sensitivity of the Eocene Climate to CO2 and Orbital Variability**

Response to D. De Vleeschouwer (Referee)

Referee comments in black

Author responses in red

**Review "Sensitivity of the Eocene Climate to CO2 and orbital variability", by John S. Keery et al. (2017, Climate of the Past, cp-2017-60). Second Round.**

The authors satisfactory answered my major concerns, raised in my review of the first version of this manuscript. I particularly appreciate the way the authors tackled the disparity in time-scales between CO2 and orbital variability, for example by adding the color scale to Figs. 5 and 6, and by integrating new Figures 12, 13 and 14 into the revised version of the manuscript.

However, I have two small questions regarding the latter three Figures.

1. In the text, the authors write "*In each of the monsoon areas, the increase in precipitation ... is more pronounced at high atmospheric concentration of CO2*". But, according to Figure 14, the American monsoon is "*wettest*" under low CO2 forcing. Is this a mistake in color-coding, or does the American monsoon system really respond differently to CO2 forcing? If the latter is true, this should be discussed in much more detail.

The colour-coding is correct. The error was in our description, which we have corrected and expanded. See our response to the next point.

2. Figures 12, 13, 14 all show some degree of non-stationarity: the monsoonal index seems to depend not only on the precession index, but also on whether the precession index is increasing or decreasing. In other words, the emulated values of the monsoon index do not represent a line, but an ellipse. This is a very interesting result, but I do not fully understand how the authors came to this result, and I would like to read more about these values were obtained. Please discuss which limb of the ellipse represents increasing precession index (and vice versa), and please discuss why the American monsoon system seems to be more "non-stationary", compared to the African monsoon system.

We have added Table 7, showing all of the terms in the linear models for the three monsoon indices, to assist in an expanded description of these results. Figures 12, 13 and 14 have been revised to indicate which parts of the plots represent values of $\omega$, and positions of perihelion at the solstices and equinoxes. We have added a sentence to Section 3.2.2 for clarification:

The precession index is at its maximum value when perihelion occurs at the December solstice, its minimum value when perihelion is at the June solstice, and has a value of 0.0 when perihelion is at either the March or September equinox.

We have revised and expanded our description of the emulated results:

All of the terms in the linear models derived from the forcing factors and the three monsoon indices are shown in Table 7. The Asian and African models are dominated by precession terms, roughly equally distributed between first order $\sin(\omega)$ and the cross product of $e$ and $\sin(\omega)$, with $|\sin(\omega)|$ being approximately five times, and eight times larger than $|\cos(\omega)|$ for the Asian and African models respectively. The American model identifies significant influence of $CO_2$, in both the negative first order, and positive second order terms, with a similar magnitude of influence from combined precession terms, and with $|\sin(\omega)|$ being approximately three times larger than $|\cos(\omega)|$. All of the models have small contributions from first or second order, or cross products of $\varepsilon$, and from those terms of $e$, in addition to

significant contributions from $e\sin(\omega)$. The terms in the models clearly reflect the relationships between the three monsoon indices and the two forcing factors $CO_2$ and $e\sin(\omega)$ shown in Fig. 6.

We apply these linear models as emulators to estimate values of monsoon indices corresponding to the full range of precession ($\omega$), with eccentricity fixed at its high limit of 0.06, low and high values of $CO_2$ (300 ppm and 3000 ppm), and low and high values of obliquity (22.0° and 24.5°). Precession index ($e\sin\omega$) and emulated values of the Asian, African and American monsoon indices for all four combinations of high and low $CO_2$ and obliquity are plotted in Figures 12, 13 and 14 respectively. The elliptical form of each of the plots is controlled by model terms which include $\cos(\omega)$, and which identify seasonal processes in the development of the monsoons. Running each of the emulators with all of the terms in $\cos(\omega)$ excluded, generates points on a straight line between each apex of the ellipses generated by the full emulator. In each of the 12 plots in Figs. 12-14, $\omega$ increases anticlockwise from a value of 0° in the centre of the lower arc of the ellipse (with perihelion at the March equinox), through a value of 180° in the centre of the upper arc (with perihelion at the September equinox). Relationships between the precession index and the monsoon indices which are visually suggested in Figure 6 are shown with clear structure in Figures 12, 13 and 14. In each of the monsoon areas, the highest levels of precipitation occur when perihelion coincides with the summer solstice, in June for the Asian monsoon in the Northern Hemisphere, and in December for the African and American monsoons in the Southern Hemisphere. For the Asian and African monsoons, precipitation is increased by high $CO_2$, particularly when perihelion is at the summer solstice, but for the American monsoon, high $CO_2$ decreases precipitation. The plots of the emulated African and American monsoons (Figs. 13 and 14) show the lowest and highest degrees of non-stationarity respectively, due to the relative magnitude of the $\cos(\omega)$ terms in the linear models.

Once the authors will have addressed these points of discussion regarding Figs. 12 – 14, I consider this manuscript suitable for publication in *Climate of the Past*.

**Technical corrections**

**Page 3, line 18:** $^{18}O$ instead of $^{13}O$

We have corrected this error.

**Page 5 and throughout the paper:** There are a lot of different opinions on how to write 1000 years in a geologic scientific context. However, the "k" shown in upper case is an incorrect usage. I prefer the use of "kyr" for durations and relative time, and "ka" for thousand years ago (i.e. absolute time). See

http://www.ldeo.columbia.edu/~ncb/Selected_Articles_all_files/25_Stratigraphy.6.100.pdf

We have amended the paper throughout, in line with this constructive suggestion.

**Page 4, lines 25 – 30**: I still do not understand why a discussion of cyclostratigraphic tuning techniques is relevant for this work.

We accept that this brief outline is unnecessary for readers with the reviewer's expertise in orbital effects and cyclostratigraphy, but since the journal has a readership working in a broad range of disciplines, we believe that this very short section may be helpful in clarifying why we have only relative, rather than absolute ages, for events before 50 Ma.

**Table 6**: Draw horizontal lines under the PC3 rows, so to visually separate the three rows that correspond to a single climatic measure (e.g. DJF_temperature).

We agree that horizontal lines separating the climatic measures would improve the clarity of this table, so we have added them, but we note that the guidelines provided by Copernicus declare: "Horizontal lines should normally only appear above and below the table, and as a separator between the head and the main body of the table. Vertical lines must be avoided." We trust that the added lines will be accepted.

[revised manuscript text omitted]